# The value of initiating a pursuit in temporal decision-making

**Elissa Sutlief[1†], Charlie Walters[2,3†], Tanya Marton[4,5]*, Marshall G Hussain Shuler[6,7]***

[1]Department of Neuroscience, Johns Hopkins University School of Medicine, Baltimore, United States; [2]Kavli Neuroscience Discovery Institute, Johns Hopkins University, Baltimore, United States; [3]Department of Neuroscience, Johns Hopkins University School of Medicine, Baltimore, United States; [4]Department of Neuroscience, Johns Hopkins University School of Medicine, Baltimore, United States; [5]Microsoft, Seattle, United States; [6]Kavli Neuroscience Discovery Institute, Johns Hopkins University, Baltimore, United States; [7]The Department of Neuroscience, Johns Hopkins University School of Medicine, Baltimore, United States

## eLife Assessment

The paper presents a **valuable** theoretical treatment of the role of passage of time in optimal decision strategies in pursuit based tasks. The computational evidence and methodologies employed are novel, and the authors offer **solid** evidence for the majority of the claims.

**\*For correspondence:**
tanya.marton@gmail.com (TM);
shuler@jhmi.edu (MGHS)

[†]These authors contributed equally to this work

**Abstract** Reward-rate maximization is a prominent normative principle in behavioral ecology, neuroscience, economics, and AI. Here, we identify, compare, and analyze equations to maximize reward rate when assessing whether to initiate a pursuit. In deriving expressions for the value of a pursuit, we show that time's cost consists of both apportionment and opportunity cost. Reformulating value as a discounting function, we show precisely how a reward-rate-optimal agent's discounting function (1) combines hyperbolic and linear components reflecting apportionment and opportunity costs, and (2) is dependent not only on the considered pursuit's properties but also on time spent and rewards obtained outside the pursuit. This analysis reveals how purported signs of suboptimal behavior (hyperbolic discounting, and the Delay, Magnitude, and Sign effects) are in fact consistent with reward-rate maximization. To better account for observed decision-making errors in humans and animals, we then analyze the impact of misestimating reward-rate-maximizing parameters and find that suboptimal decisions likely stem from errors in assessing time's apportionment—specifically, underweighting time spent outside versus inside a pursuit—which we term the 'Malapportionment Hypothesis'. This understanding of the true pattern of temporal decision-making errors is essential to deducing the learning algorithms and representational architectures actually used by humans and animals.

## Introduction

What is the worth of a pursuit? At the most universal level, temporal decision-making regards weighing the return of pursuits against their cost in time. The fields of economics, psychology, behavioral ecology, neuroscience, and artificial intelligence have endeavored to understand how animals, humans, and learning agents evaluate the worth of pursuits: how they factor the cost of time in temporal decision-making. A central step in doing so is to identify a normative principle and then to solve for how an agent, abiding by that principle, would best invest time in pursuits that compose a world. A normative principle with broad appeal identified in behavioral ecology is that of reward-rate

maximization, as expressed in Optimal Foraging Theory (OFT), where animals seek to maximize reward rate while foraging in an environment (*Charnov, 1976a*; *Charnov, 1976b*; *Krebs et al., 1977*; *Pyke et al., 1977*; *Pyke, 1984*). Solving for the optimal decision-making behavior under this objective provides the means to examine the curious pattern of adherence and deviation that is exhibited by humans and animals with respect to that ideal behavior. This difference provides clues into the process that animals and humans use to learn the value of, and represent, pursuits. Therefore, it is essential to analyze reward-rate-maximizing solutions for the worth of initiating a pursuit to clarify what behavioral signs are—and are not—deviations from optimal performance in the identification of the process (and its sources of error) actually used by humans and animals.

## Equivalent immediate reward (subjective value, $sv$)

To ask, 'what is the value of a pursuit?' is to quantify by some metric the worth of a future state—the pursuit's outcome—at the time of a prior one, the pursuit's initiation. A sensible metric for the worth of a pursuit is the magnitude of immediate reward that would be treated by an agent as equivalent to a policy of investing the requisite time in the pursuit and obtaining its reward. This *equivalent immediate reward,* as judged by the agent, is the pursuit's 'Subjective Value' (sv), in the parlance of the field (*Mischel et al., 1969*). It is widely assumed that decisions about what pursuits should be taken are made on the basis of their subjective value (*Niv, 2009*). However, a decision-making algorithm need not calculate subjective value in its evaluation of the worth of initiating a pursuit. It could, for instance, assess the reward rate of the pursuit relative to the reward rate received in the world as a whole. Indeed, algorithms leading to reward-rate optimization can arise from different underlying processes, each with their own controlling variables. Nonetheless, any algorithm's evaluation can be re-expressed in terms of equivalent immediate reward, providing a ready means to compare evaluation across different learning algorithms and representational architectures as biologically realized in animals and humans or as artificially implemented in silico.

## Decisions to initiate pursuits

As decisions occur at branch points between pursuits, the value of initiating a pursuit is of particular importance as it is on this basis that an agent would decide (1) whether to accept or *forgo* an offered pursuit; or, (2) how to *choose* between mutually exclusive pursuits. Although 'Forgo' decisions are regarded as near-optimal, as in prey selection (*Krebs et al., 1977*; *Stephens and Krebs, 1987*; *Blanchard and Hayden, 2014*), 'Choice' decisions—as commonly tested in laboratory settings—reveal a suboptimal bias for smaller-sooner (SS) rewards when selection of later-larger rewards would maximize the reward rate while foraging in an environment, that is the 'global reward rate' (*Logue et al., 1985*; *Blanchard and Hayden, 2015*; *Carter and Redish, 2016*; *Kane et al., 2019*). This curious pattern of behavior, wherein Forgo decisions can present as optimal while Choice decisions as suboptimal, poses a challenge to any theory purporting to rationalize temporal decision-making as observed in animals and humans.

## Temporal discounting functions

Historically, temporal decision-making has been examined using a temporal discounting function to describe how delays in rewards influence their valuation. Temporal discounting functions describe the subjective value of an offered reward as a function of when the offered reward is realized. To isolate the form of discount rate from any difference in reward magnitude and sign, subjective value is commonly normalized by the reward magnitude when comparing subjective value-time functions (*Strotz, 1955*; *Jimura et al., 2009*). Therefore, we use the convention that temporal discounting functions are the magnitude-normalized subjective value-time function (*Strotz, 1955*). An understanding of the form of temporal discounting has important implications in life, as steeper temporal discounting has been associated with many negative life outcomes (*Bretteville-Jensen, 1999*; *Critchfield and Kollins, 2001*; *Bickel et al., 2007*; *Bickel et al., 2012*; *Story et al., 2014*), most notably the risk of developing an addiction. Psychologists and behavioral scientists have long found that animals' temporal discounting in intertemporal choice tasks is well-fit by a hyperbolic discounting function (*Ainslie, 1974*; *Mazur, 1987*; *Richards et al., 1997*; *Monterosso and Ainslie, 1999*; *Green and Myerson, 2004*; *Hwang et al., 2009*; *Louie and Glimcher, 2010*). Other examples of motivated behavior also show hyperbolic temporal discounting (*Haith et al., 2012*).

Often, this perspective assumes that the delay in and of itself devalues a pursuit's reward, failing to carefully distinguish the impact of its delay from the impact of the time required and reward obtained *outside* the considered pursuit. As a result, the discounting function tends to be treated as a process unto itself rather than the consequence of a process. Consequently, the field has concerned itself with the form of the discounting function—exponential (*Glimcher et al., 2007*; *McClure et al., 2007*), hyperbolic (*Rachlin et al., 1972*; *Ainslie, 1975*; *Thaler, 1981a*; *Mazur, 1987*; *Benzion et al., 1989*; *Green et al., 1994*; *Frederick et al., 2002*; *Kobayashi and Schultz, 2008*; *Calvert et al., 2010*), pseudo-hyperbolic (*Laibson, 1997*; *Montague et al., 2006*; *Berns et al., 2007*), etc., as either derived from some normative principle, or as fit to behavioral observation. An exponential discounting function, for instance, was derived by Samuelson from the normative principle of time consistency (*Samuelson, 1937*) and is widely held as rational (*Samuelson, 1937*; *Koopmans, 1960*; *Laibson, 1997*; *Montague and Berns, 2002*; *McClure et al., 2004*; *Mazur, 2006*; *Schweighofer et al., 2006*; *Berns et al., 2007*; *McClure et al., 2007*; *Nakahara and Kaveri, 2010*; *Kane et al., 2019*), and by implication, reward-rate maximizing. Observed temporal decision-making behavior, however, routinely exhibits time inconsistencies (*Strotz, 1955*; *Ainslie, 1975*; *Laibson, 1997*; *Frederick et al., 2002*) and is better fit by a hyperbolic discounting function (*Ainslie, 1975*; *Mazur et al., 1985*; *Frederick et al., 2002*; *Green and Myerson, 2004*), and on that contrasting basis, humans and animals have commonly been regarded as *irrational* (*Thaler, 1981a*; *Loewenstein and Thaler, 1989*; *Loewenstein and Prelec, 1992*; *Frederick et al., 2002*; *Baker et al., 2003*; *Estle et al., 2006*; *Kalenscher and Pennartz, 2008*). In addition, the case that humans and animals are irrational is, ostensibly, furthered by the observation of the 'Magnitude Effect' (*Benzion et al., 1989*; *Green et al., 1994*; *Green et al., 1997*; *Frederick et al., 2002*; *Baker et al., 2003*; *Estle et al., 2006*; *Yi et al., 2006*; *Grace et al., 2012*; *Kinloch and White, 2013*) and the 'Sign Effect' (*Thaler and Shefrin, 1981b*; *Loewenstein and Thaler, 1989*; *Loewenstein and Prelec, 1992*; *Frederick et al., 2002*; *Baker et al., 2003*; *Estle et al., 2006*; *Kalenscher and Pennartz, 2008*) where the apparent discounting function is affected by the magnitude and the sign of the offered pursuit's outcome, respectively), and the 'Delay Effect', where preference between pursuits can switch as the time required for their obtainment changes (*Rachlin et al., 1972*; *Ainslie, 1974*; *Bateson and Kacelnik, 1996*; *Stephens and Anderson, 2001*; *Frederick et al., 2002*; *Hayden and Platt, 2007b*; *McClure et al., 2007*; *Carter et al., 2015*; *Carter and Redish, 2016*).

Here, we aim to identify equations for evaluating the worth of initiating pursuits that an agent could implement to enable reward-rate maximization. We wish to gain deeper insight into how a considered pursuit, with its defining features (its reward and time), relates to the world of pursuits in which it is embedded in determining the pursuit's worth. Specifically, we investigate how pursuits and the pursuit-to-pursuit structure of a world interact with policies of investing time in particular pursuits to determine the global reward rate reaped from an environment. We aim to provide greater clarity into what constitutes time's cost and how it can be understood with respect to the reward and temporal structure of an environment and to counterfactual time investment policies. We propose that, by determining optimal decision-making equations and converting them to their equivalent subjective value and temporal discounting functions, actual (rather than asserted) deviations from optimality exhibited by humans and animals can be truly determined. We speculate that purported anomalies deviating from ostensibly 'rational' decision-making may in fact be consistent with reward-rate optimization. Further, by identifying parameters enabling reward-rate maximization and assessing resulting errors in valuation caused by their misestimation, we aim to gain insight into which parameters humans and animals may (mis)-represent that most parsimoniously explains the pattern of temporal decision-making actually observed.

## Results

To gain insight into the manner by which animals and humans attribute value to pursuits, it is essential to first understand how a reward-rate-maximizing agent would evaluate the worth of any pursuit within a temporal decision-making world. Here, by considering *Forgo* and *Choice* temporal decisions, we re-conceptualize how an ideal reward-rate-maximizing agent ought to evaluate the worth of initiating pursuits. We begin by formalizing temporal decision-making worlds as constituted of pursuits, with pursuits described as having reward rates and weights (their relative occupancy). Then, we examine *Forgo* decisions to examine what composes the cost of time and how a policy of taking/

forgoing pursuits factors into the global reward rate of an environment and thus the worth of a pursuit. Having done so, we derive two equivalent expressions for the worth of a pursuit and from them re-express the worth of a pursuit as its equivalent immediate reward (its 'subjective value') in terms of the global reward rate achieved under policies of (1) accepting or (2) forgoing the considered pursuit type. We next examine *Choice* worlds to investigate the apparent nature of a reward-rate optimizing agent's temporal discounting function. Finally, having identified reward-rate maximizing equations, we examine what parameter misestimation leads to suboptimal pursuit evaluation that best explains behavior observed in humans and animals. Together, by considering the temporal structure of a time investment world as one composed of pursuits described by their rates and weights, we seek to identify equations for how a reward-rate-maximizing agent could evaluate the worth of any pursuit composing a world and how those evaluations would be affected by misestimation of enabling parameters.

## Temporal decision worlds are composed of pursuits with reward rates and weights

A temporal decision-making world is one composed of pursuits. A *pursuit* is a defined path over which an agent can traverse by investing time that often (but not necessarily) results in reward but which always leads to a state from which one or more potential pursuits are discoverable. Pursuits have a *reward magnitude* (*r*) and a *time* (*t*). A pursuit therefore has (1) a *reward rate* (ρ, rho) and (2) a *weight* (*w*), being its relative occupancy with respect to all other pursuits. Here we refer to the defining features of a pursuit (its reward, time, reward rate, or weight) by subscripting the pursuit's name or type to each feature's symbol (r, t, rho or w, respectively), for example ($\rho_{\text{Pursuit}}$, $w_{\text{Pursuit}}$). In this way, the pursuit structure of temporal decision-making worlds, and the qualities defining pursuits, can be adequately referenced.

The temporal decision-making worlds considered are recurrent in that an agent traversing a world under a given policy will eventually return back to its current location. As pursuits constitute an environment, the environment itself then has a reward rate, the 'global reward rate' $\rho_g$, achieved under a given decision policy, $\rho_g Policy$. Whereas the global reward rate realized under a given policy of choosing one or another pursuit path may or may not be reward-rate optimal, the global reward rate achieved under a reward-rate maximal policy will be denoted as $\rho_g^*$.

## Forgo and Choice decision topologies

Having established a nomenclature for the properties of a temporal decision-making world, we now identify two fundamental types of decisions regarding whether to initiate a pursuit: 'Forgo' decisions, and 'Choice' decisions. In a Forgo decision (*Figure 1*, left), the agent is presented with one of possibly many pursuits (and see *Figure 3—figure supplement 1* and *Figure 3—figure supplement 2*) that can either be accepted or rejected. After either the conclusion of the pursuit, if accepted, or immediately after rejection, the agent returns to a pursuit by default (the 'default' pursuit). This default pursuit effectively can be a waiting period over which reward could be received and reoccurs until the next pursuit opportunity becomes available. Rejecting the offered pursuit constitutes a policy of spending less time to make a traversal across that decision-making world, whereas accepting the offered pursuit constitutes a policy of spending more time to make a traversal. In a Choice decision (*Figure 1*, right), the agent is presented with a choice between at least two simultaneous and mutually exclusive pursuits, typically differing in their respective rewards' magnitudes and delays. Under any decision, upon exit from a pursuit, the agent returns to the same environment that it would have entered were the pursuit rejected. In the Forgo case in *Figure 1*, a policy of spending less time to traverse the world by rejecting the purple pursuit to return to the gold pursuit—and thus obtaining a smaller amount of reward (left)—must be weighed against a policy of acquiring more reward by accepting the purple pursuit at the expense of spending more time to traverse the world (right). In the Choice case in *Figure 1*, a policy of spending less time to traverse the world (left) by taking the smaller-sooner (SS) pursuit (aqua) must be weighed against a policy of spending more time to traverse the world (right) by accepting the larger-later (LL) pursuit (purple).

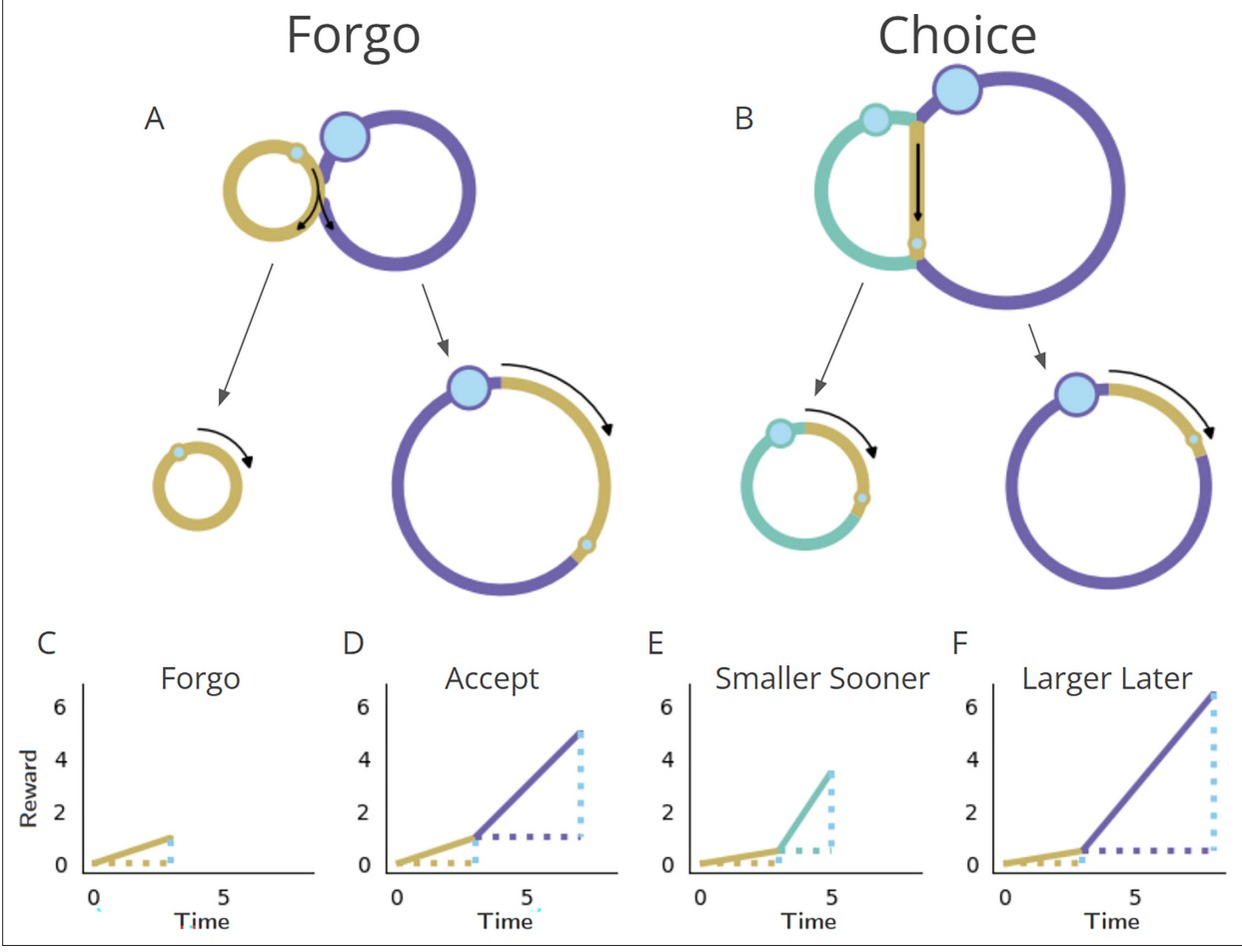

**Figure 1.** Fundamental classes of temporal decision-making regarding initiating a pursuit: 'Forgo' and 'Choice'. (**A, B**) (Top row) Topologies. The temporal structure of worlds exemplifying Forgo (**A**) and Choice (**B**) decisions mapped as their topologies. *Forgo*: A Forgo decision to accept or reject the purple pursuit. When exiting the gold pursuit having obtained its reward (small blue circle), an agent is faced with (1) a path to re-enter gold, or (2) a path to enter the purple pursuit, which, on its completion, re-enters gold. *Choice*: A Choice decision between an aqua pursuit, offering a small reward after a short amount of time, or a purple pursuit offering a larger amount of reward after a longer time. When exiting the gold pursuit, an agent is faced with a path to enter (1) the aqua or (2) the purple pursuit, both of which lead back to the gold pursuit upon their completion. (Middle row) Policies. Decision-making policies chart a course through the pursuit-to-pursuit structure of a world. Policies differ in the reward obtained and in the time required to complete a traversal of that world. Policies of investing less (left) or more (right) time to traverse the world are illustrated for the considered Forgo and Choice worlds. *Forgo*: A policy of rejecting the purple pursuit to re-enter the gold pursuit (left) acquires less reward though it requires less time to make a traversal of the world than a policy of accepting the purple option (right). *Choice*: A policy of choosing the aqua pursuit (left) results in less reward though it requires less time to traverse the world than a policy of choosing the purple pursuit (right). (**C-F**) Time/reward investment. The times (stippled horizontal lines, colored by pursuit) and rewards (stippled vertical blue lines) of pursuits, and their associated reward rates (solid lines) acquired under a policy of forgo (**C**) or accept (**D**) in the Forgo world, or, of choosing the smaller-sooner (**E**) or later-larger pursuit (**F**) in the Choice world.

## Behavioral observations under Forgo and Choice decisions

These classes of temporal decisions have been investigated by ecologists, behavioral scientists, and psychologists for decades. Forgo decisions describe instances likened to prey selection (*Krebs et al., 1977*; *Stephens and Krebs, 1987*; *Blanchard and Hayden, 2014*). Choice decisions have extensively been examined in intertemporal choice experiments (*Rachlin et al., 1972*; *Ainslie, 1974*; *Bateson and Kacelnik, 1996*; *Stephens and Anderson, 2001*; *Frederick et al., 2002*; *Hayden and Platt, 2007b*; *McClure et al., 2007*; *Carter et al., 2015*; *Carter and Redish, 2016*). Experimental observation in temporal decision-making demonstrates that animals are optimal (or virtually so) in Forgo (*Krebs et al., 1977*; *Stephens and Krebs, 1987*; *Blanchard and Hayden, 2014*), taking the offered pursuit when its rate exceeds the 'background' reward rate, and are as if sub-optimally impatient in

Choice, selecting the SS pursuit when the LL pursuit is reward-rate-maximizing (*Logue et al., 1985*; *Blanchard and Hayden, 2015*; *Carter and Redish, 2016*; *Kane et al., 2019*).

## Deriving optimal policy from Forgo decision-making worlds

We begin our examination of how to maximize the global reward rate reaped from a landscape of rewarding pursuits by examining Forgo decisions. A general formula for the global reward rate of an environment in which agents must invest time in obtaining rewards is needed in order to formally calculate a policy's ability to accumulate reward. Optimal policies maximize reward accumulation over the time spent foraging in that environment. In a Forgo decision, an agent is faced with the decision to take, or to *forgo*, pursuit opportunities. We sought to determine the reward rate an agent would achieve were it to pursue rewards with magnitudes $r_1, r_2, \ldots r_n$ each requiring an investment of time $t_1, t_2, \ldots t_n$. At any particular time, the agent is either (1) investing time in a pursuit of a specific reward and time, or (2) available to encounter and take new pursuits from a pursuit to which it defaults. With the assumption that reward opportunities become randomly encountered by the agent at a frequency of $f_1, f_2, \ldots f_n$ from the default pursuit, it becomes possible to calculate the total global reward rate of the environment, $\rho_g$, as in *Equation 1* (The global reward rate under multiple pursuits, Appendix 1)…

$$\rho_g = \frac{\sum_{i=1}^{n} f_i r_i + \rho_d}{\sum_{i=1}^{n} f_i t_i + 1} \tag{1}$$

*Equation 1*.

…where $\rho_d$ is the rate of reward attained in the default pursuit per unit time in the default pursuit. Should rewards not occur while in the default pursuit, $\rho_d$, will be zero. *Equation 1* allows for the calculation of the global reward rate achieved by any policy accepting a particular set of pursuits from the environment. This derivation of global reward rate is akin to those similarly derived for prey selection models (see *Charnov and Orians, 1973* and *Stephens and Krebs, 1987*).

## Parceling the world into the considered pursuit type ('in' pursuit) and everything else ('out' of pursuit)

In order to simplify representations of policies governing any given pursuit opportunity, we reformulate the above expression for global reward rate, $\rho_g$, from the perspective of a policy of accepting any given pursuit (*Equation 2*, The global reward rate expressed as 'in' and 'outside' considered pursuit type (Appendix 2)). The environment may be parcellated into the time spent and rewards achieved *inside* the considered pursuit on average, for every instance that time is spent and rewards achieved *outside* the considered pursuit, on average. We can pull out the inside reward ($r_{in}$) and inside time ($t_{in}$) from the equation above to isolate the inside and outside components of the equation.

$$\rho_g = \frac{r_{in} + \frac{\sum_{i \neq \in}^{n} f_i r_i + \rho_d}{f_{in}}}{t_{in} + \frac{\sum_{i \neq \in}^{n} f_i t + 1}{f_{in}}} \tag{2}$$

*Equation 2*.

From there, we define $t_{out}$ as the average time spent outside the considered pursuit for each instance that the considered pursuit is experienced (*Equation 3*, The average time spent outside of the considered pursuit type (Appendix 2)).

$$t_{out} = \frac{\sum_{i \neq in} f_i t_i + 1}{f_{in}} \tag{3}$$

*Equation 3*.

Similarly, the outside reward, $r_{out}$, encompasses the average amount of reward collected from all sources outside the considered pursuit (*Equation 4*, The average reward collected outside of the considered pursuit type (Appendix 3)).

$$r_{out} = \frac{\sum_{i \neq in} f_i r_i + \rho_d}{f_{in}} \qquad (4)$$

*Equation 4.*

Parceling a pursuit world into a considered pursuit (all instances 'inside' the considered pursuit type) and everything else (i.e. everything 'outside' the considered pursuit type), then gives the generalized form for the reward rate of an environment under a given policy (*Equation 5*, Global reward rate with respect to the considered pursuit type (Appendix 3)) as…

$$\rho_g = \frac{r_{in} + r_{out}}{t_{in} + t_{out}} \qquad (5)$$

*Equation 5.*

…which depends on the average reward earned and the average time spent between opportunities to make the decision, in addition to the average reward returned and average time spent in the considered pursuit (Appendix 3 and *Figure 2*).

*Figure 2D* depicts the global reward rate achieved with respect to the time and reward obtained from a considered pursuit ('Inside') and the time and reward obtained outside that considered pursuit type, that is that pursuit's ('Outside'). By so parsing the world into 'in' and 'outside' the considered pursuit, it can also be appreciated from *Figure 2D* that the fraction of time in the environment invested in the considered option, can be expressed as $w_{in} = \frac{t_{in}}{t_{in}+t_{out}}$ , and the fraction of time spent outside the considered pursuit as $1 - w_{in} = \frac{t_{out}}{t_{in}+t_{out}}$ . A world can thus be understood in terms of its composing pursuits' reward rates and weights (their relative occupancy), with the global reward rate being a weighted average of the reward rate from the considered pursuit type, $\rho_{in} = \frac{r_{in}}{t_{in}}$ , and the reward rate outside the considered pursuit type, $\rho_{out} = \frac{r_{out}}{t_{out}}$.

$$\rho_g = w_{in} \cdot \rho_{in} + \left(1 - w_{in}\right) \cdot \rho_{out} \qquad (6)$$

*Equation 6.*

Therefore, the global reward rate is the sum of the local reward rates of the world's constituent pursuits under a given policy when weighted by their relative occupancy: the weighted average of the local reward rates of the pursuits constituting the world (*Equation 6*, Global reward rate is the sum of weighted inside and outside rates (Appendix 4)).

## Reward-rate optimizing Forgo policy: compare a pursuit's local reward rate to its outside reward rate

We can now compare two competing policies to identify the policy that maximizes reward rate, such that it is the maximum possible reward rate, $\rho_g^*$. A policy of taking or forgoing a given pursuit type may improve the reward rate reaped from the environment as a whole (*Figure 3*). Using *Equation 5*, the policy achieving the greatest global reward rate can be realized through an iterative process where pursuits with lower reward rates than the reward rate obtained from everything other than the considered pursuit type are sequentially removed from the policy. While *Figure 3* illustrates a simple topology for clarity, this process applies to topologies comprising any n-pursuits (*Figure 3—figure supplement 1*)**,** which may occur at varying frequency (*Figure 3—figure supplement 2*). The optimal policy for forgoing can therefore be calculated directly from the considered pursuit's reward rate, $\rho_{in}$, and the reward rate outside of that pursuit type, $\rho_{out}$. Global reward rate can be maximized by iteratively forgoing the considered pursuit if its reward rate is less than its outside reward rate, $\rho_{in} < \rho_{out}$, treating forgoing and taking a considered pursuit as equivalent when $\rho_{in} = \rho_{out}$, and taking the considered pursuit when $\rho_{in} > \rho_{out}$ (Appendix 5).

Following this policy would be equivalent to comparing the local reward rate of a pursuit to the global reward rate obtained under the reward-rate-optimal policy: forgo the pursuit when its local reward rate is less than the global reward under the reward-rate-optimal policy, $\rho_{in} < \rho_g^*$, take or forgo the pursuit when the reward rate of the pursuit is equal to the global reward rate under the optimal policy $\rho_{in} = \rho_g^*$, and take the pursuit when its local reward rate is more than the global reward rate under the reward-rate-optimal policy, $\rho_{in} > \rho_g^*$ (Appendix 5). The maximum reward rate reaped from the environment can thus be eventually obtained by comparing the local reward rate of a considered

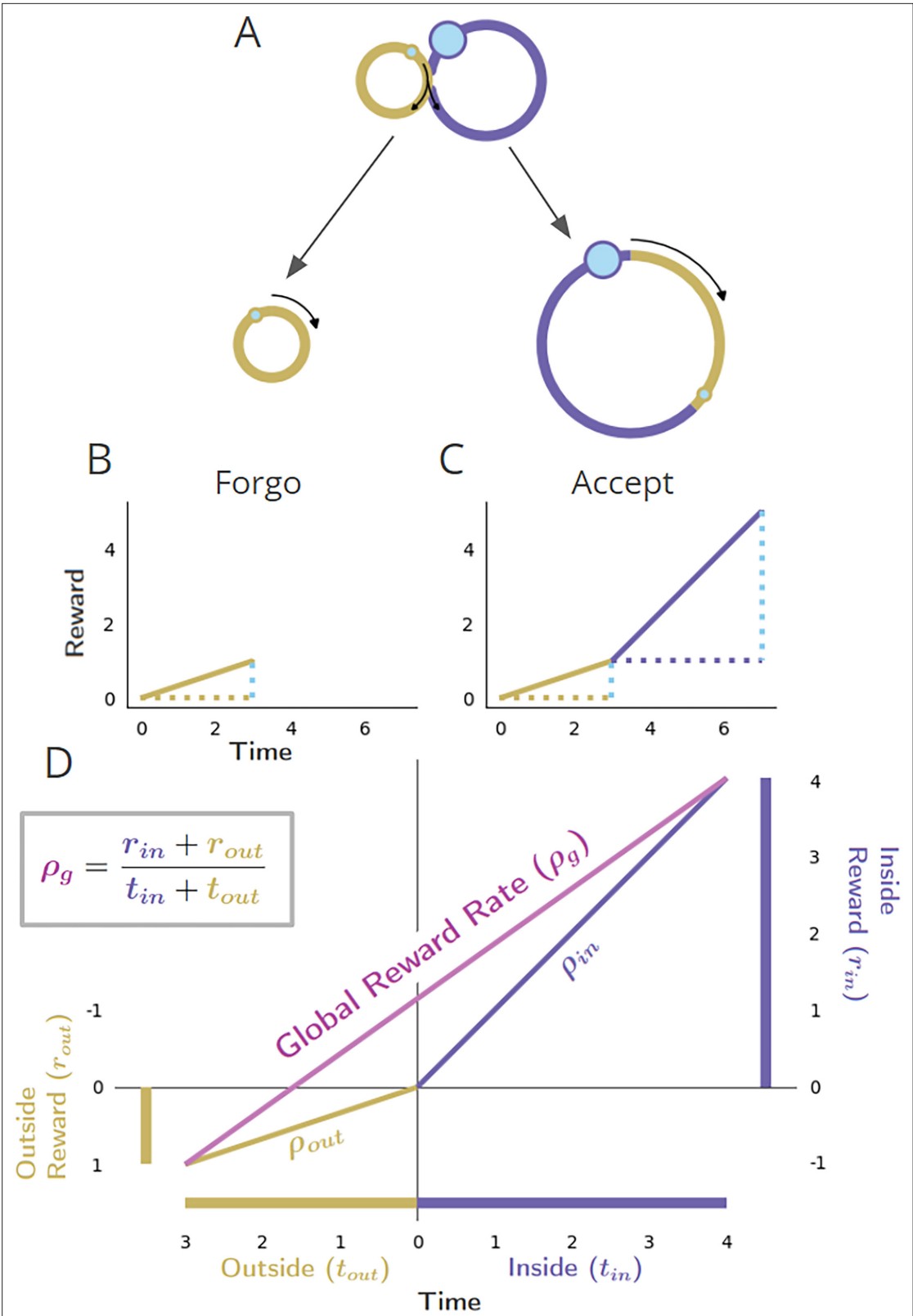

**Figure 2.** Global reward rate with respect to parceling the world into 'in' and 'outside' the considered pursuit. (**A-C**) as in *Figure 1* 'Forgo'. Conventions as in *Figure 1*. (**D**) The world divided into 'Inside' and 'Outside' the purple pursuit type, as the agent decides whether to forgo or accept. The axes are centered on the position of the agent, just before the purple pursuit, where the upper right quadrant shows the inside (purple) pursuit's average reward, time, and reward rate ($\rho_{in}$), while the bottom left quadrant shows the outside pursuit (gold) average reward, time, and reward rate ($\rho_{out}$). The

*Figure 2 continued on next page*

*Figure 2 continued*

global reward rate ( $\rho_g$ ) is shown in magenta, calculated from the boxed equation. The agent may determine the higher reward rate yielding policy by comparing the outside reward rate ( $\rho_{out}$ ) with the resulting global reward rate ( $\rho_g$ ) under a policy of accepting the considered pursuit.

pursuit to its outside reward rate (i.e. the global reward rate of a policy of *not* accepting the considered pursuit type).

### Equivalent immediate reward: the 'subjective value', *sv*, of a pursuit

Having recognized how a world can be decomposed into pursuits described by their rates and weights, and identifying optimal policies under Forgo decisions, we may now ask anew, 'What is the worth of a pursuit?' *Figure 2D* illustrates that the global reward rate obtained under a policy of taking a pursuit is not just a function of the time and return of the pursuit itself, but also the time spent and return gained outside of that pursuit type. Therefore, the worth of a pursuit relates to how much the pursuit would add (or detract) from the global reward rate realized in its acquisition.

## Subjective value of the considered pursuit with respect to the global reward rate

This relationship between a considered pursuit type, its outside, and the global reward rate can be re-expressed in terms of an immediate reward magnitude requiring no time investment that yields the same global reward rate as that arising from a policy of taking the pursuit (*Figure 4*). Thus, for any pursuit in a world, the amount of immediate reward that would be accepted in place of its initiation and attainment could serve as a metric of the pursuit's worth at the time of its initiation. Given the optimal policy above, an expression for this immediate reward magnitude can be derived (*Equation 7*, The subjective value of a pursuit expressed in terms of the global reward rate achieved under a policy of accepting that pursuit, Appendix 6). This *global reward-rate equivalent immediate reward* (see *Figure 4*) is the *subjective value* of a pursuit, $sv_{Pursuit}$ (or simply, *sv*, when the referenced pursuit can be inferred), as similarly expressed in prior foundational work (*McNamara, 1982*) and subsequent extensions (see *Fawcett et al., 2012*).

$$sv = r_{in} - \rho_g t_{in} \tag{7}$$

*Equation 7.*
The subjective value of a pursuit under the reward-rate-optimal policy will be denoted as $sv^*_{Pursuit}$.

$$sv^* = r_{in} - \rho_g^* t_{in}$$

The calculation of the subjective value of a pursuit, *sv*, quantifies precisely the worth of a pursuit in terms of an immediate reward that would result in the same global reward rate as that pursuant to its attainment. Thus, choosing either an immediate reward of magnitude *sv* or choosing to pursue the considered pursuit, investing the required time and acquiring its reward, would produce an equivalent global reward rate. An agent pursuing an optimal policy would find immediate rewards of magnitude less than *sv* less preferred than the considered pursuit and immediate rewards of magnitude greater than *sv* more preferred than the pursuit.

## The Forgo decision can also be made from subjective value

With this understanding, in the case that the considered pursuit's reward rate is greater than the optimal global reward rate, it will be greater than its outside reward rate, and therefore the subjective value under an optimal policy will be greater than zero (*Figure 3A*).

$\rho_{in} > \rho_g^* > \rho_{out} \rightarrow sv^* > 0, sv > 0$, choose considered pursuit (Appendix 7).

Should the considered pursuit's reward rate be equal to its outside reward rate, it will be equal to the optimal global reward rate, and the subjective value of the considered pursuit will be zero (*Figure 3B*).

$\rho_{in} = \rho_g^* = \rho_{out} \rightarrow sv^* = 0, sv = 0$, forgoing and choosing are equivalent (Appendix 7).

Finally, if the considered pursuit's reward rate is less than the outside reward rate, it must also be less than the global optimal reward rate; therefore, the subjective value of the considered pursuit under the optimal policy will be less than zero (*Figure 3C*). A negative subjective value thus indicates

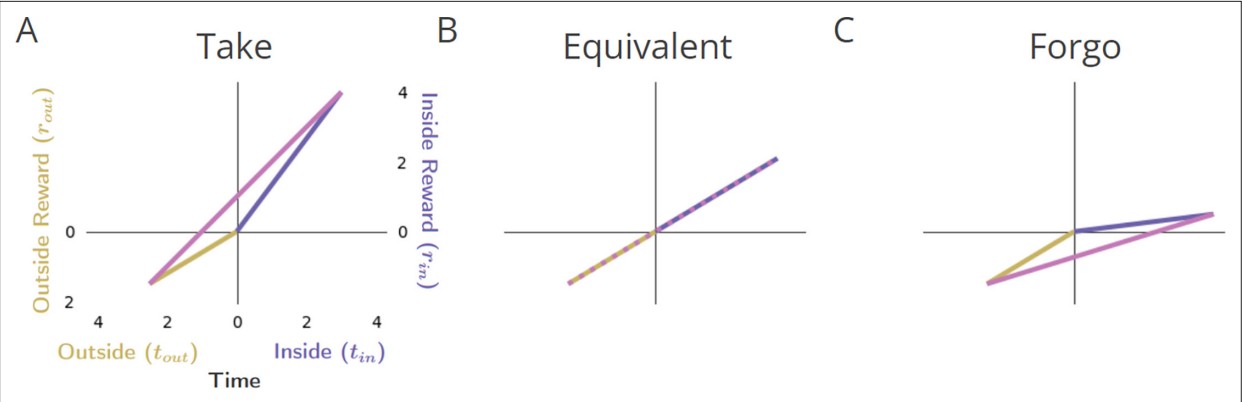

**Figure 3.** Forgo decision-making. (**A**) When the reward rate of the considered pursuit (slope of the purple line) exceeds that of its outside rate (slope of gold), the global reward rate (slope of magenta) will be greater than the outside, and therefore the agent should accept the considered pursuit. (**B**) When the reward rates inside and outside the considered pursuit are equivalent, the global reward rate will be the same when accepting or forgoing: the policies are equivalent. (**C**) When the reward rate of the considered pursuit is less than its outside rate, the resulting global reward rate if accepting the considered pursuit will be less than its outside reward rate and therefore should be forgone.

The online version of this article includes the following figure supplement(s) for figure 3:

**Figure supplement 1.** Forgo world with n-pursuits occurring at the same frequency.

**Figure supplement 2.** Forgo world with n-pursuits occurring at different frequencies.

that a policy of taking the considered pursuit would result in a global reward rate that is less than a policy of forgoing the considered pursuit. Equivalently, a negative subjective value can be considered the amount an agent ought be willing to pay to avoid having to take the considered pursuit.

$$\rho_{in} < \rho_g^* = \rho_{out} \rightarrow sv^* < 0, sv < 0, \text{ forgo considered pursuit (Appendix 7).}$$

While brains of humans and animals may not in fact calculate subjective value, converting to the equivalent immediate reward, $sv$, (1) makes connection to temporal decision-making experiments where such equivalences between delayed and immediate rewards are assessed, (2) serves as a common scale of comparison irrespective of the underlying decision-making process, and (3) deepens an understanding of how the worth of a pursuit is affected by the temporal structure of the environment's reward-time landscape.

## Subjective value with respect to the pursuit's outside: insights into the cost of time

To the latter point, *Equation 7* has a (deceptively) simple appeal: the worth of a pursuit ought be its reward magnitude less its cost of time (*Figure 5* Left Column). But what is the cost of time? The cost of time of a considered pursuit is the global reward rate of the world under a policy of accepting the pursuit, times the time that the pursuit would take, $\rho_g t_{in}$ (*Figure 5B*). Therefore, the equivalent immediate reward of a pursuit, its *subjective value*, corresponds to the subtraction of the cost of time from the pursuit's reward. The subjective value of a pursuit is how much *extra* reward is earned from the pursuit than would on average be earned by investing that amount of time, in that world, under a policy of accepting the considered pursuit.

While appealing in its simplicity, the terms on the right-hand side of *Equation 7*, $r_{in}$ and $\rho_g t_{in}$, lack independence from one another—the reward of the considered pursuit type contributes to the global reward rate, $\rho_g$. Subjective value can alternatively and more deeply be understood by re-expressing subjective value in terms that are independent of one another. Rather than expressing the worth of a pursuit in terms of the global reward rate obtained when accepting it, as in *Equation 7* (*Figure 5* left column), the worth of a pursuit can be expressed in terms of the reward obtained and time spent outside the considered pursuit type (*Figure 5* right column), as in *Equation 8* (Subjective value of a pursuit from the perspective of the considered pursuit and its outside (and see Appendix 8).

$$sv_{in} = \left( r_{in} - \frac{r_{out}}{t_{out}} t_{in} \right) \left( \frac{t_{out}}{t_{in} + t_{out}} \right) \tag{8}$$

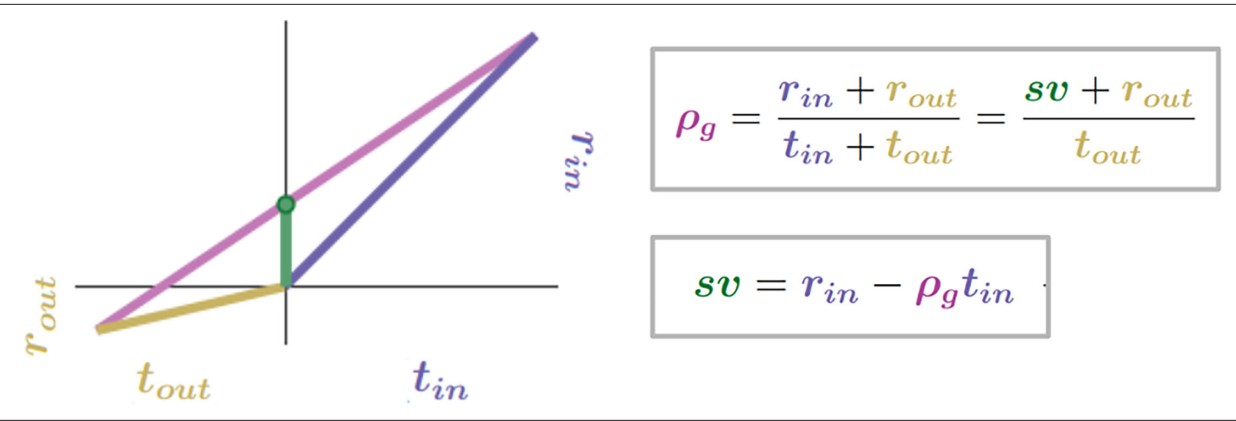

**Figure 4.** The subjective value (sv) of a pursuit is the global reward-rate-equivalent immediate reward magnitude. The subjective value (green bar) of a pursuit is that amount of reward requiring no investment of time that the agent would take as equivalent to a policy of accepting and acquiring the considered pursuit. For this amount to be equivalent, the immediate reward magnitude must result in the same global reward rate as that of accepting the pursuit. The global reward rate obtained under a policy of accepting the considered pursuit type is the slope of the line connecting the average times and rewards obtained in and outside the considered pursuit type. Therefore, the global reward rate-equivalent immediate reward (i.e. the subjective value of the pursuit) can be depicted graphically as the y-axis intercept (green dot) of the line representing the global reward rate achieved under a policy of accepting the considered pursuit. $r_{in} = 4$, $t_{in} = 4$, $r_{out} = 0.7$, $t_{out} = 3$.

*Equation 8*.
These expressions are equivalent to one another (Appendix 8 and *Figure 5*, compare left and right sides).

$$sv = r_{in} - \rho_g t_{in} = \left( r_{in} - \frac{r_{out}}{t_{out}} t_{in} \right) \left( \frac{t_{out}}{t_{in} + t_{out}} \right)$$

For an interactive exploration of these effects of changing the outside and inside reward and time on subjective value, see https://github.com/HuShuLab/InteractivePlot, copy archived at *Sutlief, 2025*.

## Time's cost: opportunity and apportionment costs determine a pursuit's subjective value

By decomposing the global reward rate into 'inside' and 'outside' the considered pursuit type, the cost of time is revealed as being determined by (1) an opportunity cost, *and* (2) an apportionment cost (*Figure 5*). We show in *Figure 5* that the left and right parenthetical terms of *Equation 8* correspond to subtracting the *opportunity* cost of time ($r_{in} - r_{out}/t_{out} * t_{in}$) and then scaling by the *apportionment* of time spent 'outside' the considered pursuit, ($t_{out}/(t_{out} + t_{in})$), that is the weight of the outside. The *opportunity cost* associated with a considered pursuit, $\rho_{out} t_{in}$, is the reward rate outside the considered pursuit type, $\rho_{out}$, times the time of the considered pursuit, $t_{in}$ (*Figure 5C*). This opportunity cost is subtracted from the reward obtained from accepting the considered pursuit to yield the opportunity cost-subtracted reward (*Figure 5D*). *Figure 5E* shows that subjective value is to the opportunity cost-subtracted reward as the outside time is to the total time to traverse the world. Therefore, the right parenthetical term, $t_{out}/(t_{out} + t_{in})$, uses time's apportionment in and outside the pursuit to scale the opportunity cost-subtracted reward to the equivalent reward magnitude requiring no time investment, that is, the subjective value. This 'apportionment scaling' term thus relates the agent's time allocation in the world: the time spent outside a pursuit type relative to the time spent to traverse the world. In so downscaling, the subjective value of a considered pursuit (green) is to the time it would take to traverse the world were the pursuit *not taken*, $t_{out}$, as its opportunity cost subtracted reward (cyan) is to the time to traverse the world *were it to be taken* ($t_{in} + t_{out}$; *Figure 5E*).

What, then, is the amount of reward by which the opportunity cost-subtracted reward is scaled down to equal the *sv* of the pursuit? This amount is the *apportionment cost of time*. The apportionment cost of time (height of the brown vertical bar, *Figure 5F*) is the global reward rate *after taking into account the opportunity cost* (slope of the magenta-gold dashed line in *Figure 5F*), times the time of the considered pursuit. Equally, the difference between the inside and outside reward rates, times the

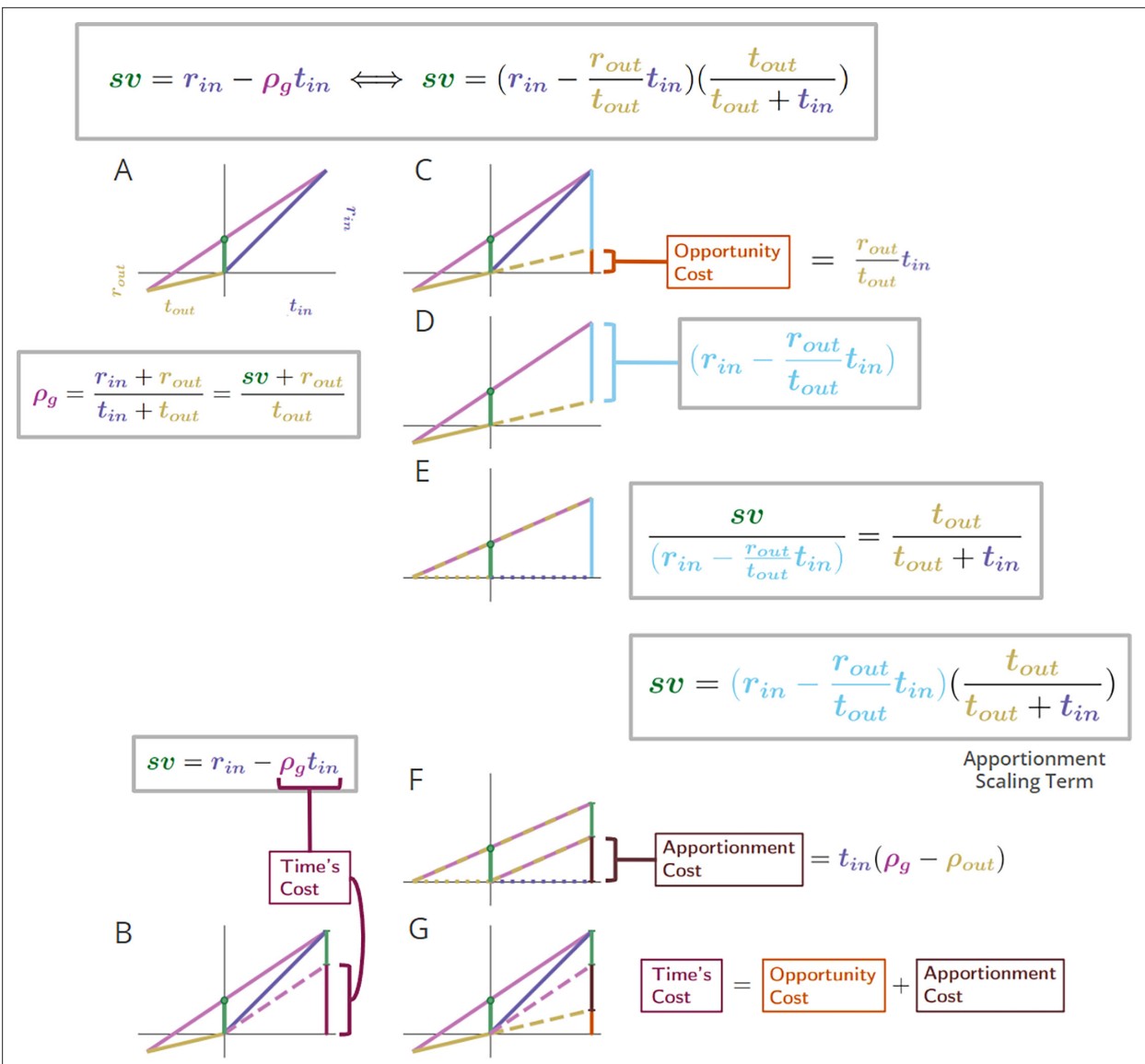

**Figure 5.** Equivalent expressions for subjective value reveal time's cost comprises an apportionment as well as opportunity cost. (**A**) The subjective value of a pursuit can be expressed in terms of the global reward rate obtained under a policy of accepting the pursuit. $r_{in}$ = 4, $t_{in}$ = 4, $r_{out}$ = .7, $t_{out}$ = 3. (**B**) The cost of time of a pursuit is the amount of reward earned on average in an environment over the time needed for its obtainment under a policy of accepting the pursuit. The reward rate earned on average is the global reward rate (slope of magenta line). Projecting that global reward over the time of the considered pursuit (dashed magenta line) provides the cost of time for the pursuit (vertical maroon bar). Therefore, the subjective value of a pursuit is equivalent to its reward magnitude less the cost of time of the pursuit (box equation above **B**). (**C**) Expressing subjective value with respect to the outside reward and time rather than the global reward rate reveals that a portion of a pursuit's time cost arises from an opportunity cost (orange bar). The opportunity cost of a pursuit is the amount of reward earned over the considered pursuit's time on average under a policy of *not* taking the considered pursuit (i.e. the outside reward rate, slope of gold line). Projecting the slope of the gold line over the time of the considered pursuit (dashed gold line) provides the opportunity cost of the pursuit (vertical orange bar). (**D**) The opportunity cost-subtracted reward (cyan bar). The triangle with sides cyan, magenta, and gold (solid and dashed) is congruent with the triangle with sides green, magenta, and gold (solid). Therefore, (**E**) $sv$ is to the opportunity cost-subtracted reward, as $t_{out}$ is to $t_{out} + t_{in}$. The opportunity cost-subtracted reward can thus be scaled by $t_{out}/(t_{out} + t_{in})$ to yield the subjective value of the pursuit. This scaling term, which we coin 'apportionment scaling', is the proportion that the outside time is to the total time to traverse the world. The slope of the dashed magenta and cyan line is the global reward rate *after accounting for the opportunity cost*. (**F**) The global reward rate after accounting for the opportunity cost (dashed magenta and gold), projected from the origin over the time of the pursuit is the apportionment cost of time (brown vertical line). It is the amount of the opportunity cost-subtracted reward that would occur on average over the pursuit's time under a policy of accepting the pursuit. (**G**) Time's cost is apportionment cost plus the opportunity cost. Whether expressed in terms of the global reward rate achieved under a policy of accepting the considered pursuit (A-B) or from the perspective of the outside time and reward (**C-G**), the subjective value expressions are equivalent.

time of the pursuit, is the apportionment cost when scaled by the pursuit's weight, that is the fraction that the considered pursuit is to the total time to traverse the world (Equation 9, right hand side). From the perspective of decision-making policies, *apportionment cost* is the difference in reward that can be expected, on average, between a policy of *taking* versus a policy of *not taking* the considered pursuit, over a time equal to its duration (Equation 9, *Apportionment Cost,* center, ***Figure 5F***).

$$Apportionment\ Cost = t_{in}\left(\rho_g - \rho_{out}\right) = t_{in}\left(\frac{r_{in}}{t_{in}} - \frac{r_{out}}{t_{out}}\right)\left(\frac{t_{in}}{t_{in} + t_{out}}\right)$$ (9)

*Equation 9.*

While this difference is the apportionment cost of time, the opportunity cost of time is the amount that would be expected from a policy of *not* taking the considered pursuit over a time equal to the considered pursuit's duration. Together, they sum to Time's Cost (***Figure 5G***). Expressing a pursuit's worth in terms of the global reward rate obtained under a policy of accepting the pursuit type (***Figure 5 left column***), or from the perspective of the outside reward and time (***Figure 5 right column***), are equivalent. However, the latter expresses *sv* in terms that are independent of one another, conveys the constituents giving rise to global reward rate, and provides the added insight that time's cost comprises an apportionment as well as an opportunity cost.

## The effect of increasing the outside reward on the subjective value of a pursuit

Let us now consider the impact that changing the outside reward and/or outside time has on these two determinants of time's cost—opportunity and apportionment cost—to further our understanding of the subjective value of a pursuit. ***Figure 6*** illustrates the impact of changing the reward reaped from outside the pursuit on the pursuit's subjective value. By holding the time spent outside the considered pursuit constant, changing the outside reward thus changes the outside reward rate. When the considered pursuit's reward rate is greater than its outside reward rate, the subjective value is positive (***Figure 6A***, green dot). The subjective value diminishes linearly (***Figure 6B***, green dots) to zero as the outside reward rate increases to match the pursuit's reward rate, and turns negative as the outside reward rate exceeds the pursuit's reward rate, indicating that a policy of accepting the considered pursuit would result in a lower attained global reward rate than that garnered under a policy of forgoing the pursuit. Under these conditions, the subjective value is shown to decrease linearly (***Figure 6C***, green dotted line) as the outside reward increases because the cost of time increases linearly (***Figure 6C***, maroon dotted region).

Time's cost is the sum of the opportunity cost (***Figure 6D***) and apportionment cost (***Figure 6E***) of time. ***Figure 6F*** overlays the opportunity cost and apportionment cost of time to demonstrate how their sum constitutes time's cost, and thus accounts for subjective value (green dotted line). When the outside reward is zero, there is zero opportunity cost of time (***Figure 6D and F*** orange dots), with time's cost being entirely constituted by the apportionment cost of time (***Figure 6E and F*** brown annuli). Apportionment cost (***Figure 6E and F***) decreases as outside reward increases because the difference between the inside and outside reward rate diminishes, thus making how time is apportioned in and outside the pursuit less relevant. At the same time, as outside reward increases, the opportunity cost of time increases (***Figure 6D***). When inside and outside rates are the same, how the agent apportions its time in or outside the pursuit does not impact the global rate of reward. At this point, the apportionment cost of time has fallen to zero, while the opportunity cost of the pursuit has now come to entirely constitute time's cost. Further increases in the outside reward now result in the outside rate being increasingly greater than the inside rate making the apportionment of time in/outside the pursuit increasingly relevant. Now, however, although the opportunity cost of time continues to grow positively, the apportionment cost of time grows increasingly *negative* (which is to say the pursuit has an apportionment *gain*). Summing the opportunity cost of the pursuit and the now *negative* apportionment cost (i.e. the apportionment gain) continues to yield time's cost; therefore, subtracting time's cost from the pursuit's reward continues to yield the subjective value of the pursuit (green dotted line).

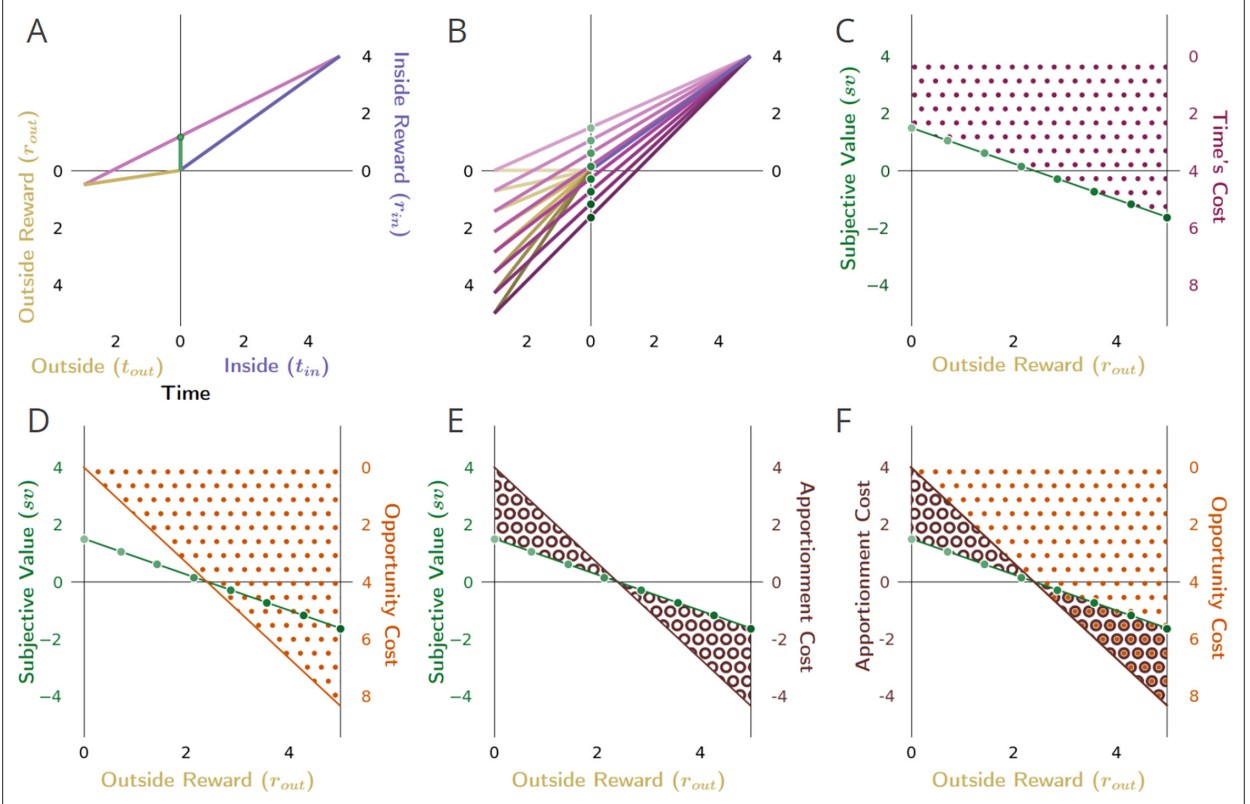

**Figure 6.** The impact of outside reward on the subjective value of a pursuit. (**A**) The subjective value (green dot) of a considered pursuit type (purple) in the context of its 'outside' (gold) is the resulting global reward rate vector's (magenta) intersection of the y-axis in the present (t=0). (**B**) A family of vectors showing that increasing the outside reward while holding the outside time constant increases the outside reward rate (slope of gold lines), resulting in increasing the global reward rate (slope of the purple lines), and decreasing the subjective value (green dots) of the pursuit. As the reward rate of the environment outside the considered pursuit type increases from lower than to higher than that of the considered pursuit, the subjective value of the pursuit decreases, becomes zero when the in/outside rates are equivalent, and goes negative when $\rho_{out}$ exceeds $\rho_{in}$. (**C**) Plotting the subjective value of the pursuit (green dots in **B**) as a function of increasing the outside reward (green dotted line in **C** through **F**) reveals that the subjective value of the pursuit decreases linearly. This linear decrease is due to the linear increase in the cost of time of the pursuit (maroon dotted region) as outside reward, and thus global reward rate, increases. Subtracting time's cost from the pursuit's reward yields the pursuit's subjective value as outside reward increases (green-dotted line, **C** through **F**). Time's cost (the area, as in **C**, between the pursuit's reward magnitude and its subjective value) is the sum of (**D**) the opportunity cost of time (orange dotted region), and (**E**) the apportionment cost of time (brown annuli region), as shown in (**F**). (**D**) As the outside reward increases, the opportunity cost of time increases linearly. (**E**) As the outside reward increases, the apportionment cost of time decreases linearly, becomes zero when the inside and outside reward rates are equal, and then becomes negative (becomes an apportionment gain). (**F**) When the outside reward rate is zero, time's cost is composed entirely of an apportionment cost. As the outside reward increases, opportunity cost increases linearly as apportionment cost decreases linearly, summing to time's cost. This continues until the reward rates in and outside the pursuit become equivalent, at which point the subjective value of the pursuit is zero. When subjective value is zero, the cost of time is entirely composed of opportunity cost. As the outside rate exceeds the inside rate, opportunity cost continues to increase while the apportionment cost becomes increasingly negative (which is to say, the apportionment cost of time becomes an apportionment gain of time). Adding the positive opportunity cost and the now negative apportionment cost (the region of overlap between brown annuli and orange dots) continues to sum to yield time's cost.

## The effect of changing the outside time on the subjective value of the considered pursuit

*Figure 7* examines the effect of changing the outside time on the subjective value of a pursuit, while holding the outside reward constant at a value of zero (*Figure 7A and B*). Doing so affords a means to examine the apportionment cost of time in isolation from the opportunity cost of time. Despite there being no opportunity cost, there *is* yet a cost of time (*Figure 7C*) composed entirely of the apportionment cost (*Figure 7D*). When the portion of time spent outside dominates, time's apportionment cost of the pursuit is small. As the portion of time spent outside the pursuit decreases and the relative apportionment of time spent in the pursuit increases, the apportionment cost of the pursuit increases purely hyperbolically, resulting in the subjective value of the pursuit decreasing purely hyperbolically

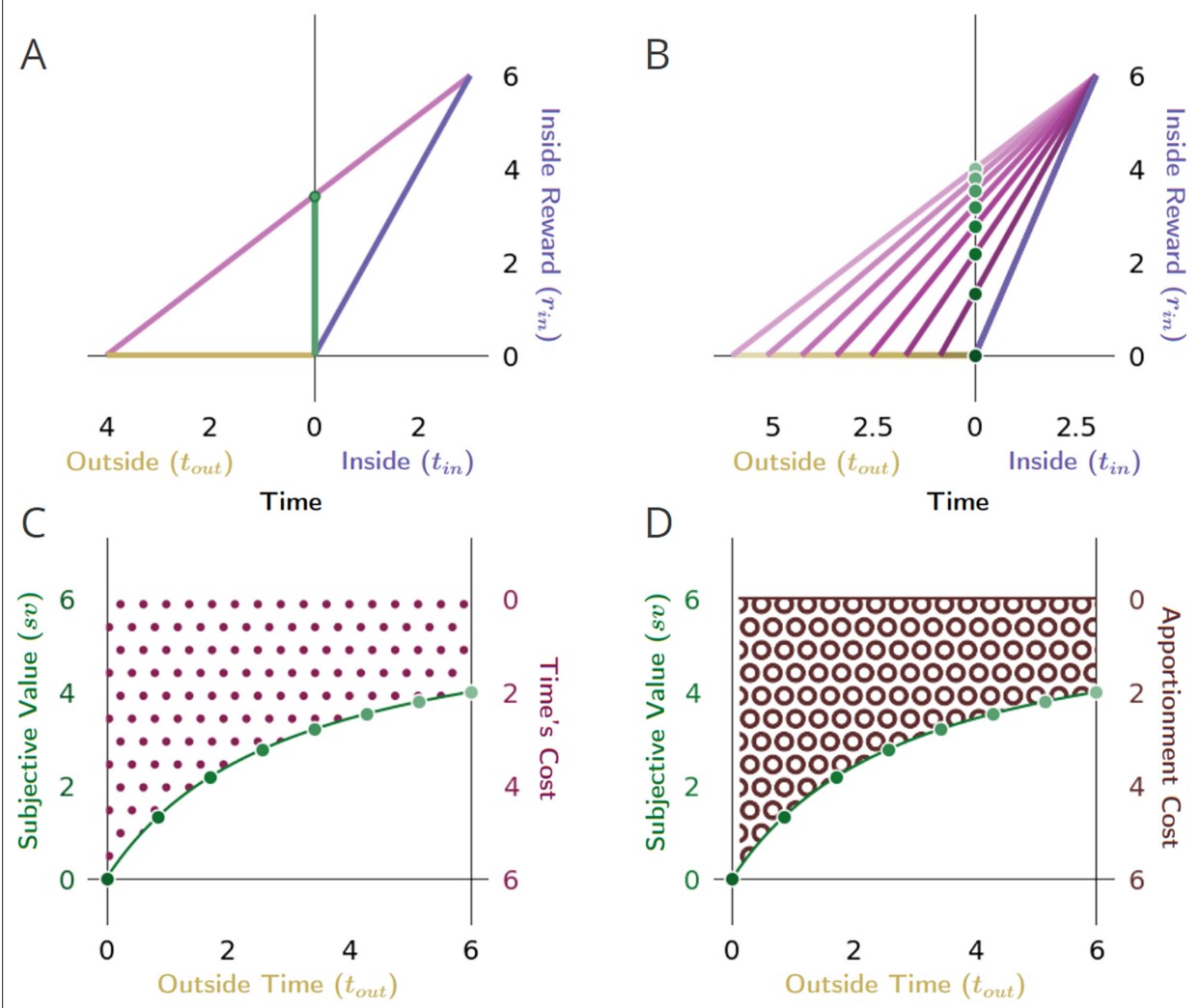

**Figure 7.** The impact of the apportionment cost of time on the subjective value of a pursuit. (**A**) The apportionment cost of time can best be illustrated dissociated from the contribution of the opportunity cost of time by considering the special instance in which the outside has no net reward, and therefore has a reward rate of zero. (**B**) In such instances, the considered pursuit type (purple vector) still has a cost of time, however, which is entirely composed of apportionment cost. Time's apportionment cost will decrease as the time in the outside (the family of outside time golden vectors) increases, decreasing the slope of corresponding global reward rate vectors (family of magenta vectors) and, thus, increasing the subjective value (green dots) of the considered pursuit. (**C**) Here, the cost of time is entirely composed of apportionment cost, which arises from the fact that the considered pursuit is contributing its proportion to the global reward rate. How much the pursuit's time cost is (maroon dots) is thus determined by the ratio of the time spent in the pursuit versus outside the pursuit; the more time is spent outside the pursuit, the less the apportionment cost of time of the pursuit, and therefore, the greater the subjective value of the pursuit (green dotted line). (**D**) When apportionment cost (brown annuli) solely composes the cost of time, the cost of time decreases hyperbolically as the outside time increases, resulting in the subjective value of a pursuit increasing hyperbolically (green dotted line). Conventions as in *Figure 6*.

(*Figure 7D*). As time spent outside the considered pursuit becomes diminishingly small, the pursuit comprises more and more of the world, until the apportionment of time is entirely devoted to the pursuit, at which point the apportionment cost of time equals the pursuit's reward rate ($\rho_{in}$) times the pursuit's time ($t_{in}$) (i.e. the pursuit's reward magnitude, $r_{in}$).

### The effect of changing the outside time and outside reward rate on the subjective value of a pursuit

In having examined the effect of varying outside reward and thus outside reward rate (*Figure 6*), and outside time while holding outside reward rate constant (*Figure 7*), let us now consider the impact of varying the outside time and the outside reward rate (*Figure 8*). By changing the outside time

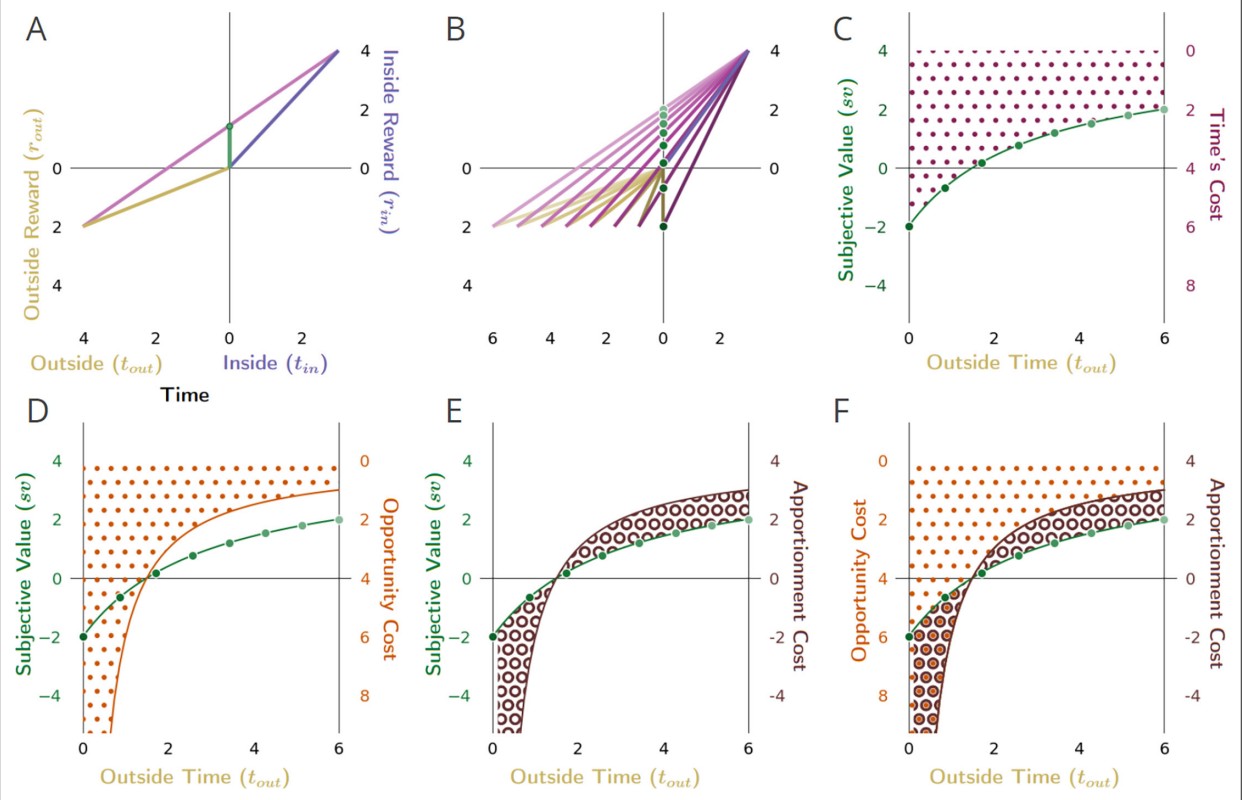

**Figure 8.** The effect of changing the outside time and thus the outside reward rate, on the subjective value of a pursuit. (**A**) The subjective value (green dot) of the considered pursuit, when (**B**) changing the outside time, and thus, outside reward rate (green dots). (**C**) As outside time increases under these conditions (holding outside reward constant), the subjective value of the pursuit increases hyperbolically (green dotted line), from the negative of the outside reward magnitude to, in the limit, the inside reward magnitude. Conversely, time's cost (right hand axis, maroon dots) decreases hyperbolically. (**D**) Opportunity cost (orange stippling, right hand y-axis) decreases hyperbolically as outside time increases. (**E**) Apportionment cost (right hand y-axis) is the difference of two hyperbolas (brown annuli area). It is initially negative, meaning that apportionment cost is a *gain*, as the outside reward rate is greater than the inside reward rate. This apportionment gain decreases to zero as outside time increases to make the outside and inside rates equal, and then becomes an apportionment cost (positive). (**F**) Summing opportunity cost and apportionment cost yields time's cost; subtracting time's cost from the pursuit's reward magnitude yields the subjective value (green dotted curve). Conventions as in *Figure 6*.

while holding the outside reward constant (*Figure 8B*), the reward rate obtained in the outside will be varied while the apportionment of time in and outside the pursuit changes, thus impacting the opportunity and apportionment cost of time. Plotting the subjective value-by-outside time function (green dotted line) then reveals that subjective value increases hyperbolically under these conditions as outside time increases, which is to say, time's cost decreases hyperbolically (*Figure 8C*). Decomposing time's cost into its constituent opportunity (*Figure 8C*) and apportionment costs (*Figure 8D*) illustrates how these components vary when varying outside time. Opportunity cost (orange dots, *Figure 8C*) decreases hyperbolically as the outside time increases. Apportionment cost varies as the difference of two hyperbolas (brown annuli area, *Figure 8D*), its sign being initially negative (an apportionment gain), decreases to zero as the outside and inside rates become equal, and then becomes positive (an apportionment cost). Taken together (*Figure 8F*), their sum (opportunity and apportionment costs) decreases hyperbolically as outside time increases, resulting in subjective values that hyperbolically increase, spanning from the negative of the outside reward magnitude to the inside reward magnitude.

## The value of initiating pursuits in Choice decision-making

Above, we determined how a reward-rate-maximizing agent would evaluate the worth of a pursuit, thereby identifying the impact of a policy of taking (or forgoing) that pursuit on the realized global reward rate and expressing that pursuit's worth as subjective value. We did so by opposing a pursuit with its equivalent offer requiring no time investment—a special and instructive case. In this section,

we consider what decision should be made when an agent is simultaneously presented with a choice of more than one pursuit of any potential magnitude and time investment. Using the subjective value under these Choice decisions, we more thoroughly examine how the duration and magnitude of a pursuit, and the context in which it is embedded (its 'outside'), impacts reward-rate-optimal valuation. We then re-express subjective value as a temporal discounting function, revealing the nature of the *apparent* temporal discounting function of a reward-rate-maximizing agent as one determined wholly by the temporal structure and magnitude of rewards in the environment. We then assess whether hyperbolic discounting, the 'Delay' effect, and the 'Magnitude' and 'Sign' effect—purported signs of suboptimal decision-making (*McDiarmid and Rilling, 1965*; *Rachlin et al., 1972*; *Ainslie, 1974*; *Snyderman, 1983*; *Green et al., 1994*; *Kirby and Herrnstein, 1995*; *Myerson and Green, 1995*; *Bateson and Kacelnik, 1996*; *Ostaszewski, 1996*; *Stephens and Anderson, 2001*; *Cheng et al., 2002*; *Frederick et al., 2002*; *Hayden and Platt, 2007b*; *Hayden et al., 2007a*; *McClure et al., 2007*; *Beran and Evans, 2009*; *Peters and Büchel, 2011*; *Stevens and Mühlhoff, 2012*; *Carter et al., 2015*; *Carter and Redish, 2016*)—are in fact consistent with optimal decision-making.

## Choice decision-making

Consider a temporal decision in which two or more mutually exclusive options are simultaneously presented following a period that is common to policies of choosing one or another of the considered options (*Figure 9*). In such scenarios, subjects choose between outcomes differing in magnitude and the time at which they will be delivered. Of particular interest are choices between a smaller, sooner reward pursuit ('SS' pursuit) and a larger, later reward pursuit ('LL' pursuit) (*Myerson and Green, 1995*; *Frederick et al., 2002*; *Madden and Bickel, 2010*; *Peters and Büchel, 2011*). Such intertemporal decision-making is commonplace in the laboratory setting (*McDiarmid and Rilling, 1965*; *Rachlin et al., 1972*; *Ainslie, 1974*; *Snyderman, 1983*; *Myerson and Green, 1995*; *Bateson and Kacelnik, 1996*; *Ostaszewski, 1996*; *Stephens and Anderson, 2001*; *Cheng et al., 2002*; *Frederick et al., 2002*; *Hayden et al., 2007a*; *Hayden and Platt, 2007b*; *McClure et al., 2007*; *Beran and Evans, 2009*; *Peters and Büchel, 2011*; *Stevens and Mühlhoff, 2012*; *Carter et al., 2015*; *Carter and Redish, 2016*).

### Global reward rate equation and optimal choice policy

With the global reward rate equation previously derived, which choice policy (i.e. choosing SS, or LL) would maximize global reward rate can be identified. The optimal choice between the SS and the LL pursuit is as follows:

- $\frac{r_{LL} - r_{SS}}{t_{LL} - t_{SS}} < \rho_g^*$, choose SS pursuit (Appendix 9).
- $\frac{r_{LL} - r_{SS}}{t_{LL} - t_{SS}} = \rho_g^*$, both SS and LL pursuits are equivalent (Appendix 9).
- $\frac{r_{LL} - r_{SS}}{t_{LL} - t_{SS}} > \rho_g^*$, choose LL pursuit (Appendix 9).

These policies' optimality is intuitive. By choosing option LL, the subject earns $r_{LL} - r_{SS}$ more reward than when choosing SS but spends $t_{LL} - t_{SS}$ more time. If the reward rate from that extra time spent exceeds the reward rate of the environment generally, it would be optimal to spend the extra time on the LL option. In other words, if the agent were to choose pursuit SS, $t_{LL} - t_{SS}$ time would be spent earning reward at a global reward rate under that policy, $\rho_{g,choose\,SS}$, with the magnitude $\rho_g (t_{LL} - t_{SS})$. If $\rho_g (t_{LL} - t_{SS})$ exceeds the extra reward $r_{LL} - r_{SS}$ that could be earned with that extra time by investing the LL pursuit, more reward would be earned in the same amount of time by choosing the SS Pursuit.

### Optimal choice policies based on subjective value

As under Forgo decision-making, we can now also identify the global reward rate optimizing choice policies based on subjective value (*Figure 9*). The following policies would optimize reward rate when choosing between two options of different magnitude that require different amounts of time invested:

- $sv_{LL} < sv_{SS}$, take pursuit SS (Appendix 10).
- $sv_{LL} = sv_{SS}$, SS and LL pursuits are equivalent (Appendix 10).
- $sv_{LL} > sv_{SS}$, take pursuit LL (Appendix 10).

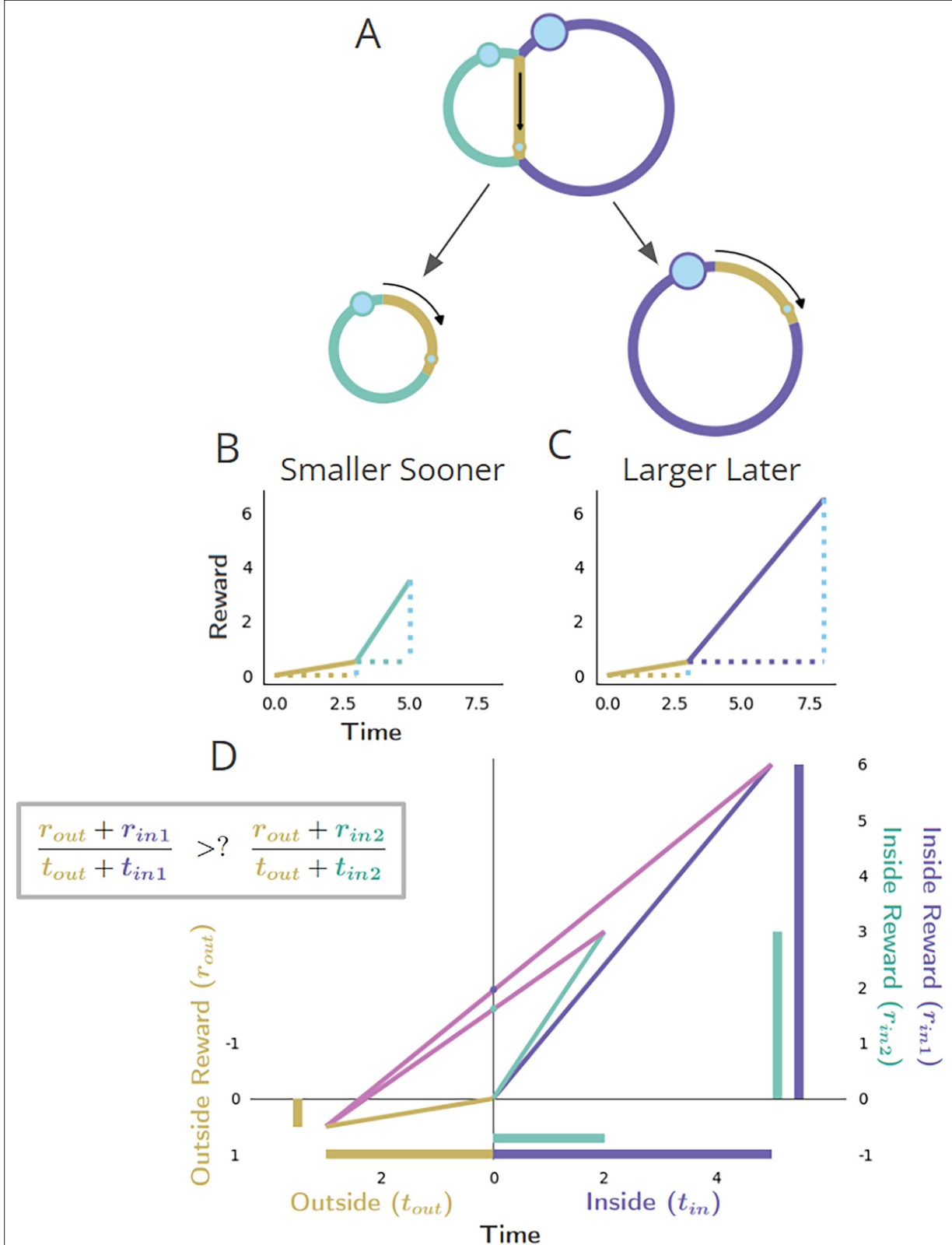

**Figure 9.** Policy options considered during the initiation of pursuits in worlds with a 'Choice' topology. (**A-C**) Choice topology, and policies of choosing the smaller-sooner or larger-later pursuit, as in *Figure 1* 'Choice'. (**D**) The world divided into 'Inside' and 'Outside' the selected pursuit type, as the agent decides whether to accept the SS (aqua) or the LL (purple) pursuit. The global reward rate ( $\rho_g$ ) under a policy of choosing SS or LL (slopes of the magenta lines) is calculated from the equation in the box to the right.

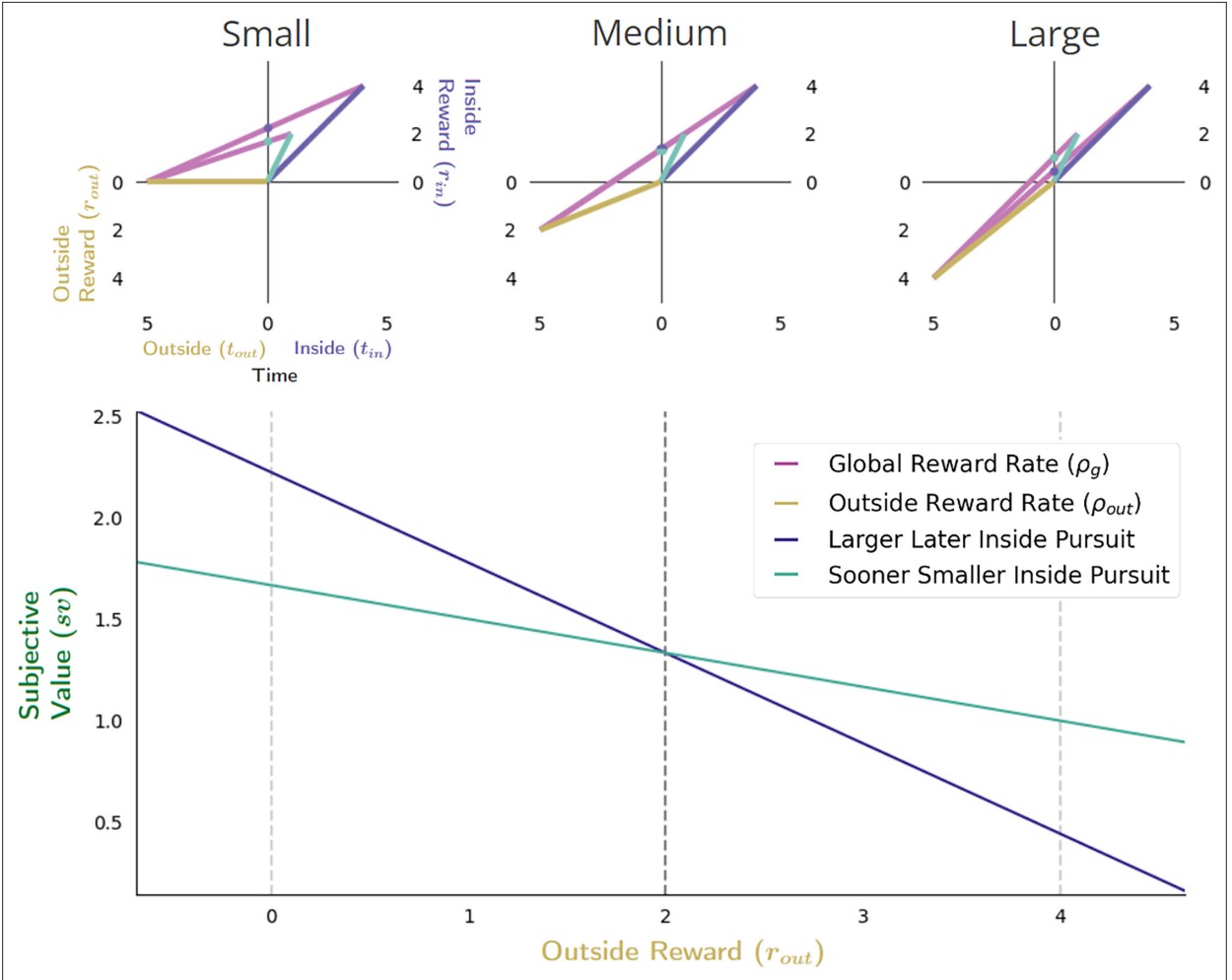

**Figure 10.** The effect of increasing outside reward on subjective value in choice decision-making. The effect of increasing the outside reward while holding the outside time constant is to linearly increase the cost of time, thus decreasing the subjective value of pursuits considered in choice decision-making. When the outside reward is sufficiently small (top left), the subjective value of the LL pursuit can exceed the SS pursuit, indicating that selection of the LL pursuit would maximize the global reward rate. As outside reward increases, however, the subjective value of pursuits will decrease linearly as the opportunity cost of time increases. Since a policy of choosing the LL pursuit will have the greater opportunity cost, the slope of its function relating subjective value to outside reward will be greater than that of a policy of choosing the SS pursuit (bottom). Thus, outside reward can be increased sufficiently such that the subjective value of the LL and SS pursuits will become equal (top middle), past which the agent will switch to choosing the SS pursuit (top right). Vertical dashed lines (bottom) correspond to instances along top.

## The impact of opportunity and apportionment costs on choice decision-making

With optimal policies for choice expressed in terms of subjective value, the impact of time's opportunity and apportionment costs on Choice decision-making can now be more deeply appreciated. Keeping the outside time constant, the opportunity cost of time increases as the outside reward (and thus the outside reward rate) increases. As increasing the outside reward rate will also impact the apportionment cost of time linearly, (see *Figure 6*), adding these costs results in a net linear decrease in the subjective value of the considered pursuits (*Figure 10*), with the decrease being driven steadily more by the opportunity cost. When the outside reward rate is relatively small, reward-rate maximization may dictate that the agent choose the LL pursuit (as in *Figure 10*, upper left). However, as the opportunity cost of the LL pursuit is greater than that of the SS due to its greater reward and time requirement, the subjective value-by-outside time slope will be more negative for the LL than the SS pursuit, resulting in their subjective values crossing at a critical outside reward threshold (see *Figure 10*, upper middle). This critical threshold occurs when the global reward rates under a policy of

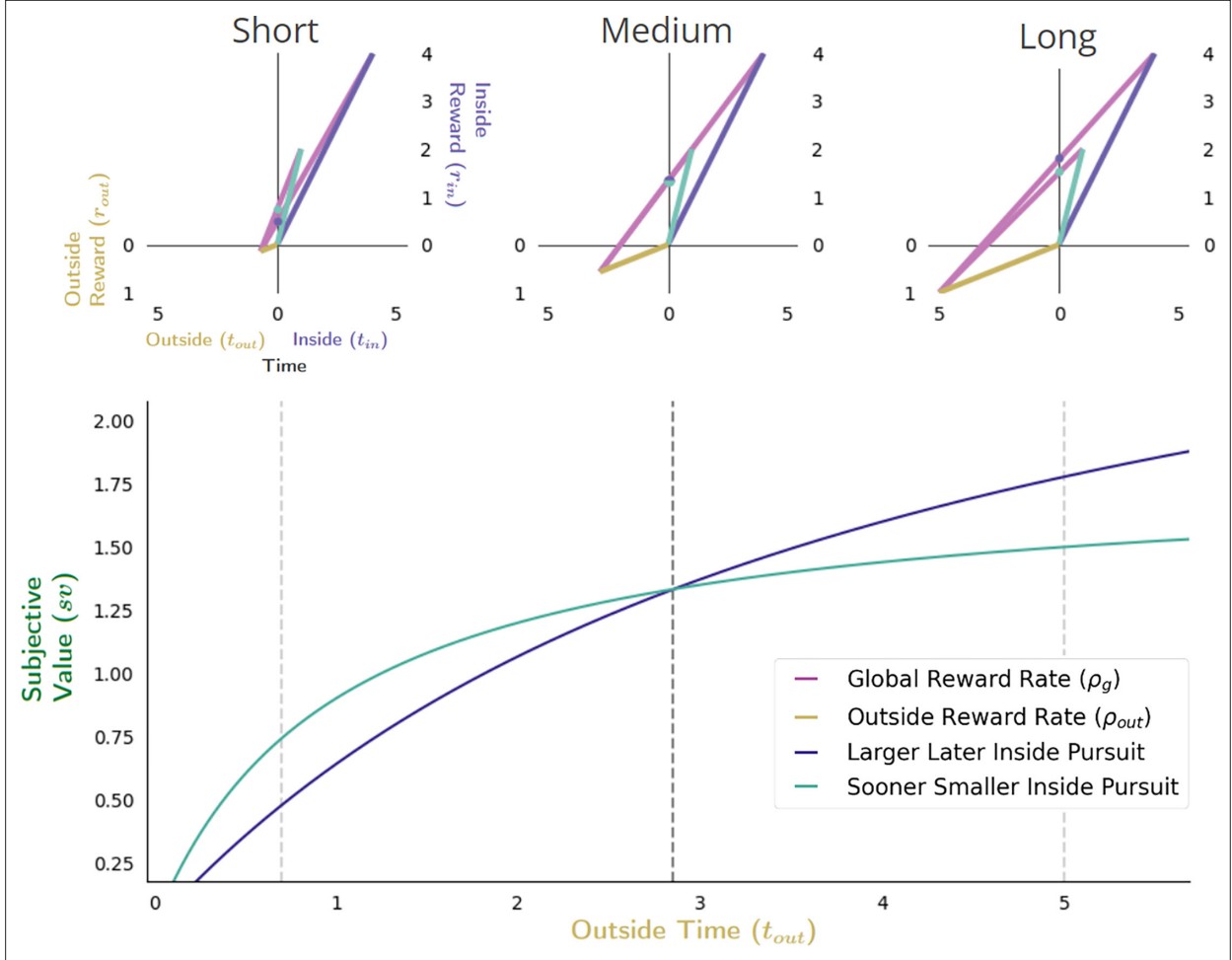

**Figure 11.** Effect of apportionment cost on subjective value in Choice decision-making. The effect of apportionment cost can be isolated from the effect of opportunity cost by increasing the outside time while holding outside rate constant. Doing so results in decreasing the apportionment cost of the considered pursuit, thus increasing its subjective value (bottom). When the outside time is sufficiently small (top left), the apportionment cost for LL and SS pursuits will be relatively large, but can be greater still for the LL pursuit given its proportionally longer duration to the outside time and greater reward magnitude. As outside reward time increases, however, the subjective value of pursuits increase as the apportionment cost of time of the considered pursuit decreases. As apportionment costs diminish and the magnitudes of pursuits' rewards become more fully realized, the subjective value of the LL and SS pursuits will become equal (top middle), with the LL pursuit eventually exceeding that of the SS pursuit at sufficiently long outside times (top right). Vertical dashed lines (bottom) correspond to instances along top.

accepting the LL or of accepting the SS pursuits are equal. A switch in preference from the LL pursuit to that of the SS pursuit will then be observed as outside reward values become larger, past this critical threshold (see *Figure 10*, upper right).

A switch in preference between the SS and LL pursuits will also occur when the time spent outside the considered pursuit increases, even if the outside reward rate earned remains constant (*Figure 11*). In this case, the opportunity cost of time will remain unchanged, with only the apportionment cost of time driving changes in the subjective value (the aqua (SS) and purple (LL) dots on the y-axis, *Figure 11*) of pursuits. As any inside time will constitute a greater fraction of the total time under a LL versus a SS pursuit policy, the apportionment cost of the LL pursuit will be greater. This can result in the subjective value of the SS pursuit being greater initially (when outside time is short, *Figure 11* upper left), than the LL pursuit. As the outside time increases, however, the ordering of subjective value will switch as apportionment costs becoming diminishingly small.

Finally, the effect of varying the outside time and the outside reward rate on subjective value in Choice behavior is considered (*Figure 12*). Opportunity and apportionment costs can simultaneously be varied, for instance, by maintaining outside reward but increasing outside time. Recall

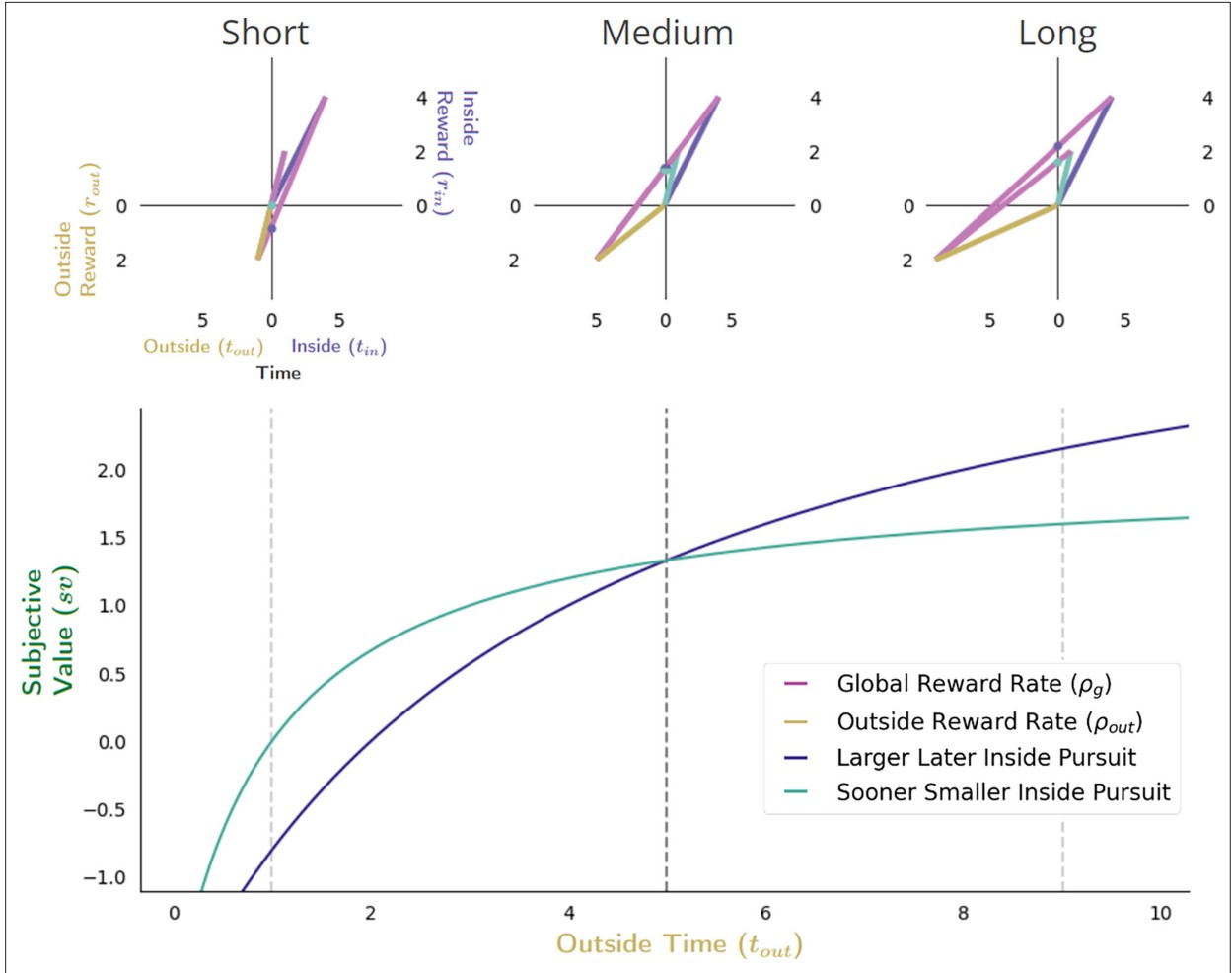

**Figure 12.** Effect of varying outside time and outside reward rate. The effect of increasing the outside time while maintaining outside reward is to decrease the apportionment as well as the opportunity cost of time, thus increasing pursuit's subjective value. Increasing outside time, which in turn, also decreases outside reward rate, results in the agent appearing as if to become more patient, being willing to switch from a policy of selecting the SS pursuit (upper left) to a policy of treating them equally (upper middle) at some critical threshold (vertical dashed black line, bottom), to a policy of selecting the LL pursuit (upper right). Vertical dashed lines (bottom) correspond to instances along top.

that doing so decreases the opportunity cost of time hyperbolically while the apportionment cost of time varies as the difference of two hyperbolas; their sum (time's cost) decreases hyperbolically (see *Figure 8*). Therefore, the subjective value-by-outside time functions of the LL and SS pursuits *increase* hyperbolically (*Figure 12*, bottom), but since they span different ranges (from the negative of the outside reward magnitude to their respective inside reward magnitudes) these functions will cross, thus switching from the SS to the LL pursuit (*Figure 12*) at a critical outside time.

A reward-rate-optimal agent will thus appear as if more patient (1) the longer the time spent outside a considered pursuit, (2) the lower the outside reward rate, or (3) both, switching from a policy of choosing the SS to choosing the LL option at some critical outside reward rate and/or time. Having analyzed the impact of time spent and reward obtained outside a pursuit on a pursuit's valuation, we now examine the impact time spent within a pursuit has on its valuation.

## The discounting function of a reward-rate-optimal agent

How does the value of a pursuit change as the time required for its obtainment grows? Intertemporal decision-making between pursuits requiring differing time investments resulting in different reward magnitudes has typically been examined using a 'temporal discounting function' to describe how delays in reward influence their valuation. This question has been investigated experimentally by pitting SS against LL options to experimentally determine the *subjective value* of the delayed reward

(**Mischel et al., 1969**), with the best fit to many such observations across delays determining the subjective value-time function. After normalizing by the magnitude of reward, the curve of subjective values as a function of delay is the 'temporal discounting function' (**Green et al., 1994**; **Kirby and Herrnstein, 1995**), and for review see **Frederick et al., 2002**). While the temporal discounting function has historically been used in many fields, including economics, psychology, ethology, and neuroscience to describe how delays influence rewards' subjective value, its origins—from a normative perspective—remain unclear. What, then, is the temporal discounting function of a reward-rate-optimal agent? And would its determination provide insight into why experimentally derived discounting functions present in the way they do, with their varied forms and curious sensitivity to the context, delay, magnitude, and sign of pursuit outcomes?

## The discounting function of an optimal agent is a hyperbolic function

To examine the temporal discounting function of a reward-rate-optimal agent, we begin with the subjective value-time function introduced in Equation 8, rearranging the apportionment scaling term to resemble the standard temporal discounting function form (see **Equation 10** RHS below, Subjective value-time function rearranged to resemble the standard temporal discounting function form).

$$sv\left(r,t\right) = \left(r_{in} - \rho_{out}t_{in}\right)\left(\frac{t_{out}}{t_{out} + t_{in}}\right) = \frac{r_{in} - \rho_{out}t_{in}}{1 + \frac{1}{t_{out}}t_{in}} \tag{10}$$

*Equation 10*.

The temporal discounting function of an optimal agent can then be expressed by normalizing this subjective value-time function by the considered pursuit's magnitude (**Equation 11**).

$$Discounting\ Function = \frac{sv\left(r,t\right)}{r} = \frac{r_{in} - \rho_{out}t_{in}}{1 + \frac{1}{t_{out}}t_{in}} * \left(\frac{1}{r_{in}}\right) = \frac{1 - \rho_{out}*\frac{t_{in}}{r_{in}}}{1 + \frac{1}{t_{out}}t_{in}} \tag{11}$$

*Equation 11*.

To illustrate the discounting function of a reward-rate maximal agent, **Figure 13** depicts how the worth of a pursuit's reward would change as its context—the 'outside' world in which it is embedded—changes. We do so by examining the apparent discounting function in three different world contexts: a world in which there is, (A) zero outside reward rate and large outside time, (B) zero outside reward rate and small outside time, and, (C) positive outside reward rate and small outside time. **Figure 13** first graphically depicts the subjective values of the pursuit's reward at increasing temporal delays (the y-intercepts of the lines depicting the resulting global reward rates, green dots) in each of these world contexts (A-C). Then, by replotting these subjective values at their corresponding delays, the subjective value-time function is created for this increasingly delayed reward in each of these worlds (D-F). By normalizing by the reward magnitude, these subjective value-time functions are then converted to their corresponding discounting functions (color coded) and overlaid so that their shapes may be compared (G).

Doing so illustrates how the mathematical form of the temporal discount function—as it appears for the optimal agent—is a hyperbolic function. This function's form depends wholly on the temporal reward structure of the environment and is composed of hyperbolic and linear components which relate to the apportionment and to the opportunity cost of time. To best appreciate the contributions of opportunity and apportionment costs to the discounting function of a reward rate-optimal agent, consider the following instances exemplified in **Figure 13**. First, in worlds in which no net reward is received outside a considered pursuit, the apparent discounting function is *purely* hyperbolic (**Figure 13A**). Purely hyperbolic discounting is therefore optimal when the subjective value function follows the equation

$$sv = \frac{r_{in}}{1 + \frac{t_{in}}{t_{out}}}$$

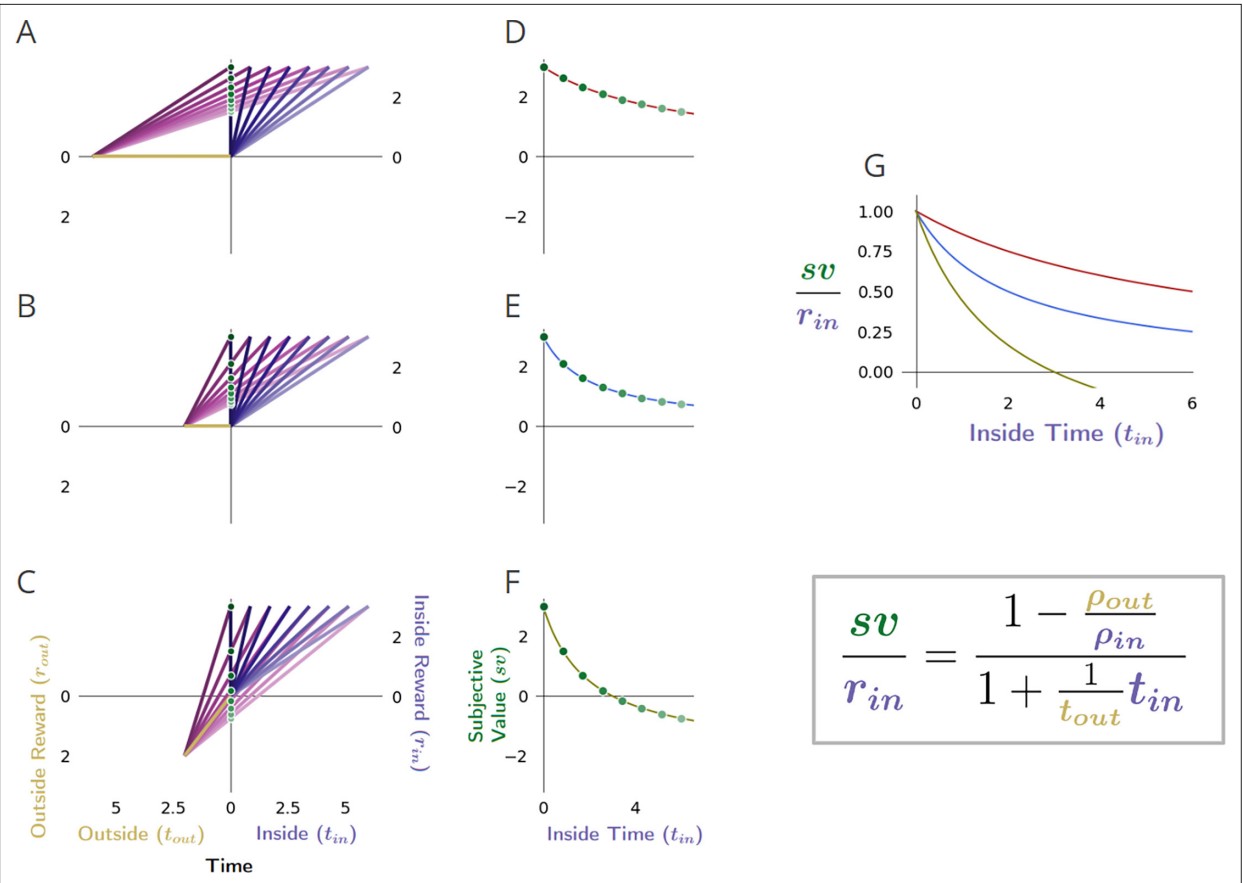

**Figure 13.** The temporal discounting function of a global reward-rate-optimal agent is a hyperbolic function relating the apportionment and opportunity cost of time. (**A-C**) The effect, as exemplified in three different worlds, of varying the outside time and reward on the subjective value of a pursuit as its reward is displaced into the future. The subjective value, sv, of this pursuit as its temporal displacement into the future increases is indicated as the green dots along the y-intercept in these three different contexts: a world in which there is (**A**) zero outside reward rate and large outside time, (**B**) zero outside reward rate and small outside time, and (**C**) positive outside reward rate and the small outside time as in (**B**). (**D-F**) Replotting these subjective values at their corresponding temporal displacement yields the subjective value-time function of the offered reward in each of these contexts. (**G**) Normalizing these subjective value functions by the reward magnitude and superimposing the resulting temporal discounting functions reveals how the steepness and curvature of the apparent discounting function of a reward-rate-maximizing agent changes with respect to the average reward and time spent outside the considered pursuit. When the time spent outside is increased (compare B to A)—thus decreasing the apportionment and opportunity cost of time—the temporal discounting function becomes less curved, making the agent appear as if more patient. When the outside reward is increased (compare B to C)—thus increasing the opportunity and apportionment cost of time—the temporal discounting function becomes steeper, making the agent appear as if less patient.

Therefore, a purely hyperbolic discounting function would arise from a reward-rate maximizing agent in a world common to many experimental designs where, for instance, $t_{out}$ may represent the intertrial interval with no reward. Second, as less time is apportioned outside the considered pursuit type (*Figure 13B*), this hyperbolic curve becomes more curved as the pursuit's time apportionment cost increases. The curvature of the hyperbolic component is thus controlled by how much time the agent spends 'in' versus 'outside' the considered pursuit: with the more time spent outside the pursuit, the gentler the curvature of apparent hyperbolic discounting and the more patient the agent appears to become for the considered pursuit. Third, in worlds in which reward is received outside a considered pursuit (compare *B to C*), the apparent discounting function will become steeper the more outside reward is obtained, as the linear component relating the opportunity cost of time increases.

Thus, by expressing the worth of a pursuit as would be evaluated by a reward-rate-optimal agent in terms of its discounting function, we find that its form is consonant with what is commonly reported experimentally in humans and animals and will exhibit apparent changes in curvature and steepness that relate directly to the reward acquired and time spent outside the considered pursuit for every time spent within it.

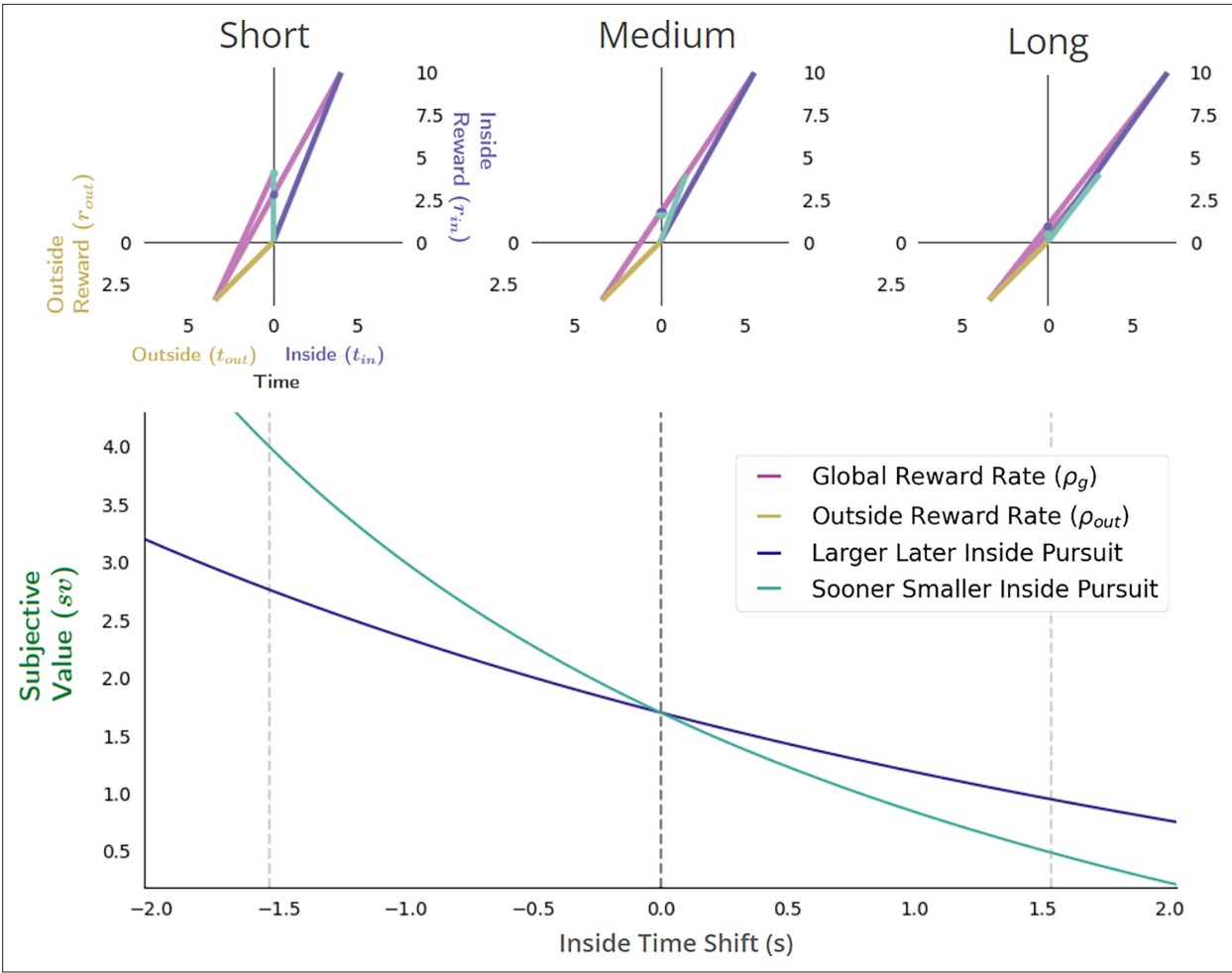

**Figure 14.** Reward-rate-maximizing agents would exhibit the 'Delay Effect'. A 'switch' in preference from a SS—when the delay to the pursuits is relatively short (upper left)—to a LL pursuit, when the delay to the pursuits is relatively long (upper right), would occur as a consequence of global reward-rate maximization. Bottom: The subjective value-time functions varying the delay to the SS (cyan) and LL (purple) pursuit with respect to the delay in which selecting either pursuit results in the same global reward-rate. Vertical dashed lines (Bottom) correspond to instances along top.

### The Delay Effect, the Magnitude Effect, and the Sign Effect

With this insight into how opportunity and apportionment costs impact the cost of time, and therefore the subjective value of pursuits in Choice decision-making, reward-rate-optimal agents are now understood to exhibit a hyperbolic form of discounting, as commonly exhibited by humans and animals (*Rachlin et al., 1972*; *Rachlin et al., 2000*; *Ainslie, 1975*; *Thaler, 1981a*; *Mazur, 1987*; *Benzion et al., 1989*; *Green et al., 1994*; *Kobayashi and Schultz, 2008*; *Calvert et al., 2010*; *Fedus et al., 2019*). As hyperbolic discounting is not a sign of suboptimal decision-making, as is widely asserted, are other purported signs of suboptimal decision-making, namely the 'Delay Effect', the 'Magnitude Effect' and 'Sign Effect' also consistent with optimal temporal decisions?

### The Delay Effect

The Delay Effect (also known as preference reversal) refers to the experimental observation made in Choice decisions where preference switches from the SS to the LL option as the delay to the options increase (*Green et al., 1994*; *Kirby and Herrnstein, 1995*). The switch in preference, despite the magnitudes and the time between the options remaining unchanged, is taken as a sign of anomalous decision making (*Cruz Rambaud et al., 2023*). *Figure 14* illustrates why preference reversal under increasing delay is not a sign of suboptimal decision-making but rather is consistent with reward-rate maximization. When the delay to the options is sufficiently short, reward-rate maximization can dictate selecting the SS pursuit (*Figure 14 top left*) in a choice between SS and LL. As the delay increases, the

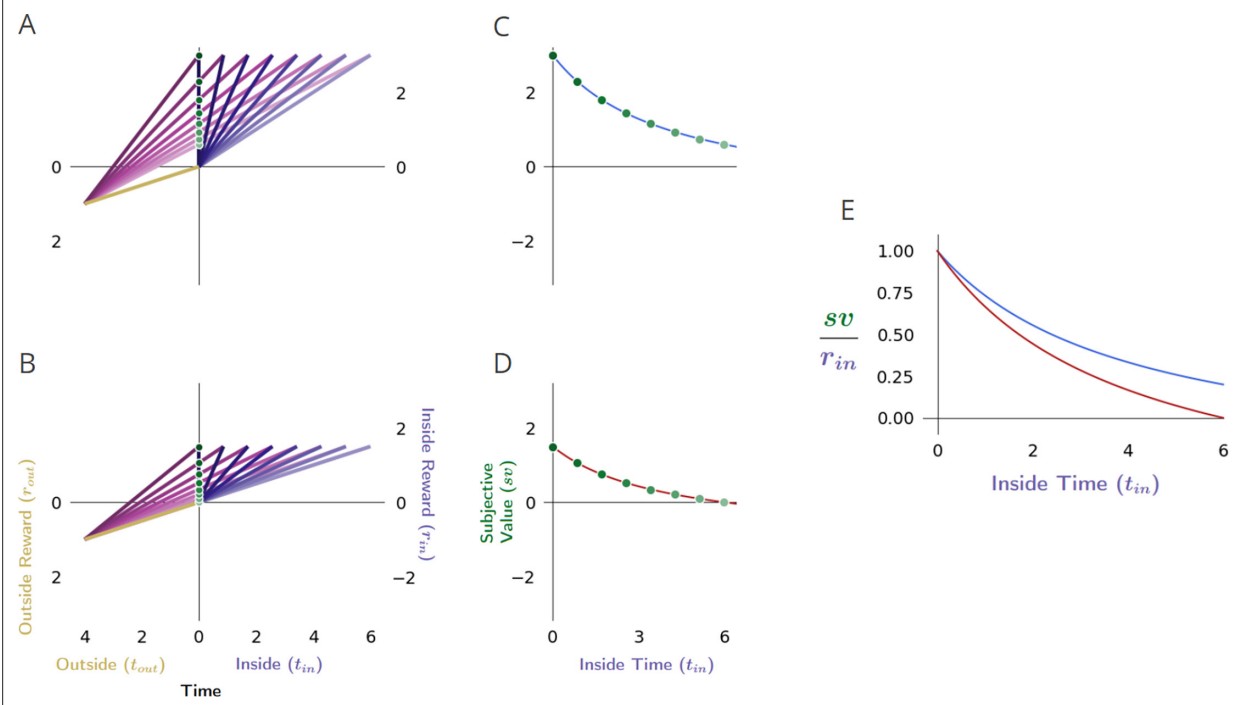

**Figure 15.** Reward-rate-maximizing agents would exhibit the 'Magnitude effect'. (**A, B**) The global reward rate (the slope of magenta vectors) that would be obtained when acquiring a considered pursuit's reward of a given size (either relatively large as in A or small as in B) but at varying temporal removes, depicts how a considered pursuit's subjective value (green dots, y-intercept) would decrease as the time needed for its obtainment increases in environments that are otherwise the same. (**C, D**) Replotting the subjective values of the considered pursuit to correspond to their required delay forms the subjective value-time function for the 'large' reward case (**C**), and the 'small' reward case (**D**). (**E**) Normalizing the subjective value-time functions by their reward magnitude transforms these functions into their corresponding discounting functions (blue: large reward DF; red: small reward DF), and reveals that a reward-rate-maximizing agent would exhibit the 'Magnitude Effect' as the steepness of the apparent discounting function would change with the size of the pursuit, and manifest as being less steep the greater the magnitude of reward.

global reward-rate maximizing policy under choose LL and choose SS policies would drop until they become equivalent, at which time their subjective values would be the same (*Figure 14* top middle, and dashed vertical black line *Figure 14* bottom). At delays greater than this critical value, the agent would switch to choosing the LL pursuit in Choice tasks (*Figure 14 top right*). Additionally, we note that further delay can result in the subjective value of the SS pursuit turning negative. At this point, the agent would choose the LL pursuit in a Choice context and would forgo the SS were it to be offered in a Forgo context. At even greater delay, the LL pursuit's subjective value would turn negative. This would result in the agent choosing LL under *forced* choice conditions but forgoing either pursuit in a Forgo context.

## The Magnitude Effect

The Magnitude Effect refers to the observation that the temporal discounting function, as experimentally determined, is observed to become less steep the larger the offered reward. If brains apply a discounting function to account for the delay to reward, why, as it is posed, do different magnitudes of reward appear as if discounted with different temporal discounting functions? *Figure 15* considers how a reward-rate-maximizing agent would appear to discount rewards of two magnitudes (large - top row; small - bottom row), first by determining the subjective value (green dots) of differently sized rewards (*Figure 15A and B*) across a range of delays, and second by replotting the *sv*'s at their corresponding delays (*Figure 15C and D*), to form their subjective value functions (blue and red curves, respectively). After normalizing these subjective value functions by their corresponding reward magnitudes, the resulting temporal discounting functions that would be fit for a reward-rate-maximizing agent are then shown in (*Figure 15E*). The pursuit with the larger reward outcome (blue) thus would appear as if discounted by a less steep discounting function than the

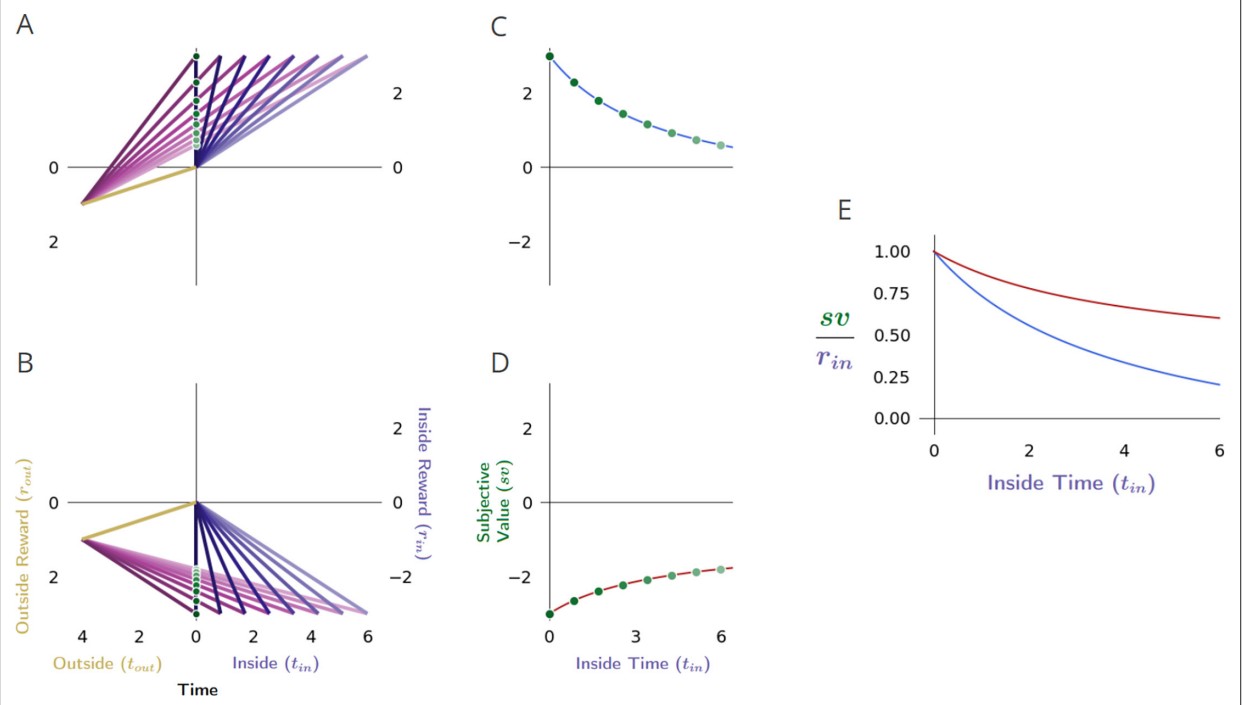

**Figure 16.** Reward-rate-maximizing agents would exhibit the 'Sign effect'. (**A, B**) The global reward rate (the slope of magenta lines) that would be obtained when acquiring a considered pursuit's outcome of a given magnitude but differing in sign (either rewarding as in A, or punishing as in B), depicts how the subjective value (green dots, y-intercept) would decrease as the time of its obtainment increases in environments that are otherwise the same (one in which the agent spends the same amount of time and receives the same amount of reward outside the considered pursuit for every instance within it). (**C, D**) Replotting the subjective values of the considered pursuit to correspond to their required delay forms the subjective value-time function for the reward (**C**) and for the punishment (**D**). (**E**) Normalizing the subjective value-time functions by their outcome transforms these functions into their corresponding discounting functions (blue: reward DF; red: punishment DF). This reveals that a reward-rate-maximizing agent would exhibit the 'Sign Effect', as the steepness of the apparent discounting function would change with the sign of the pursuit, manifesting as being less steep for punishing than for rewarding outcomes of the same magnitude.

smaller pursuit (red), under what are otherwise the same circumstances. Therefore, the 'Magnitude Effect', as observed in humans and animals, would also be exhibited by a reward-rate-maximizing agent.

## The Sign Effect

The Sign Effect refers to the observation that the discounting functions for outcomes of the same magnitude but opposite valence (rewards and punishments) appear to discount at different rates, with punishments discounting less steeply than rewards. Should the brain apply a discounting function to outcomes to account for their temporal delays, why does it seemingly use different discount functions for rewards and punishments of the same magnitude? *Figure 16* considers how a reward-rate-maximizing agent would appear to differently discount outcomes of the same magnitude but opposite valence (rewards-*Figure 16A* v punishments-*Figure 16B*) when the time spent outside the considered pursuit type generates a non-zero reward rate. By determining the subjective value of these oppositely signed rewarding (*Figure 16C*) and punishing (*Figure 16D*) outcomes across a range of delays and plotting their normalized subjective values at their corresponding delay (*Figure 16E*), the apparent discounting function for reward and punishment, as expressed by a reward-rate-maximizing agent, exhibits the 'Sign effect', as observed in humans and animals. In addition, we note that the difference in the discounting function slopes between rewards and punishments of equal magnitude would diminish as the outside reward rate approached zero. With an outside reward rate equal to zero, the discounting functions would be identical. If the outside reward rate became increasingly negative, the effect would invert and rewards would increasingly appear to discount less steeply than punishments.

## Summary

In the above sections, we provide a richer understanding of the origins of time's cost in evaluating the worth of initiating a pursuit. We demonstrate that the intuitive, if deceptively simple, equation for subjective value (*Equation 7*), where time's cost is subtracted from the reward magnitude, is equivalent to subtracting an opportunity cost *and* an apportionment cost of time from the reward magnitude (*Equation 8*). Where time's cost in the simple equation is calculated from the global reward rate under a policy of accepting the considered pursuit (*Equation 7*), parceling the world into contributions from time spent 'inside' and 'outside' the considered pursuit type (*Equation 8*) reveals that the opportunity cost of time arises from the global reward rate achieved under a policy of *not* accepting the considered pursuit (its outside reward rate), and that the apportionment cost of time arises from the allocation of time spent in, versus outside, the considered pursuit. These equivalent expressions for the normatively-defined (reward-rate maximizing) subjective value of a pursuit give rise to an apparent discounting function that is (1) a hyperbolic function of time, (2) whose curvature is determined by the apportionment cost of time, and (3) whose scaling is linearly determined by the opportunity cost of time. By re-expressing reward-rate maximization as its apparent temporal discounting function, we demonstrate how fits of hyperbolic discounting, as well as observations of the Delay, Magnitude, and Sign effect—commonly taken as signs of suboptimal decision-making—are in fact consistent with optimal temporal decision-making.

## Sources of error and their consequences

While these added insights enrich our understanding of time's cost and reveal how purported signs of irrationality can in fact be consistent with a reward-rate-maximizing agent, it nonetheless remains true that animals and humans *are* suboptimal temporal decision makers—exhibiting an 'impatience' by selecting SS options in cases where selecting LL options would maximize global reward rate. However, when decisions to accept or reject pursuits are presented in Forgo situations, they are observed to be optimal. As the equivalent immediate reward equations enabling global reward rate optimization may potentially be instantiated by neural representations of their underlying variables, we conjecture that misrepresentation of one or another variable may best explain the particular ways in which observed behavior deviates, *as well as accords*, with optimality. Therefore, we now ask what errors in temporal decision-making behavior would result from misestimating these variables, with the aim of identifying the nature of misestimation that best accounts for the pattern actually observed in animals and humans regarding whether to initiate a given pursuit.

To understand how systematic error in an agent's estimation of different time and/or reward variables would affect its behavior, we examine the agent's pattern of behavior in both Choice and Forgo decisions across different outside reward rates. First, we ask whether the agent would choose a SS or LL pursuit as in a choice task. Then we ask whether the agent would take or forgo the same LL and SS pursuits when either are presented alone in a forgo task. The actions taken by the agent can therefore be described as a triplet of policies referring to the two pursuits (e.g. *choose SS, forgo LL, forgo SS*).

Let us first consider how a reward-rate-optimal agent would transition from one to another pattern of decision-making as outside reward rate increases for the situation of fundamental interest: where the reward rate of the SS pursuit is greater than that of the LL pursuit (*Figure 17*). When the outside reward rate (slope of golden line) is sufficiently low (*Figure 17A*), the agent should prefer LL in Choice, be willing to take the LL pursuit in Forgo, and be willing to take the SS pursuit in Forgo (choose LL, take LL, take SS). Here, a 'sufficiently low' outside rate is one such that the resulting global reward rate (slope of magenta line) is less than the difference in the reward rates of the SS and LL pursuits. When the outside reward rate increases to greater than this difference in the pursuits' reward rates but is less than the reward rate of the LL option, the agent should choose SS in Choice and be willing to take either in Forgo (choose SS, take LL, take SS *Figure 17B*). Further increases in outside rate up to that equaling the reward rate of the SS results in the agent selecting the SS in Choice, forgoing LL in Forgo, and taking SS in Forgo (choose SS, forgo LL, take SS; *Figure 17C*). Finally, any additional increase in outside rate would result in choosing the SS pursuit under Choice, and forgoing both pursuits in Forgo (choose SS, forgo LL, forgo SS; *Figure 17D*). Colored regions thus describe the pattern of decision-making behavior exhibited by a reward-rate-optimal agent under any combination of outside reward and time.

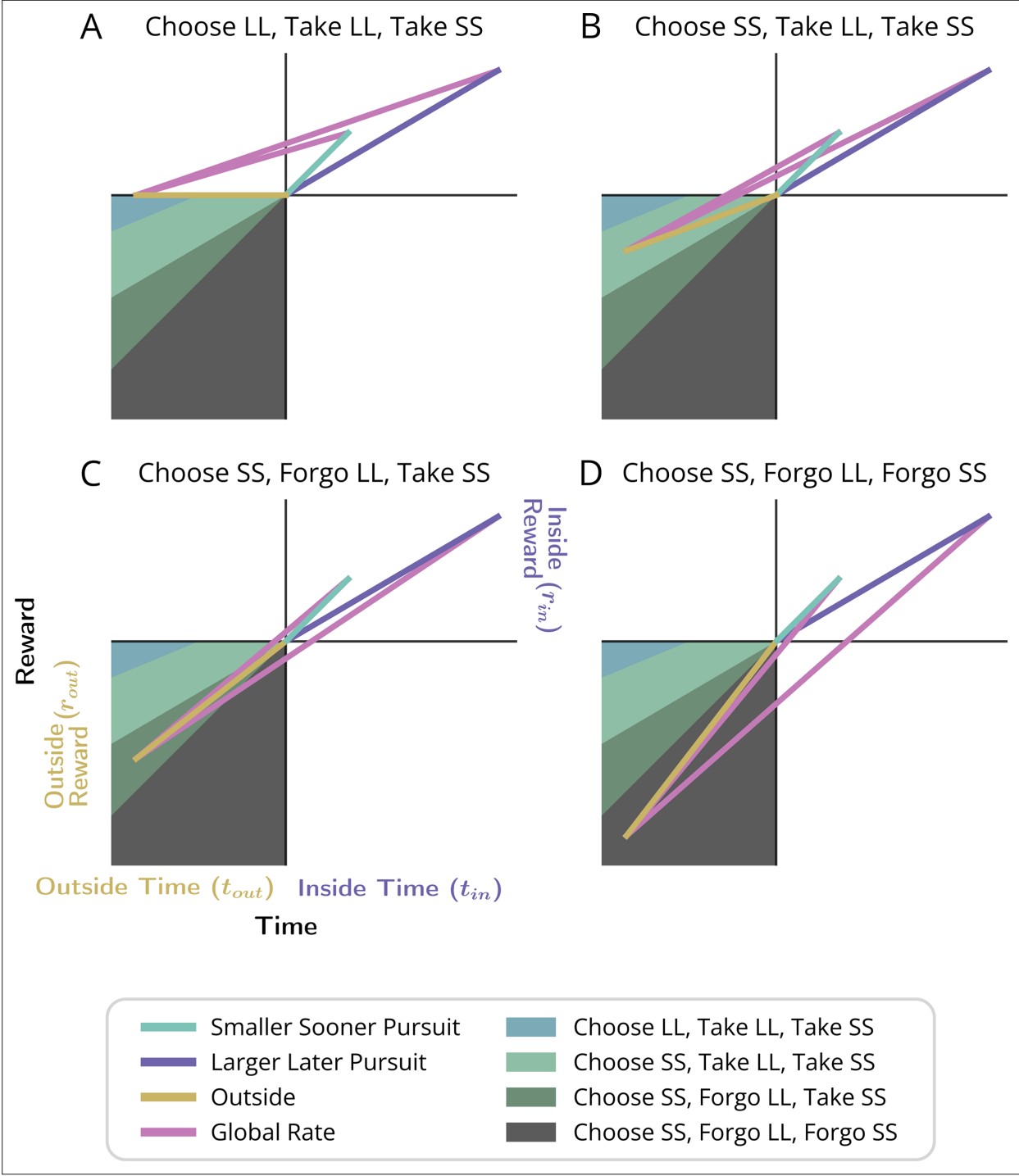

**Figure 17.** Relationship between outside time and reward with optimal temporal decision-making behavioral transitions. An agent may be presented with three decisions: the decision to take or forgo a smaller, sooner reward of 2.5 units after 2.5 s (SS pursuit), the decision to take or forgo a larger, later reward of 5 units after 8.5 s (LL pursuit), and the decision to choose between the SS and LL pursuits. The slope of the purple line indicates the global reward rate ($\rho_g$) resulting from a Choice or Take policy, while the slope of 'outside' the pursuit (golden line) indicates the outside reward rate (i.e. global reward rate resulting from a Forgo policy). In each panel (**A-D**), an example outside reward rate is plotted, illustrating the relative ordering of $\rho_g$ slopes for each policy. Location in the lower left quadrant is thereby shaded according to the combination of global rate-maximizing policies for each of the three decision types. $r_{ss}$ = 2.5, $t_{ss}$ = 2.5, $r_{ll}$ = 5, $t_{ll}$ = 8.5, $t_{out}$ = 6, $r_{out}$ = 0, 0.4, 0.8, 1.3 in A-D, respectively.

With this understanding of the optimal thresholds between behavior policies, we can now examine the impact on decision-making behavior of different types of error in the agent's understanding of the world (*Figure 18*). We introduce an error term, $\omega$, such that different parameters impacting the global reward rate of each considered policy are underestimated ($\omega<1$) or overestimated ($\omega>1$) (*Figure 18* column 1, see *Table 1* for formal definitions). Resulting global reward rate mis-estimations are equivalent to introducing error in the considered pursuit's subjective value, which will result in various deviations from reward-rate maximization (*Figure 18*). Conditions wherein overestimation of global reward rate would lead to suboptimal choice behavior are identified formally in Appendix 11.

The sources of error considered are mis-estimations of the reward obtained and/or time spent 'outside' (rows B-D) and 'inside' (rows E-G) the considered pursuit. When both reward and time are misestimated, we examine the case in which the reward rate of that portion of the world is maintained (rows D and G). The agent's resulting policies in Choice (second column) and both Forgo situations (third and fourth columns) are determined across a range of outside reward rates (x-axes) and degrees of parameter misestimation (y-axes) and color-coded, with the boundary between the colored regions indicating the outside reward rate threshold for transitions in the agent's behavior. These individual policies are collapsed into the triplet of behavior expressed across the decision types (fifth column). In this way, characterization of the nature of suboptimality is aided by the use of the outside reward rate as the independent variable influencing decision-making, with the outside reward rate thresholds for optimal behavior being compared to the outside reward rate thresholds under any given parameter misestimation (comparing top 'optimal' row A, against any subsequent row B-G). Any deviations in this pattern of behavior from that of the optimal agent (row A) are suboptimal, resulting in a failure to maximize reward rate in the environment.

While misestimation of any of these parameters will lead to suboptimal behavior, only specific sources and directions of error may result in behavior that qualitatively matches human and animal behavior observed experimentally. Misestimation of outside time (B), outside reward (C), inside time (E), and inside reward (F) all display Choice behavior that is qualitatively similar to experimentally observed behavior, either via underestimation or overestimation of the key variable. For example, underestimation of the outside time (B, $\omega<1$) leads to selection of the SS pursuit at sub-optimally low outside reward rates. However, agents with these types of error never display optimal Forgo behavior. By contrast, misestimation of either outside time and reward (D) or inside time and reward (G) display suboptimal Choice while maintaining optimal Forgo. Specifically, underestimation of outside time and reward (D, $\omega<1$) and overestimation of inside time and reward (G, $\omega>1$) both result in suboptimal preference for SS at low outside rates. Therefore, and critically, if the rates of both inside and outside are maintained despite misestimating reward and time magnitudes, the resulting errors allow for optimal Forgo behavior while displaying suboptimal 'impatience' in Choice, and thus match experimentally observed behavior.

## Discussion

In order to understand why humans and animals factor time the way they do in temporal decision-making, our initial step has been to understand how a reward-rate-maximizing agent would evaluate the worth of initiating a pursuit within a temporal decision-making world. We did so in order to identify what are and are not signs of suboptimality and to gain insight into how animals' and humans' valuation of pursuits actually deviate from optimality. By analyzing fundamental temporal decisions, we identified equations enabling reward-rate maximization that evaluate the worth of initiating a pursuit. We first considered *Forgo* decisions to appreciate that a world can be parcellated into its constituent pursuits, revealing how pursuits' rates and relative occupancies (their 'weights'), along with the decision policy, determine the global reward rate. In doing so, we derived an expression for the worth of a pursuit in terms of the resulting global reward rate. From it, we re-expressed the pursuit's worth in terms of its global reward rate-equivalent immediate reward, that is its 'subjective value', reprising McNamara's foundational expression (*McNamara, 1982*). We then show that subjective value, rather than being calculated from the global reward rate under a policy of accepting the considered pursuit, can equally be calculated in terms of the outside reward rate (a policy of *not* accepting the considered pursuit type) and the proportion of time spent outside the pursuit. Expressing subjective value in terms of a pursuit's outside reward rate and time reveals that time's cost is constituted by an apportionment cost, as well as an opportunity cost. By then examining *Choice* decisions, we provide

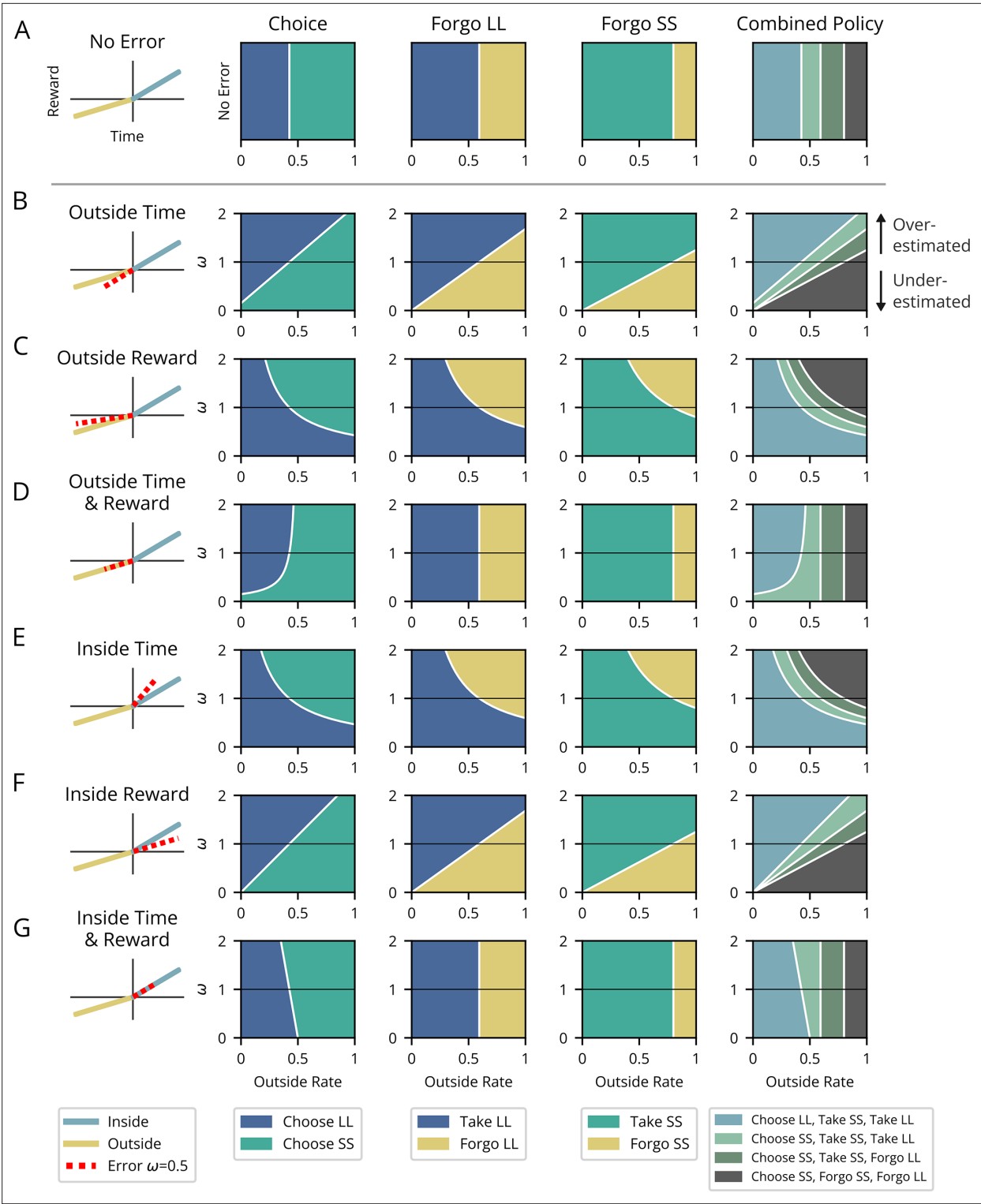

**Figure 18.** Patterns of suboptimal temporal decision-making behavior resulting from time and/or reward misestimation. Patterns of temporal decision-making in Choice and Forgo situations deviate from optimal (top row) under various parameter misestimations (subsequent rows). Characterization of the nature of suboptimality is aided by the use of the outside reward rate as the independent variable influencing decision-making (x-axis), plotted against the degree of error (y-axis) of a given parameter ($\omega<1$ underestimation, $\omega=1$ actual, $\omega>1$ overestimation). The leftmost column provides a schematic exemplifying true outside (gold) and inside (blue) pursuit parameters and the nature of parameter error (dashed red) investigated per row (all showing an instance of underestimation). For each error case, the agent's resulting choice between SS and LL pursuits (2nd column), decision to take or forgo the LL pursuit (3rd column), and decision to take or forgo the SS pursuit (4th column) are indicated by the shaded color (legend, bottom

*Figure 18 continued on next page*

*Figure 18 continued*

of columns) for a range of outside rates and degrees of error. The rightmost column depicts the triplet of behavior observed, combined across tasks. Rows: (**A**) 'No error' - Optimal choice and forgo behavior. Vertical white lines show outside reward rate thresholds for optimal forgo behavior. (**B-G**) Suboptimal behavior resulting from parameter misestimation. (**B-D**) The impact of outside pursuit parameter misestimation. (**B**) 'Outside Time'- The impact of misestimating outside time (and thus misestimating outside reward rate). (**C**) 'Outside Reward'- The impact of misestimating outside reward (and thus misestimating outside reward rate). (**D**) 'Outside Time and Reward'- The impact of misestimating outside time and reward, but maintaining outside reward rate. (**E-G**) The impact of inside pursuit parameter misestimation. (**E**) 'Pursuit Time'- The impact of misestimating inside pursuit time and thus misestimating inside pursuit reward rate. (**F**) 'Pursuit Reward' - The impact of misestimating the pursuit reward (and thus misestimating the pursuit reward rate). (**G**) 'Pursuit Time and Reward' - The impact of misestimating the pursuit reward and time, but maintaining the pursuit's reward rate. For this illustration, we determined the policies for a SS pursuit of 2 reward units after 2.5 s, a LL pursuit of 4.75 reward units after 8 s, and an outside time of 10 s. The qualitative shape of each region and resulting conclusions are general for all situations where the SS pursuit has a higher rate than the LL pursuit (and where a region exists where the optimal agent would choose LL at low outside rates).

a deeper understanding of the nature of apparent temporal discounting in reward-rate-maximizing agents and establish that hyperbolic discounting, the Delay Effect, the Magnitude Effect, and the Sign Effect, are *not* signs of suboptimal decision-making, but rather are consistent with reward-rate maximization. While these purported signatures of suboptimality would in fact arise from reward-rate maximization, humans and animals are, nonetheless, suboptimal temporal decision makers, exhibiting apparent discounting functions that are too steep. By examining misestimation of the parameters that enable reward-rate maximization identified here, we implicate underestimation of the relative time spent outside versus in the considered pursuit type as the likely source of error committed by animals and humans in temporal decision-making that underlies their suboptimal pursuit valuation. We term this "The Malapportionment Hypothesis".

## Temporal decision-making theories and frameworks

Two theories have predominated over the course of theorizing about how animals should invest time when pursuing rewards of a diversity of magnitudes and delays: a theory of exponential discounting (*Samuelson, 1937*; *Frederick et al., 2002*; *Kalenscher and Pennartz, 2008*) and a theory of optimal foraging (*Charnov, 1976b*; *Pyke et al., 1977*; *Stephens and Krebs, 1987*; *Stephens, 2008*). According to the former, exhibiting a permanent preference for one option over another through time was argued to be rational (*Montague and Berns, 2002*; *Mazur, 2006*; *Nakahara and Kaveri, 2010*), as in Discounted Utility Theory (DUT) (*Samuelson, 1938*). Discounting functions operating under this principle would then be exponential, with the best fit exponent controlling and embodying the agent's appreciation of the cost of time. In contrast, OFT invoked reward-rate maximization as the normative principle. Referenced by a wide assortment of ethologists and ecologists (for review, see

**Table 1.** Definitions for misestimating global reward rate-enabling parameters.

Each misestimated variable (column 1) is multiplied by an error term, $\omega$, to give $\hat{\rho}_g$, the misestimated global reward rate (column 2). When $\omega = (0, 1)$ the variable is underestimated, when $\omega = (1, 2)$ the variable is overestimated, and when $\omega = 1$ the variable is correctly estimated and $\hat{\rho}_g = \rho_g$.

| Misestimated Variable | Misestimated Global Reward Rate |
| --- | --- |
| True (No Misestimation) | $\rho_g = \frac{r_{in} + \rho_{out} t_{out}}{t_{in} + t_{out}}$ |
| Outside Time | $\hat{\rho}_g = \frac{r_{in} + \rho_{out} t_{out}}{t_{in} + \omega t_{out}}$ |
| Outside Reward | $\hat{\rho}_g = \frac{r_{in} + \omega \rho_{out} t_{out}}{t_{in} + t_{out}}$ |
| Outside Time and Reward (maintaining $\rho_{out}$) | $\hat{\rho}_g = \frac{r_{in} + \omega \rho_{out} t_{out}}{t_{in} + \omega t_{out}}$ |
| Inside Time | $\hat{\rho}_g = \frac{r_{in} + \rho_{out} t_{out}}{\omega t_{in} + t_{out}}$ |
| Inside Reward | $\hat{\rho}_g = \frac{\omega r_{in} + \rho_{out} t_{out}}{t_{in} + t_{out}}$ |
| Inside Reward and Time (maintaining $\rho_{in}$) | $\hat{\rho}_g = \frac{\omega r_{in} + \rho_{out} t_{out}}{\omega t_{in} + t_{out}}$ |

*Pyke, 1984*), the specific formulation proponents of OFT generally use would result in an apparent discounting function that is hyperbolic. Indeed, in controlled laboratory experiments in which animals make decisions about how to spend time between rewarding options (*Hariri et al., 2006*; *Hayden et al., 2011*; *Wikenheiser et al., 2013*; *Blanchard and Hayden, 2014*; *Blanchard and Hayden, 2015*; *Carter et al., 2015*; *Carter and Redish, 2016*), experimental observations have demonstrated that hyperbolic functions are better fits to choice behavior in intertemporal choice tasks than exponential functions (*Ainslie, 1975*; *Thaler and Shefrin, 1981b*; *Frederick et al., 2002*; *Green and Myerson, 2004*; *Kim et al., 2008*; *Blanchard and Hayden, 2015*). Nonetheless, and problematically for OFT, in most intertemporal choice tasks, animal behavior is far from optimal for maximizing reward rate (*Reynolds and Schiffbauer, 2004*; *Hayden et al., 2011*; *Blanchard et al., 2013*; *Blanchard and Hayden, 2015*).

## Hyperbolic temporal discounting functions

Indeed, with respect to global reward-rate maximization, animals and humans typically exhibit much too great a preference for SS rewards in apparent discounting of delayed rewards (*Chung and Herrnstein, 1967*; *Rachlin et al., 1972*; *Ainslie, 1974*; *Thaler and Shefrin, 1981b*; *Ito and Asaki, 1982*; *Grossbard and Mazur, 1986*; *Mazur, 1988*; *Benzion et al., 1989*; *Loewenstein and Prelec, 1992*; *Green et al., 1994*; *Bateson and Kacelnik, 1996*; *Kacelnik and Bateson, 1996*; *Cardinal et al., 2001*; *Stephens and Anderson, 2001*; *Bennett, 2002*; *Frederick et al., 2002*; *Holt et al., 2003*; *Winstanley et al., 2004*; *Kalenscher et al., 2005*; *Roesch et al., 2007*; *Kobayashi and Schultz, 2008*; *Louie and Glimcher, 2010*; *Pearson et al., 2010*). More precisely, we show here that what is meant by this suboptimal bias for SS is that the switch in preference from LL to SS occurs at an outside reward rate that is lower—and/or an outside time that is greater—than what an optimal agent would exhibit. It was *Ainslie, 1975* who first understood that the empirically observed 'preference reversals' between SS and LL pursuits could be explained if temporal discounting took on a hyperbolic form, which he initially conjectured to arise simply from the ratio of reward to delay (*Grüne-Yanoff, 2015*). This was problematic, however, on two fronts: (1) as the time nears zero, the value curve goes to infinity, and (2) there is no accommodation of differences observed within and between subjects regarding the steepness of discounting. *Mazur, 1987* addressed these issues by introducing 1+k into the denominator, providing for the now standard hyperbolic discounting function, $discounting\ function = \frac{sv}{r} = \frac{1}{1+kt}$. Introduction of '1' solved the first issue, although 'it never became fully clear how to interpret this 1' (*Grüne-Yanoff, 2015* interviewing Ainslie). Introduction of the free-fit parameter, $k$, accommodated the variability observed across and within subjects by controlling the curvature of temporal discounting, and has become widely interpreted as a psychological trait, such as patience, or willingness to delay gratification (*Frederick et al., 2002*).

In this way, the Discounting Function framework has often been reified into a function possessed by the brain, an intrinsic property used to reduce, in a manner idiosyncratic to the agent, the value of delayed reward. Indeed, discounting functions have been directly incorporated into numerous models (*Laibson, 1997*; *McClure et al., 2004*; *al-Nowaihi and Dhami, 2008*; *Killeen, 2009*), motivating the search for its neurophysiological signature (*Montague et al., 2006*). In addition to accommodating intra- and inter-subject variability through the use of this free-fit parameter, discounting function formulations must also contend with the fact that best fits differ in steepness (1) when the time spent and (2) reward gained outside the pursuit changes (*Lea, 1979*; *Stephens and Dunlap, 2009*; *Blanchard et al., 2013*; *Blanchard and Hayden, 2015*; *Carter et al., 2015*; *Smethells and Reilly, 2015*; *Carter and Redish, 2016*), (3) as the delay to SS and LL pursuits increase (the Delay Effect), (4) when the reward magnitude of the pursuit changes (the Magnitude Effect), and (5) when considering the sign of the outcome of the pursuit (the Sign Effect). This sensitivity to conditions and variability across and within subjects has spurred a hunt for the 'perfect' discounting function (*Namboodiri and Hussain Shuler, 2016*) in an effort to better fit behavioral observations, resulting in formulations of increasing complexity (*Laibson, 1997*; *McClure et al., 2004*; *al-Nowaihi and Dhami, 2008*; *Killeen, 2009*). While such accommodations may provide for better fits of data, the uncertain origins of discounting functions (*Hayden, 2016*) pose a challenge to the utility of this framework in rationalizing observed behavior.

## The apparent discounting function of global reward-rate-optimal agents exhibits purported signs of suboptimality

Of the array of temporal decision-making behaviors commonly observed and viewed through the lens of discounting, what might be better accounted for by a deeper understanding of how a reward-rate-optimal agent would evaluate the worth of initiating a pursuit? To address this, we derived expressions of reward-rate maximization, translated them into subjective value, and then re-expressed subjective value in terms of the apparent discounting function that would be exhibited by a reward-rate-maximizing agent. We demonstrate that a simple and intuitive equation subtracting time's cost is equivalent to a hyperbolic discounting equation that accounts for opportunity costs and the agent's apportionment of time in the environment. This analysis determines that the form and sensitivity to conditions that temporal discounting is experimentally observed to exhibit would actually be expressed by a reward-rate-maximizing agent. In doing so, we emphasize how discounting functions should be considered as descriptions of the result of a process, rather than being the process itself.

Regarding form, our analysis reveals that the apparent discounting function of a reward-rate-maximizing agent is a hyperbolic function...

$$Discounting\ Function = \frac{1 - \dfrac{\rho_{out} * t_{in}}{r_{in}}}{1 + \dfrac{1}{t_{out}} t_{in}}$$

...which resembles the standard hyperbolic discounting function, $\frac{1}{1+kt}$, in the denominator, where $k = \frac{1}{t_{out}}$. Whereas Mazur introduced $1+k$ to $t$ in the denominator to (1) force the function to behave as $t$ approaches zero and (2) provide a means to accommodate differences observed within and between

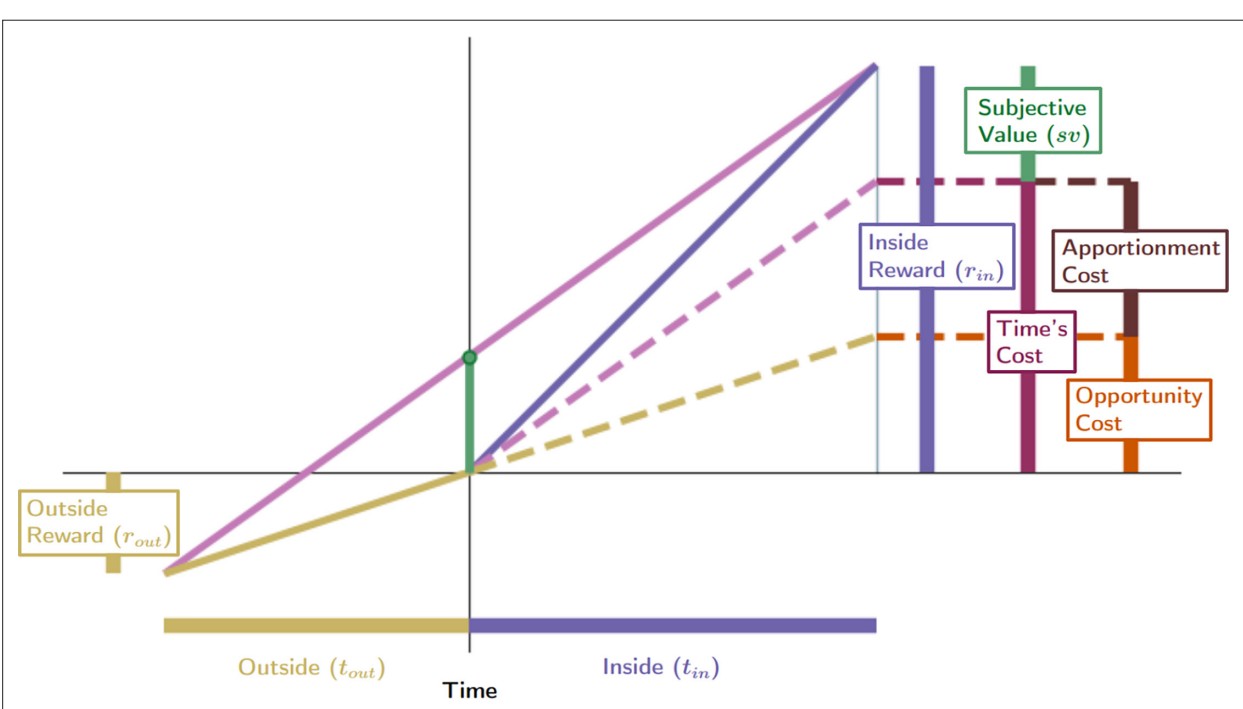

**Figure 19.** The cost of time of a pursuit comprises both an opportunity as well as an apportionment cost. The global reward rate under a policy of accepting the considered pursuit type (slope of magenta time), times the time that pursuit takes ($t_{in}$), is the pursuit's *time's cost* (height of maroon bar). The subjective value of a pursuit (height of green bar) is its reward magnitude (height of the purple bar) less its cost of time. Opportunity and apportionment costs are shown to compose the cost of time of a pursuit. Opportunity cost associated with a considered pursuit, $\rho_{out}*t_{in}$ (height of orange bar) is the reward rate of the world under a policy of *not accepting* the considered pursuit (its outside rate), $\rho_{out}$, times the time of the considered pursuit, $t_{in}$. Therefore, *opportunity cost* is the amount of reward that would be expected from a policy of not taking the considered pursuit over a time equal to the considered pursuit's duration. The difference in reward that can be expected, on average, between a policy of taking versus a policy of not taking the considered pursuit, over a time equal to its duration, is the *apportionment cost* of time (height of brown bar). Together, they sum to time's cost.

**Table 2.** Opportunity cost, apportionment cost, time cost, and subjective value functions by change in outside and inside reward and time.

Functions assume positive inside and outside rewards and times.

| | Reward | | Time | |
| --- | --- | --- | --- | --- |
| | **Outside** | **Inside** | **Outside** | **Inside** |
| **Opportunity Cost*** | Linear<br>Positive slope | No Effect | Hyperbolic<br>Negative slope | Linear<br>Positive slope |
| **Apportionment Cost** | Linear<br>Negative slope | Linear<br>Positive slope | Hyperbolic - Hyperbolic[†]<br>Negative slope | Hyperbolic - Linear[†]<br>Negative slope |
| **Time's Cost** | Linear<br>Positive slope | Linear<br>Positive slope | Hyperbolic<br>Negative slope | Hyperbolic<br>Positive slope |
| **Subjective Value** | Linear<br>Negative slope | Linear<br>Positive slope | Hyperbolic<br>Positive Slope | Hyperbolic<br>Negative slope |

*If outside reward rate is zero, opportunity cost becomes a constant at zero.

[†]If outside reward rate is zero, as outside or inside time is varied, apportionment cost becomes purely hyperbolic.

subjects, our derivation gives cause to the terms 1 and k, their relationship to one another, and to $t$ in the denominator. First, from our derivation, '1' actually signifies taking $t_{out}$ amount of time expressed in units of $t_{out}$, ($t_{out}/t_{out} = 1$) and adding it to $t_{in}$ amount of time expressed in units of $t_{out}$ (ie, the total time to make a full pass through the world expressed in terms of how the agent apportions its time under a policy of accepting the considered pursuit). Absent from the numerator in the standard hyperbolic formulation, the solution for a reward-rate-maximizing agent gives rise to a term that accounts for the opportunity cost of time. Together, the diminishment of the value of a pursuit as its time investment increases is thus due to time's cost—itself hyperbolic—which is shown to be composed of an apportionment (hyperbolic – linear) as well as an opportunity cost (linear) (*Figure 19*; *Table 2*).

In addition to demonstrating the form of the discounting function of an optimal agent, we can now also rationalize why it would appear to change in relationship to the features of the temporal decision-making world. *First*, rather than being a free-fit parameter like $k$ in hyperbolic discounting models (*Figure 20A*), the reciprocal of the time spent outside the considered pursuit type controls the degree of curvature in reward-rate optimizing agents (*Figure 20B*, denominator). Therefore, changes in the apparent 'willingness' of a reward-rate-optimal agent to wait for reward would accompany any change in the amount of time that that agent needs to spend outside the considered pursuit, making the agent act as if more patient the greater the time spent outside a pursuit for every instance spent within it. Indeed, experiments with shorter intertrial intervals, and thus higher global reward rates, have a higher cost of time, and therefore exhibit steeper apparent temporal discounting (*Lea, 1979*; *Stephens and Dunlap, 2009*; *Blanchard et al., 2013*; *Smethells and Reilly, 2015*; *Carter and Redish, 2016*; *Hayden, 2016*).

*Second*, discounting frameworks must also rationalize why the steepness of apparent discounting changes as the reward rate acquired outside the considered pursuit changes. We show increasing the outside reward rate to be related to the linear opportunity cost of time in a reward-rate-maximizing agent (*Figure 13*), subtraction of opportunity cost occurring in the numerator. The greater the opportunity cost of time, the steeper the apparent discounting function and the less patient the agent would appear to be. In animal experiments in which a menu of options with different magnitudes and delays are presented as choice pairs, the outside reward rate will be greater than zero (*Lea, 1979*; *Blanchard et al., 2013*; *Carter et al., 2015*; *Carter and Redish, 2016*; *Hayden, 2016*). In these cases, hyperbolic fits should be steeper than would otherwise be the case if there were no outside reward rate. As typical hyperbolic discounting functions are of the form...

$$Discount\ Function = \frac{1}{1 + kt}$$

...they lack an accounting of the opportunity cost, and therefore must compensate for the lack of an opportunity cost term by overestimating $k$ (which rightfully should only relate to the apportionment cost). Discounting models of the standard hyperbolic form are therefore only appropriate in

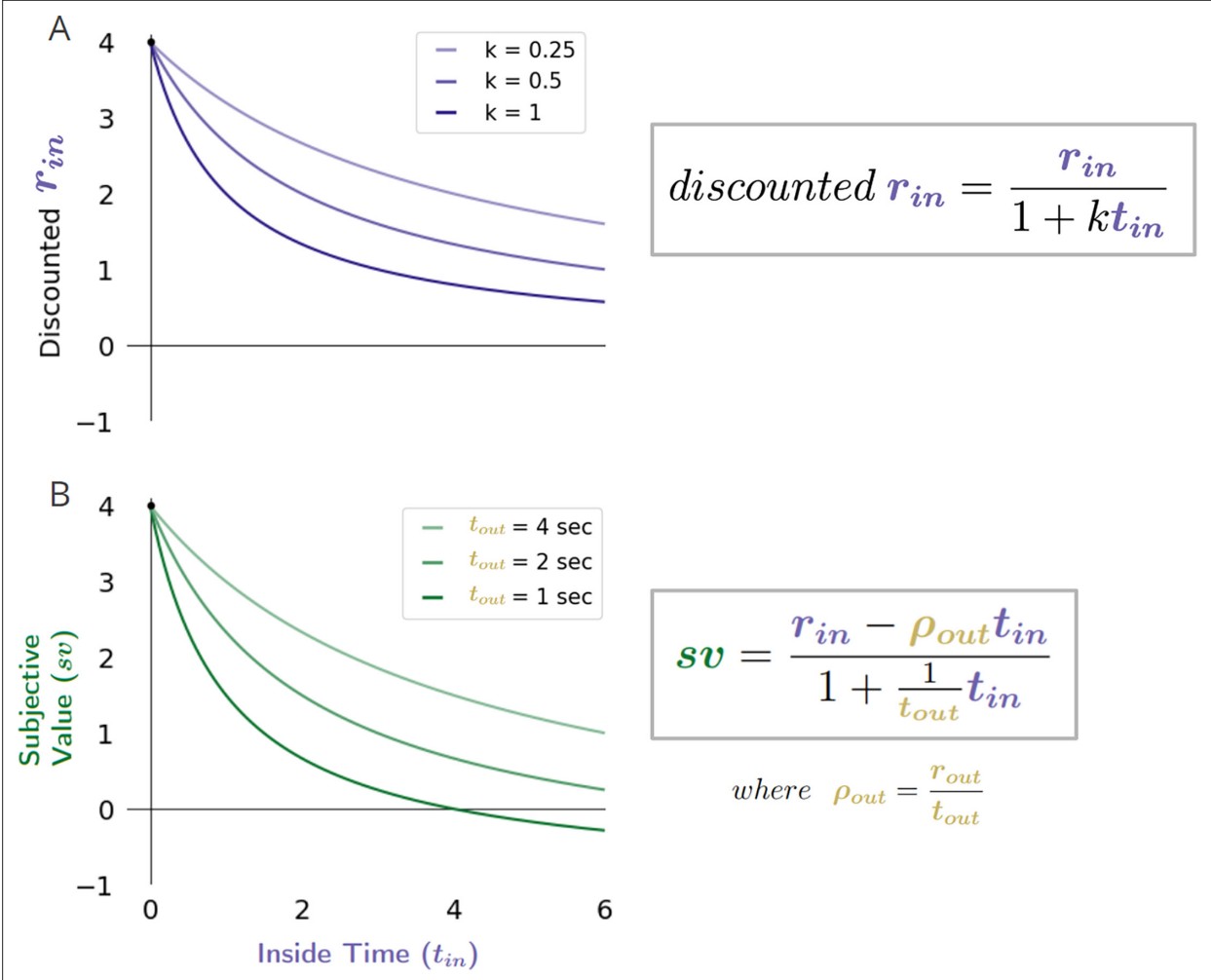

**Figure 20.** Comparison of typical hyperbolic discounting versus apparent discounting of a reward-rate-optimal agent. Whereas (**A**) the curvature of hyperbolic discounting models is typically controlled by the free fit parameter $k$, (**B**) the curvature and steepness of the apparent discounting function of a reward-rate-optimal agent is controlled by the time spent and reward rate obtained outside the considered pursuit. Understanding the shape of discounting models from the perspective of a reward-rate-optimal agent reveals that $k$ ought relate to the apportionment of time spent in, versus outside, the considered pursuit, underscoring, how typical hyperbolic discounting models fail to account for the opportunity cost of time (and thus cannot yield negative $sv$'s no matter the temporal displacement of reward). Should $k$ be understood as representing time's apportionment cost, the failure to account for the opportunity cost of time would lead to aberrantly high values of $k$.

worlds with no 'outside' reward, as under such conditions opportunity cost is zero. In such worlds, the discounting function of a reward-rate-maximizing agent becomes...

$$Discount\ Function = \frac{1}{1 + \frac{1}{t_{out}} t_{in}}$$

which fits a purely hyperbolic equation, where

$$k = \frac{1}{t_{out}}.$$

Relating to the treatment of opportunity cost, we also note that many investigations into temporal discounting often do not make an explicit distinction between situations in which (1) subjects continue to receive the usual rewards from the environment during the delay to a chosen pursuit and (2) situations in which during a chosen pursuit's delay no other rewards or opportunities will occur (**Glimcher et al., 2007**; **McClure et al., 2004**). Commonly, human subjects are asked to answer questions about

their preferences between options for amounts they will not actually earn after delays they will not actually have to wait, during which it is unclear whether they are really investing time away from other options or not (*Rosati et al., 2007*). In contrast, in most animal experiments, subjects actually receive reward after different delays during which they *do not* receive new options or rewards. By our formulation, when being engaged in a pursuit does not exclude the agent from receiving rewards at the rate that occurs outside, the opportunity cost of time drops out of the subjective value equation (see *Equation 12*, The value of initiating a pursuit when pursuit does not exclude receiving rewards at the outside rate, and see Appendix 12).

$$sv = \frac{r_{in}}{1 + t_{in}/t_{out}} \tag{12}$$

*Equation 12*.

Therefore, the reward-rate-maximizing discounting function in these worlds is functionally equivalent to the situation in which the outside reward rate is zero, and will—lacking an opportunity cost—be less steep. This rationalizes why human discounting functions are often reported to be less steep than animal discounting functions: they are typically tested in conditions that negate opportunity cost, whereas animals are typically tested in conditions that enforce opportunity costs. Indeed, when humans are made to wait for actually received reward, their observed discounting functions are much steeper (*Jimura et al., 2009*).

Another consequence of opportunity cost as it relates to apparent temporal discounting functions is that the presence of opportunity cost means that the apparent temporal discounting function will become negative at sufficiently long delays, so that even rewarding pursuits, if not forced, should be forgone. The greater the opportunity cost of time, the steeper the apparent discounting function, and the earlier a rewarding pursuit will become that the agent ought forgo (when their acceptance would yield rates less than the outside rate, i.e. when *sv* <0). The standard hyperbolic discounting function, lacking accounting of opportunity cost, cannot fit negative subjective values and therefore would not predict when—or even that—an agent will forgo a rewarding pursuit.

Third, fourth, and fifth, discounting frameworks must make an accounting of the Delay Effect, Magnitude Effect, and Sign Effect, respectively, as they are considered important 'anomalous' departures from microeconomic theory (*Loewenstein and Thaler, 1989*; *Cruz Rambaud et al., 2023*). To do so, rationalizations from previous works have invoked additional assumptions, such as separate processes for small and large rewards (*Thaler, 1981a*), reference points (*Kahneman and Tversky, 1979*), or the inclusion of an elastic utility function (*Loewenstein and Prelec, 1992*; *Killeen, 2009*), among a myriad of other explanations (see (*Frederick et al., 2002*; *Kalenscher and Pennartz, 2008*) for reviews), including scalar timing (*Gibbon, 1977*) accounts of non-stationary time preference. While any of these proposals may indeed impact valuation, we emphasize here that qualitative features that they are invoked to explain are consistent with expectations of a reward-rate-maximizing agent.

## Delay Effect

The inside/outside pursuit perspective makes clear that the 'Delay Effect', for instance, is not to be thought of as a sign of suboptimal decision making, but rather is what one should expect from a reward-rate-maximizing agent. As the resulting global reward rates drop due to an increasing delay to SS and LL pursuits, a switch will occur (see *Figure 14*) from a policy of choosing the SS pursuit to a policy of choosing the LL when the difference in the reward rates of these pursuits equals the global reward rates that would result from their selection. From the perspective of temporal discounting, this is understood to result from a hyperbolic discounting form (*Ainslie, 1975*; *Kalenscher and Pennartz, 2008*; *Fawcett et al., 2012*), shown here to be exhibited by a reward-rate-maximizing agent. Preference reversal, then, is not a sign of irrational decision making, as it would be exhibited by a reward-rate-maximizing agent. Rather, a sign of irrationality is that a preference reversal would occur at delays greater or less than what a reward-rate-maximizing agent would exhibit.

## Magnitude Effect

We also demonstrate how the 'Magnitude Effect' would be a natural consequence of a process that would maximize reward rate, without invoking specialized processes or additional functions (*Figure 15*). Further, from our reward-rate-maximizing framework, how the size of the Magnitude

Effect would be affected by changing, experimentally, the outside time, reward, and rate parameters can be predicted. The Magnitude Effect should be observed, experimentally, to diminish when (1) increasing the outside time while holding the outside reward constant, (thus decreasing the outside reward rate), or when (2) decreasing the outside reward while holding the outside time constant (thus decreasing the outside reward rate). However, (3) the Magnitude Effect would exaggerate as the outside time increased while holding the outside reward rate constant.

## Sign Effect

Whereas discounting frameworks need to invoke separate discounting functions to contend with different discounting rates for positive (rewarding) and negative (punishing) outcomes of the same magnitude (the Sign Effect), here too, we demonstrate how this is consistent with a reward-rate-maximizing process (*Figure 16*). The asymmetry in the steepness of apparent discounting to rewards and punishments results from the average time and magnitude of rewards (or punishments) received outside the considered pursuit, forming a bias in evaluating equivalently sized outcomes of opposite sign. From the global reward-rate-maximizing perspective, we then also predict that the size of the Sign effect would diminish as the outside reward rate decreases (and as the outside time increases), and in fact would invert should the outside reward rate turn negative (become net punishing), such that punishments would appear to discount more steeply than rewards.

Collectively, our analysis of discounting functions reveals that features typically taken as signs of suboptimal/irrational decision-making are, in fact, consistent with reward-rate maximization. In this way, the general form and sensitivity to conditions of discounting functions, as observed experimentally, can be better understood from the perspective of a reward-rate-optimal agent (*Table 2*), providing a more parsimonious accounting of a confusing array of temporal decision-making behaviors reported.

## Humans and animals are nonetheless suboptimal. What is the nature of this suboptimality?

These insights into the behavior of a reward-rate-maximizing agent inform on the meaning of the concept "patience". Patience oughtn't imply a willingness to wait a longer time, as it is not correct to say that an agent that chooses a pursuit requiring a long time investment is more patient that one that does not, for the amount of time a reward-rate-maximizing agent is willing to invest isn't an intrinsic property of the agent itself. Rather, it is a consequence of the temporal decision-making world's reward-time structure. So, if patience is to mean investing the 'correct' amount of time (i.e. the reward-rate-maximizing time), then a reward-rate-optimal agent doesn't *become* more or less patient as the context of what is otherwise the same pursuit changes; rather, it is *precisely* patient, under all circumstances. Impatience and over-patience then are terms to describe the behavior of a global reward-rate *suboptimal* agent that invests either too little, or too much time into a pursuit policy than one that would maximize global reward rate.

Having clarified what behaviors are and are not signs of suboptimality, actual differences to optimal performance exhibited by humans and animals can now be identified and quantified. So, what then are the decision-making behaviors of humans and animals when tasked with valuing the initiation of a pursuit, as in Forgo and Choice decisions? In controlled experimental situations, forgo decision-making is observed to be near optimal, consistent with observations from the field of behavioral ecology (*Stephens and Dunlap, 2009*; *Smethells and Reilly, 2015*; *Hayden, 2016*). In contrast, a suboptimal bias for SS rewards is widely reported in Choice decision-making in situations where selection of later-larger rewards would maximize global reward rate (*Logue et al., 1985*; *Blanchard and Hayden, 2015*; *Carter and Redish, 2016*; *Kane et al., 2019*). Collectively, the pattern of temporal decision-making behavior observed under Forgo and Choice decisions shows that humans and animals act as if sub-optimally impatience under Choice, while exhibiting near-optimal decision-making under Forgo decisions.

## The Malapportionment Hypothesis

How can animals and humans be sub-optimally impatient in Choice, but optimal in Forgo decisions? We postulated that previous behavioral findings of suboptimality can be understood from the perspective of overestimating the global reward rate. While misestimation of any variable underlying global

reward rate calculation will lead to errors, not all misestimations will lead to errors that match the behavioral pattern of decisions observed experimentally. Having identified equations and their variables enabling reward-rate maximization, we sought to identify the likely source of error committed by animals and humans by analyzing the pattern of behavior consequent to misestimating one or another parameter. To do so, we identified the reward rate obtained outside a considered pursuit type as a useful variable to characterize departure from optimal decision-making behavior. Sweeping over a range of these values as the independent variable, we determined change points in decision-making behavior that would arise from misestimation (over- and under-estimations) of given reward-rate-maximizing parameters.

Our analysis shows how, precisely, misestimation of the inside and outside time or reward will lead to suboptimal temporal decision-making behavior. What errors, however, result in decisions that best accord with what is observed experimentally (i.e. result in suboptimal impatience in Choice and optimal Forgo decision-making)? Overestimating outside time, underestimating outside reward, underestimating inside time, or overestimating inside reward would fail to match suboptimal 'impatience' in Choice *and* would result in suboptimal Forgo. Underestimating outside time, overestimating outside reward, overestimating inside time, or underestimating inside reward would match experimentally observed 'impatience' in Choice, but fail to match experimentally observed optimal Forgo behavior. To exhibit optimal forgo behavior, the inside and outside reward rates must be accurately appreciated. Therefore, misestimations of reward *and* time that preserve the true reward rates in and outside the pursuit would permit optimal forgo decisions while still misestimating the global reward rate. Overestimation of the outside time or underestimation of the inside time—while maintaining reward rates—fails to match experimentally observed 'impatience' in choice tasks while achieving optimal forgo decisions. However, underestimation of the outside time or overestimation of the inside time—while maintaining true inside and outside reward rates—*would* allow optimal forgo decision-making behavior while resulting in impatient choice behavior, as experimentally observed.

Previous experimental observations are consistent with, and have been interpreted as an agent underestimating the time spent outside the considered pursuit (*Stephens and Dunlap, 2009*; *Blanchard et al., 2013*; *Smethells and Reilly, 2015*), as would occur with underestimation of post-reward delays (*Bateson and Kacelnik, 1996*; *Blanchard et al., 2013*; *Namboodiri et al., 2014a*; *Fung et al., 2021*). Therefore, observed behavioral errors point to misestimating time apportionment in/outside the pursuit, either by (1) overestimating the occupancy of the considered choice or (2) underestimating the time spent outside the considered pursuit type, but not by (3) a misestimation of either the inside or outside reward rate. Only errors in time apportionment that underweight the outside time, (or, equivalently, overweight the inside time)—while maintaining the true inside and outside reward rates—will accord with experimentally observed temporal decision-making regarding whether to initiate a pursuit.

Thus, when a temporal decision world can effectively be bisected into two components, as often the case in experimental situations, only the reward rates, *but not the weights* of those portions need be accurately appreciated for the agent to optimally perform forgo decisions. Therefore, when tested in such situations, even agents that misestimate the apportionment of time can still make optimal forgo decisions based solely from a comparison of the reward rate in versus outside the pursuit. However, when faced with a choice between two or more pursuits when emerging from a path in common to any choice policy, optimal pursuit selection based on relative rate comparisons is no longer guaranteed, as *not only* the reward rates of pursuits, but *also their weights* must then be accurately appreciated. Misestimation of the weights of pursuits comprising a world would then result in errors in valuation regarding the initiation of a pursuit under choice instances.

We term this reckoning of the source of error committed by animals and humans the *Malapportionment Hypothesis*, which identifies the underweighting of the time spent outside versus inside a considered pursuit *but not the misestimation of pursuit rates*, as the source of error committed by animals and humans (*Figure 21*). This hypothesis therefore captures previously published behavioral observations (*Figure 21A*) showing that animals can make decisions to take or forgo reward options that optimize reward accumulation (*Krebs et al., 1977*; *Stephens and Krebs, 1987*; *Blanchard and Hayden, 2014*), but make suboptimal decisions when presented with simultaneous and mutually exclusive choices between rewards of different delays (*Logue et al., 1985*; *Blanchard and Hayden, 2015*; *Carter and Redish, 2016*; *Kane et al., 2019*). The Malapportionment Hypothesis further predicts that

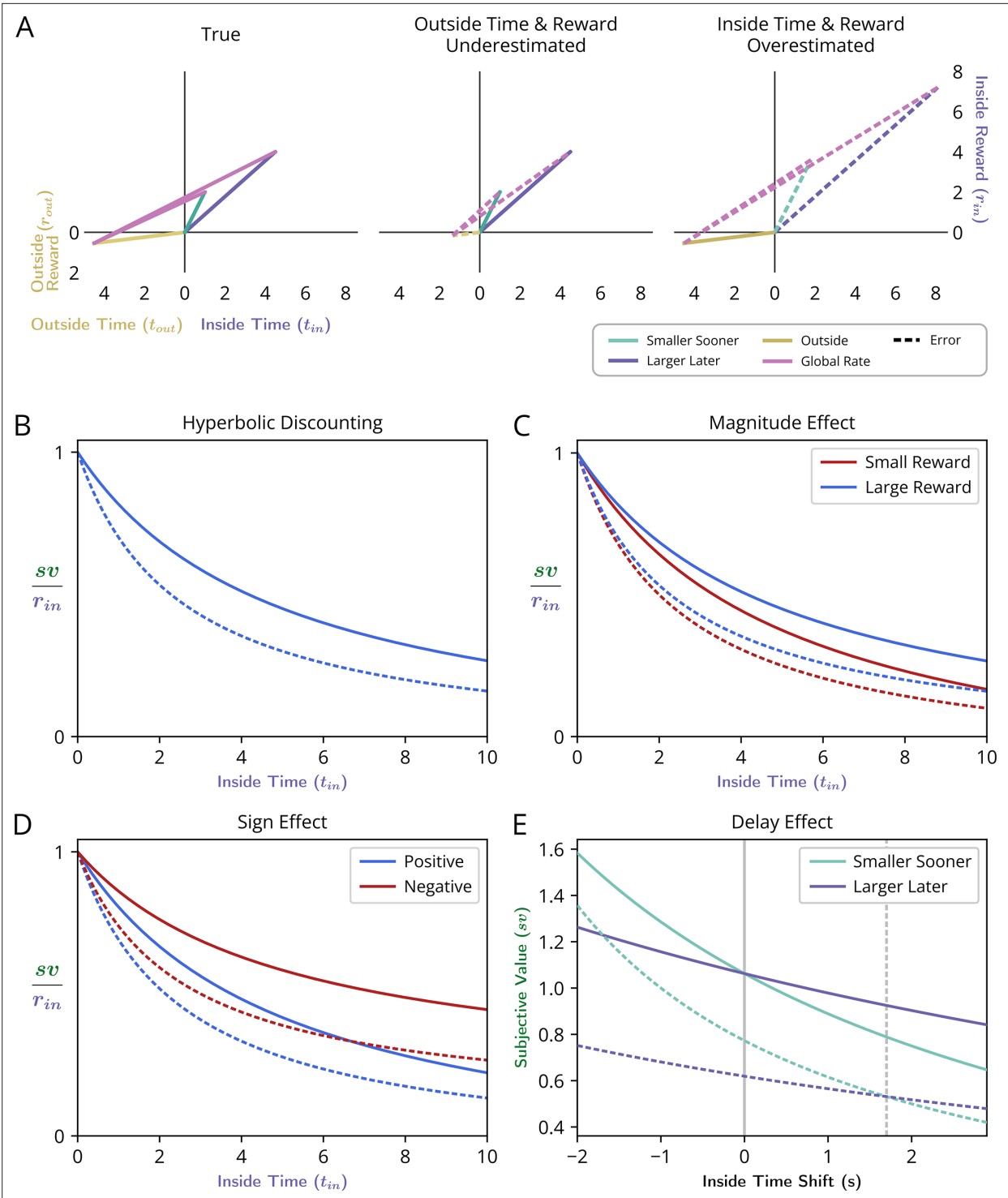

**Figure 21.** The Malapportionment Hypothesis. (**A-E**) Solid lines indicate true reward-rate maximizing values. Dashed lines indicate those of an agent described by the Malapportionment Hypothesis that underweights the apportionment of time outside relative to inside the considered pursuit type. The Malapportionment Hypothesis predicts the following: (**A**) *Suboptimal Choice, Optimal Forgo.* Suboptimally 'mpatient' decision-making, as revealed under Choice decision-making, arises in humans and animals as a consequence of the valuation process underweighting the contribution of accurately assessed pursuit reward rates outside versus inside the considered pursuit type. (Top Left) An example Choice situation where the global reward rate is maximized by choosing a larger later reward over a smaller-sooner reward. (Top Middle) An agent that underweights the outside time but accurately appreciates the outside and inside reward rates, overestimates the global reward rate resulting from each policy, and thus exhibits suboptimal impatience by selecting the smaller-sooner reward. $\omega$ = 0.3. (Top Right) Similarly, an agent that overweights the time inside the considered pursuit but

*Figure 21 continued on next page*

*Figure 21 continued*

accurately appreciates the outside and inside reward rates also overestimates the global reward rate and selects the smaller-sooner reward. As inside and outside reward rates are accurately assessed, forgo decisions can correctly be made despite any misappreciation of the relative time spent in/outside the considered pursuit. $\omega$ = 1.8. (**B**) *Hyperbolic discounting with greater curvature.* The Malapportionment Hypothesis predicts that humans and animals exhibit temporal discounting with greater curvature than a reward-rate maximizing agent. $\omega$ = 0.5, $r_{in}$ = 5, $r_{out}$ = 0.5, $t_{out}$ = 5. (**C**) *Less pronounced Magnitude Effect.* The Malapportionment Hypothesis predicts that the difference in apparent discounting related to the magnitude of the pursuit will be less pronounced than a reward-rate-maximizing agent. $\omega$ = 0.5, $r_{out}$ = 0.5, $t_{out}$ = 5, $r_{SS}$ = 2, $r_{LL}$ = 5. (**D**) *Less pronounced Sign Effect.* The Sign Effect is also predicted to be less pronounced than a reward-rate-maximizing agent. $\omega$ = 0.5, $r_{out}$ = 0.5, $t_{out}$ = 5, positive reward = 3, negative reward (punishment) = –3. (**E**) *More pronounced Delay Effect.* The Malapportionment Hypothesis predicts that the Delay Effect will be more pronounced than that exhibited by a reward-rate-maximizing agent. $\omega$ = 0.5, $r_{out}$ = 0.5, $t_{out}$ = 5, $r_{SS}$ = 2, $t_{SS}$ = 3, $r_{LL}$ = 5, $t_{LL}$ = 12.6.

apparent discounting functions will present with greater curvature than what a reward-rate-maximizing agent would exhibit (*Figure 21B*). While experimentally observed temporal discounting would have greater curvature, the Malapportionment Hypothesis also predicts that the Magnitude (*Figure 21C*) and Sign Effect (*Figure 21D*) would be less pronounced than what a reward-rate-maximizing agent would exhibit, with these effects becoming less pronounced the greater the underweighting. Finally, with regards to the Delay Effect (*Figure 21E*), the Malapportionment Hypothesis predicts that preference reversal would occur at delays greater than that exhibited by a reward-rate-maximizing agent, with the delay becoming more pronounced the greater the underweighting outside versus inside the considered pursuit by the agent.

## Comparisons to prior models

As our description of global reward rate-optimizing valuation is motivated by the same normative principle, how is our formalism unique from OFT, and, more generally, from other models proposing some form of reward-rate maximization? Firstly, the specific formulation proponents of OFT have used fails to adequately recognize how outside rewards influence the value of considered pursuits. Additionally, the relationship between time's cost and apparent temporal discounting has not been explicitly identified in prior OFT explanations. By contrast, our formulation, because of its specificity, can potentially align with neural representations of the variables we propose, and their misestimations may explain the ways in which observed animal behavior may deviate from optimality. Models inspired by OFT's objective of global reward-rate maximization but that seek to make a better accounting of observed deviations make the concession that, while global reward-rate maximization is sought, it is not achieved. Rather, some *non*-global reward-rate maximization is obtained by the agent (*Bateson and Kacelnik, 1996*; *Blanchard et al., 2013*; *Namboodiri et al., 2014a*; *Fung et al., 2021*). Of particular interest, the Heuristic model (*Blanchard et al., 2013*) and the TIMERR model (*Namboodiri et al., 2014c*) both assume non-global reward-rate maximization.

### Heuristic model

In the 'Heuristic' model (*Blanchard et al., 2013*), as in Ecological Rationality Theory, ERT (*Stephens et al., 2004*), it is thought that animals prioritize the local reward rate of considered pursuits, rather than the global reward rate. In the Heuristic model, however, suboptimal 'impatience' is rationalized as being the consequence of the animal's inability to fully appreciate post-reward delays (time subsequent to reward until re-entry into states/pursuits common to one or another policy). Indeed, while animals are demonstrated to be sensitive to post-reward delays, they act as if they significantly underestimate post-reward delays incurred, exhibiting a suboptimal bias for SS pursuits when LL pursuits would maximize global reward rate (*Blanchard et al., 2013*). Through a parameter, $\omega$, which adjusts the degree in which post-reinforcer delays are underestimated, the Heuristic model can be sufficient to capture observed animal behavior in intertemporal choice tasks that have been assessed (*Blanchard et al., 2013*). However, as the Heuristic model is quite specific as to the source of error—the underestimation post-reward delays—it would well fit observed behavior only in certain experimental conditions. Should appreciable (1) reward be obtained or (2) time be spent outside of a considered pursuit type and its post-reward interval, then the Heuristic model would fail to make a good accounting of observed behavior.

The Heuristic model can be modified to specify the uniform downscaling of all *non*-pursuit intervals (rather than just post-reward delays), as in the implementation by *Carter and Redish, 2016*.

This modification would bring the Heuristic model closer into alignment with the Malapportionment Hypothesis. But, as temporal underestimation would not apply to pursuits occurring outside the currently considered one, fits to observed behavior would be strained in worlds composed predominantly of pursuits with little non-pursuit time. Further, by underestimating the time spent outside the considered pursuit without a corresponding underestimation of reward earned outside the considered pursuit, the Heuristic model ought to overestimate the outside reward rate and thus the global reward rate.

So, while impatience under Choice could be fit under some experimental circumstances, behavior under Forgo instances would then be expected to also be sub-optimally impatient. Therefore, to bring the Heuristic model fully into alignment with the Malapportionment Hypothesis, it must be further assumed that the reward rate from the considered pursuit can be compared to the true outside or true global reward rate of the environment (as assumed in *Carter and Redish, 2016*), *as well as* expanded to underestimate *all* intervals of time occurring outside a considered pursuit type.

### TIMERR model

The essential feature of the TIMERR model (*Namboodiri et al., 2014b*; *Shuler and Namboodiri, 2018*) is that the agent looks back into its near past to estimate the reward rate of the environment, with this 'look-back' time, $T_{ime}$, being the model's free-fit parameter. In contrast to the reward-rate-optimal agent, this look-back time, then, is not a basic feature of the external world, but rather is related to how the animal uses its experience. TIMERR's policy is then determined by the reward rate obtained across this interval and that of the considered pursuit. In this way, TIMERR includes sources outside of the considered pursuit type in its evaluation, and because of this, exhibits many of the behaviors that the reward-rate-optimal agent is demonstrated here to express (*Blanchard et al., 2013*). Indeed, the TIMERR model and the optimal agent share the same mathematical form, though, critically, the meaning of their terms differs. An important additional difference is that IMERR is specific in the manner in which reward obtained outside the current instance of the considered pursuit is used: as recently experienced rewards from the past contribute to the estimation of the average reward rate of the environment, this 'look-back' time can include rewards from the pursuit type currently under consideration. Therefore, TIMERR commits an overestimation of the outside reward rate, and thus, an overestimation of global reward rate, manifesting as suboptimal impatience in Choice *and* Forgo decisions. In this way, while TIMERR is appealing in assuming that the recent past is used to estimate the global reward rate, and reproduces a number of sensitivities to conditions observed behaviorally, it is not in accordance with the Malapportionment Hypothesis as it mistakes pursuits' rates as well as their weights—something is still amiss.

## Conclusion

An enriched understanding of how a reward-rate-optimal agent evaluates temporal decision-making empowers insight into the nature of human and animal valuation. It does so not by advancing the claim that we are optimal, but rather by clarifying what are and are not signs of optimality, which then permits quantification of the intriguing pattern of adherence and deviation from this normative expectation. Therein lies clues for deducing the learning algorithm and representational architecture used by brains to attribute value to representations of the temporal structure of the world. Here we have conceptualized and generalized temporal decision-making worlds as composed of pursuits, described by their rates and weights, and in so doing, come to better appreciate the cost of time, how policies impact the reward rates reaped from those worlds, and how processes that fail to accurately appreciate those features would misvalue the worth of initiating pursuits. We propose the Malapportionment Hypothesis, which identifies a failure to accurately appreciate the weights rather than the rates of pursuits, as the root cause of errors made, to reckon with the curious pattern of behavior observed regarding whether to initiate a pursuit. We postulate that the value learning algorithm and representational architecture selected for by evolution has favored the ability to appreciate the reward rates of pursuits over that of their weights.

## Acknowledgements

We thank Roman Galperin, Vijay Namboodiri, David Linden, Chris Fetsch, and members of the Hussain Shuler lab for their insightful comments and input.

# Additional information

### Competing interests
Tanya Marton: Employee of Microsoft. The other authors declare that no competing interests exist.

### Funding

| Funder | Grant reference number | Author |
| --- | --- | --- |
| National Institute of Mental Health | 5R01MH123446 | Marshall G Hussain Shuler |
| National Institute on Aging | RF1AG063783 | Marshall G Hussain Shuler |
| National Eye Institute | T32EY007143 | Elissa Sutlief<br>Tanya Marton |
| Kavli Neuroscience Discovery Institute | | Charlie Walters<br>Marshall G Hussain Shuler |

The funders had no role in study design, data collection and interpretation, or the decision to submit the work for publication.

### Author contributions
Elissa Sutlief, Conceptualization, Resources, Software, Formal analysis, Validation, Investigation, Visualization, Methodology, Writing – review and editing; Charlie Walters, Conceptualization, Formal analysis, Validation, Investigation, Visualization, Methodology, Writing – review and editing, Software; Tanya Marton, Conceptualization, Resources, Formal analysis, Validation, Investigation, Visualization, Methodology, Writing – original draft, Writing – review and editing; Marshall G Hussain Shuler, Conceptualization, Resources, Data curation, Software, Formal analysis, Supervision, Funding acquisition, Validation, Investigation, Visualization, Methodology, Writing – original draft, Project administration, Writing – review and editing

### Author ORCIDs
Marshall G Hussain Shuler https://orcid.org/0000-0002-1927-0970

Reviewer #2 (Public review): https://doi.org/10.7554/eLife.99957.3.sa1
Reviewer #3 (Public review): https://doi.org/10.7554/eLife.99957.3.sa2
Author response https://doi.org/10.7554/eLife.99957.3.sa3

---

# Additional files

### Supplementary files
MDAR checklist

### Data availability
The manuscript is a theoretical study, so no data have been generated for this manuscript.

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

## Appendix 1

### Derivation of equation for global reward rate under multiple pursuits

$E(r)$ : the expected reward magnitude for each reward opportunity

$E(t)$ : the expected time between the initiation of reward pursuits

$\rho_g = \frac{E(r)}{E(t)}$ global reward rate: the average reward per pursuit divided by the average time per pursuit.

$\rho_d$: the average rate of collecting rewards while in the default pursuit

$p_i$ : reward opportunities $i$ as a proportion of total pursued rewards

$$p_i = \frac{f_i}{\sum_{j=1}^{n} f_j}$$

$E(r_{pursuit}) = \sum_{i=1}^{n} p_i r_i$ the average reward received per reward opportunity

$$E(r_{pursuit}) = \sum_{i=1}^{n} \frac{f_i r_i}{\sum_{j=1}^{n} f_j} = \frac{\sum_{i=1}^{n} f_i r_i}{\sum_{i=1}^{n} f_i}$$

$E(t_{pursuit}) = \sum_{i=1}^{n} p_i t_i$ the average time invested per reward opportunity

$$E(t_{pursuit}) = \sum_{i=1}^{n} \frac{f_i t_i}{\sum_{j=1}^{n} f_j} = \frac{\sum_{i=1}^{n} f_i t_i}{\sum_{i=1}^{n} f_i}$$

$E(t_{default}) = \frac{1}{\sum_{i=1}^{n} f_i}$: the average time spent in the default pursuit between reward opportunities

$E(r_{default}) = \rho_d \frac{1}{\sum_{i=1}^{n} f_i}$: the average reward received in the default pursuit between reward opportunities

$\rho_g = \frac{E(r_{pursuit}) + E(r_{default})}{E(t_{pursuit}) + E(t_{default})}$ the global reward rate of the reward opportunity landscape

$$\rho_g = \frac{\frac{\sum_{i=1}^{n} f_i r_i}{\sum_{i=1}^{n} f_i} + \frac{\rho_d}{\sum_{i=1}^{n} f_i}}{\frac{\sum_{i=1}^{n} f_i t_i}{\sum_{i=1}^{n} f_i} + \frac{1}{\sum_{i=1}^{n} f_i}}$$

$$\rho_g = \frac{\sum_{i=1}^{n} f_i r_i + \rho_d}{\sum_{i=1}^{n} f_i t_i + 1} \text{(Equation 1 in Main Text)}$$

Thus, *Equation 1* is formulated to calculate the average reward received and average time spent per unit time spent in the default pursuit. So, $f_i$ is the encounter rate of pursuit $i$ for one unit of time spent in the default pursuit. Added to the summation in the numerator, we have the average reward obtained in the default pursuit per unit time ($\rho_d * 1$) and in the denominator we have the time spent in the default pursuit per unit time (1).

## Appendix 2

### Average time spent outside $t_{out}$ the considered pursuit type, *in*, and the average reward rate earned outside that pursuit type $\rho_{out}$,

In order to simplify representations of policies governing any given pursuit opportunity, the expression for global reward rate, $\rho_g$, can be reformulated from the perspective of a policy of accepting any given pursuit.

$$\rho_{\forall i} = \frac{f_{in} r_{in} \sum_{i \neq in} f_i r_i + \rho_d}{f_{in} t_{in} \sum_{i \neq in} f_i t_i + 1}$$

$$\rho_g = \frac{r_{in} + \left( \sum_{\neq in} f_i r_i + \rho_d \right) / f_{in}}{t_{in} + \left( \sum_{i \neq in} f_i t_i + 1 \right) / f_{in}} \text{(Equation 2 in Main Text)}$$

$$t_{out} = t_{\forall i \neq in} = E(t_{invested, \forall i \neq in}) + E(t_{avail, \forall i \neq in})$$

$$t_{out} = t_{\forall i \neq in} = \left( \sum_{i \neq in} f_i t_i + 1 \right) / f_{in} \text{(Equation 3 in Main Text)}$$

$f_i$ is the probability that pursuit *i* will be encountered during a single unit of time spent in the default pursuit. The numerator of the expression is the average amount of time spent across all pursuits, excepting the considered pursuit, per unit time spent in the default pursuit. Note that the +1 in the numerator is accounting for the unit of time spent in the default pursuit and is added outside of the sum. Since $f_{in}$ is the probability that the considered pursuit will be encountered per unit of time spent in the default pursuit, $\frac{1}{f_{in}}$ is the average amount of time spent in the default pursuit between encounters of the considered pursuit. By multiplying the average time spent across all outside pursuits per unit of time in the default pursuit by the average amount of time spent in the default pursuit between encounters of the considered pursuit, we get the average amount of time spent outside the considered pursuit per encounter of the considered pursuit. This is calculated as if the pursuit encounters are mutually exclusive within a single unit of time spent within the default pursuit, as this is the case as the length of our unit time (delta t) approaches zero.

$$\rho_{out} = \rho_{\forall i \neq in} = \frac{\sum_{i \neq in} f_i r_i + \rho_d}{\sum_{i \neq in} f_i t_i + 1}$$

$\rho_{out}$ is the reward rate achieved from all the time spent outside the considered pursuit,, which is also the reward rate achieved if the considered pursuit,, is never pursued.

## Appendix 3

### Reformulation of global reward rate in terms of $\rho_{out}$ and $t_{out}$

Parceling a pursuit world into 'inside' the considered pursuit type and everything 'outside' the considered pursuit type, gives a generalized form for the reward rate of an environment under a given policy.

$$\rho_{\forall i} = \frac{r_{in} + (\sum_{i \neq in} f_i r_i + \rho_d)/f_{in}}{t_{in} + (\sum_{i \neq in} f_i t_i + 1)/f_{in}}$$

$$r_{out} = (\sum_{i \neq in} f_i r_i + \rho_d)/f_{in} \text{ (Equation 4 in Main Text)}$$

$$t_{out} = (\sum_{i \neq in} f_i t_i + 1)/f_{in}$$

$$\rho_g = \rho_{\forall i} = \frac{r_{in} + r_{out}}{t_{in} + t_{out}} \text{ (Equation 5 in Main Text)}$$

## Appendix 4

### Global reward rate is a weighted average of an option's reward rate and its outside reward rate

$$\rho_g = \frac{r_{in} + \rho_{out} t_{out}}{t_{in} + t_{out}}$$

$$\rho_g = \frac{r_{in}}{t_{in}} \frac{t_{in}}{t_{in} + t_{out}} + \rho_{out} \frac{t_{out}}{t_{in} + t_{out}}$$

$$Let\ w = \frac{t_{in}}{t_{in} + t_{out}}$$

$$\rho_g = w \cdot \rho_{in} + (1 - w) \cdot \rho_{out} \text{ (Equation 6 in Main Text)}$$

## Appendix 5

### Derivation of reward-rate-maximizing forgo policies

Forgo the considered pursuit if $\rho_{\forall i} < \rho_{\forall i \neq in}$

$$\rho_{\forall i} < \rho_{\forall i \neq in}$$

$$\frac{r_{in} + (\sum\limits_{i \neq in} f_i r_i + \rho_d)/f_{in}}{t_{in} + (\sum\limits_{i \neq in} f_i t_i + 1)/f_{in}} < \frac{(\sum\limits_{i \neq in} f_i r_i + \rho_d)/f_{in}}{(\sum\limits_{i \neq in} f_i t_i + 1)/f_{in}}$$

$$\frac{r_{in} + r_{out}}{t_{in} + t_{out}} < \rho_{out}$$

$$r_{in} + r_{out} < \rho_{out} t_{in} + \rho_{out} t_{out}$$

$$r_{in} + r_{out} < \rho_{out} t_{in} + r_{out}$$

$$r_{in} < \rho_{out} t_{in}$$

$$\frac{r_{in}}{t_{in}} < \rho_{out}$$

$$\rho_{in} < \rho_{out}$$

$$\rho_{out} = \rho_g^*$$

$$\rho_{in} < \rho_g^*$$

$$\rho_{\forall i} < \rho_{\forall i \neq in} \leftrightarrow \rho_{in} < \rho_{out} \leftrightarrow \rho_{in} < \rho_g^*$$

Choose considered pursuit if $\rho_{\forall i} > \rho_{\forall i \neq in}$

$$\rho_{\forall i} > \rho_{\forall i \neq in}$$

$$\frac{r_{in} + (\sum\limits_{i \neq in} f_i r_i + \rho_d)/f_{in}}{t_{in} + (\sum\limits_{i \neq in} f_i t_i + 1)/f_{in}} > \frac{(\sum\limits_{i \neq in} f_i r_i + \rho_d)/f_{in}}{(\sum\limits_{i \neq in} f_i t_i + 1)/f_{in}}$$

$$\frac{r_{in} + r_{out}}{t_{in} + t_{out}} > \rho_{out}$$

$$r_{in} + r_{out} > \rho_{out} t_{in} + \rho_{out} t_{out}$$

$$r_{in} + r_{out} > \rho_{out} t_{in} + r_{out}$$

$$r_{in} > \rho_{out} t_{in}$$

$$\frac{r_{in}}{t_{in}} > \rho_{out}$$

$$\rho_{in} > \rho_{out}$$

$$\frac{r_{in}}{t_{in}} \frac{t_{out}}{t_{in} + t_{out}} > \rho_{out} \frac{t_{out}}{t_{in} + t_{out}}$$

$$\frac{r_{in}}{t_{in}}\frac{t_{out}}{t_{in}+t_{out}} + \frac{r_{in}}{t_{in}+t_{out}} > \frac{r_{in}+\rho_{out}t_{out}}{t_{in}+t_{out}}$$

$$\frac{r_{in}}{t_{in}}\frac{t_{out}}{t_{in}+t_{out}} + \frac{r_{in}}{t_{in}}\frac{t_{in}}{t_{in}+t_{out}} > \frac{r_{in}+\rho_{out}t_{out}}{t_{in}+t_{out}}$$

$$\frac{r_k}{t_k} > \rho_g^*$$

$$\rho_{\forall i} > \rho_{\forall i \neq in \leftrightarrow \rho_{in} > \rho_g^*}$$

$$\rho_{\forall i} > \rho_{\forall i \neq in \leftrightarrow \rho_{in} > \rho_{out} \leftrightarrow \rho_{in} > \rho_g^*}$$

Choosing and forgoing the considered option are equivalent if

$$\rho_{\forall i} = \rho_{\forall i \neq in}$$

$$\frac{r_{in} + (\sum\limits_{i \neq in} f_i r_i + \rho_d)/f_{in}}{t_{in} + (\sum\limits_{i \neq in} f_i t_i + 1)/f_{in}} = \frac{(\sum\limits_{i \neq in} f_i r_i + \rho_d)/f_{in}}{(\sum\limits_{i \neq in} f_i t_i + 1)/f_{in}}$$

$$\frac{r_{in} + r_{out}}{t_{in} + t_{out}} = \rho_{out}$$

$$r_{in} + r_{out} = \rho_{out}t_{in} + \rho_{out}t_{out}$$

$$r_{in} + r_{out} = \rho_{out}t_{in} + r_{out}$$

$$r_{in} = \rho_{out}t_{in}$$

$$\frac{r_{in}}{t_{in}} = \rho_{out}$$

$$\rho_{in} = \rho_{out}$$

$$\frac{r_{in}}{t_{in}}\frac{t_{out}}{t_{in}+t_{out}} = \rho_{out}\frac{t_{out}}{t_{in}+t_{out}}$$

$$\frac{r_{in}}{t_{in}}\frac{t_{out}}{t_{in}+t_{out}} + \frac{r_{in}}{t_{in}+t_{out}} = \frac{r_{in}+\rho_{out}t_{out}}{t_k+t_{out}}$$

$$\frac{r_{in}}{t_{in}}\frac{t_{out}}{t_{in}+t_{out}} + \frac{r_{in}}{t_{in}}\frac{t_{in}}{t_{in}+t_{out}} = \frac{r_{in}+\rho_{out}t_{out}}{t_{in}+t_{out}}$$

$$\frac{r_{in}}{t_{in}} = \rho_g^*$$

$$\rho_{in} = \rho_g^*$$

$$\rho_{\forall i} = \rho_{\forall i \neq in \leftrightarrow \rho_{in} = \rho_{out} \leftrightarrow \rho_{in} = \rho_g^*}$$

## Appendix 6

### Derivation of the equivalent immediate reward (i.e. the subjective value) for optimal global reward rate

Pursuit $in1$ and pursuit $in2$ produce the equivalent global reward rate if $\frac{r_{in1}-r_{in2}}{t_{in1}-t_{in2}} = \frac{r_{in2}+\rho_{out}t_{out}}{t_{in2}+t_{out}} = \frac{r_{in1}+\rho_{out}t_{out}}{t_{in1}+t_{out}}$

By definition, if $t_{in2} = 0$, pursuit $in2$ is an immediate reward. Finding $r_{in2}$ such that $\frac{r_{in1}-r_{in2}}{t_{in1}-t_{in2}} = \frac{r_{in2}+\rho_{out}t_{out}}{t_{in2}+t_{out}} = \frac{r_{in1}+\rho_{out}t_{out}}{t_{in1}+t_{out}}$ describes the equivalent immediate subjective value of pursuit $in1$

$$\text{If } \frac{r_{in2} + \rho_{out}t_{out}}{t_{out}} = \frac{r_{in1} + \rho_{out}t_{out}}{t_{in1} + t_{out}}, sv_{in1} = r_{in2},$$

$$\frac{r_{in1} - sv_{in1}}{t_{in1}} = \frac{sv_{in1} + \rho_{ou}t_{out}}{t_{out}} = \frac{r_{in} + \rho_{out}t_{out}}{t_{in1} + t_{out}} = \rho_g$$

$$\frac{r_{in1} - sv_{in1}}{t_{in1}} = \rho_g$$

$$r_{in1} - sv_{in1} = \rho_g t_{in1}$$

$$sv_{in1} = r_{in1} - \rho_g t_{in1}$$

Therefore, for a considered pursuit, $in$,

$$sv = r_{in} - \rho_g t_{in} \text{ (Equation 7 in Main Text)}$$

## Appendix 7

### Equivalent immediate subjective value need not be calculated from option-specific estimations of global reward rate

if $sv_{in1} < sv_{in2}$

$$sv_{in1} < sv_{in2}$$

$$r_{in1} - \rho_g\left(in1\right)t_{in1} < r_{in2} - \rho_g\left(in2\right)t_{in2}$$

$$\rho_g\left(in1\right) = \frac{r_{in1} + \rho_{out}t_{out}}{t_{in1} + t_{out}}$$

$$\rho_g\left(in1\right) < \rho_g\left(in2\right) = \rho_g^*$$

$$\rho_g\left(in1\right)t_{in1} < \rho_g\left(in2\right)t_{in1}$$

$$r_{in1} - \rho_g\left(in2\right)t_{in1} < r_{in1} - \rho_g\left(in1\right)t_{1in}$$

$$r_{in1} - \rho_g\left(in2\right)t_{in1} < r_{in1} - \rho_g\left(in1\right)t_{in1} < r_{in2} - \rho_g\left(in2\right)t_{in2}$$

$$r_{in1} - \rho_g^*t_{in1} < r_{in2} - \rho_g^*t_{in2}$$

$$sv_{in1}^* < sv_k^*$$

$$sv_{in1} < sv_{in2} \leftrightarrow sv_{in1}^* < sv_{in2}^*$$

If $sv_{in1} = sv_{in2}$

$$sv_{in1} = sv_{in2}$$

$$r_{in1} - \rho_g\left(in1\right)t_{in1} = r_{in2} - \rho_g\left(in2\right)t_{in2}$$

$$\rho_g\left(in1\right) = \frac{r_{in1} + \rho_{out}t_{out}}{t_{in1} + t_{out}}$$

$$\rho_g\left(in1\right) = \rho_g\left(in2\right) = \rho_g^*$$

$$r_{in1} - \rho_g\left(in2\right)t_{in1} = r_{in1} - \rho_g\left(in1\right)t_{in1}$$

$$r_{in1} - \rho_g^*t_{in1} = r_{in2} - \rho_g^*t_{in2}$$

$$sv_{in1}^* = sv_{in2}^*$$

$$sv_{in1} = sv_{in2} \leftrightarrow sv_{in1}^* = sv_{in2}^*$$

If $\rho_{in} = \rho_{out}$

$$sv_{in1} > sv_{in2}$$

$$r_{in1} - \rho_g\left(in1\right)t_{in1} > r_{in2} - \rho_g\left(in2\right)t_{in2}$$

$$\rho_g\left(in1\right) = \frac{r_{in1} + \rho_{out}t_{out}}{t_{in1} + t_{out}}$$

$$\rho_g^* = \rho_g\left(in1\right) > \rho_g\left(in2\right)$$

$$\rho_g \left(in1\right) t_{in2} > \rho_g \left(in2\right) t_{in2}$$

$$r_{in2} - \rho_g \left(in2\right) t_{in2} > r_{in2} - \rho_g \left(in1\right) t_{in2}$$

$$r_{in1} - \rho_g \left(in1\right) t_{in1} > r_{in2} - \rho_g \left(in1\right) t_{in2}$$

$$r_{in1} - \rho_g^* t_{in1} > r_{in2} - \rho_g^* t_{in2}$$

$$sv_{in1}^* > sv_{in2}^*$$

$$sv_{in1} > sv_{in2} \leftrightarrow sv_{in1}^* > sv_{in2}^*$$

If $\rho_{in} < \rho_{out}$

$$\rho_{in} < \rho_{out}$$

$$\rho_g^* = \rho_{out}$$

$$\rho_{in} < \rho_g^* = \rho_{out}$$

$$\rho_{in} < \rho_g^*, \rho_{in} < \rho_{out}$$

$$r_{in} - \rho_g^* t_{in} < 0, r_{in} - \rho_{out} t_{in} < 0$$

$$sv_{in}^* < 0, sv_{in} < 0,$$

If $\rho_{in} = \rho_{out}$

$$\rho_{in} = \rho_{out}$$

$$\rho_g^* = \rho_{in} = \rho_{out}$$

$$\rho_{in} = \rho_g^* = \rho_{out}$$

$$\rho_{in} = \rho_g^*, \rho_{in} = \rho_{out}$$

$$r_{in} - \rho_g^* t_{in} = 0, r_{in} - \rho_{out} t_{in} = 0$$

$$sv_{in}^* = 0, sv_{in} = 0$$

If $\rho_{in} > \rho_{out}$

$$\rho_{in} > \rho_{out}$$

$$\rho_g^* > \rho_{out}$$

$$\rho_{in} > \rho_g^* > \rho_{out}$$

$$\rho_{in} > \rho_g^*, \rho_{in} > \rho_{out}$$

$$r_{in} - \rho_g^* t_{in} > 0, r_{in} - \rho_{out} t_{in} > 0$$

$$sv_{in}^* > 0, sv_{in} > 0$$

## Appendix 8

### Reformulation of equivalent immediate subjective value in terms of outside parameters

$$sv_{in} = r_{in} - \rho g t_{in}$$

$$\rho g = \frac{r_{in} + \frac{r_{out}}{t_{out}} t_{out}}{t_{in} + t_{out}}$$

$$sv_{in} = r_{in} - \frac{r_{in} + \frac{r_{out}}{t_{out}} t_{out}}{t_{in} + t_{out}} t_{in}$$

$$sv_{in} = r_{in} \frac{t_{in} + t_{out}}{t_{in} + t_{out}} - \frac{r_{in}t_{in} + \left(\frac{r_{out}}{t_{out}} t_{out}\right) t_{in}}{t_{in} + t_{out}}$$

$$sv_{in} = \frac{r_{in}t_{out} - \frac{r_{out}}{t_{out}} t_{in}t_{out}}{t_{in} + t_{out}}$$

$$sv_{in} = \left(r_{in} - \frac{r_{out}}{t_{out}} t_{in}\right) \left(\frac{t_{out}}{t_{in} + t_{out}}\right) \text{ (Equation 8 in Main Text)}$$

The above expression of subjective value is arranged to emphasize an understanding of value in terms of an opportunity cost subtraction (the left parenthetical clause) and an apportionment scaling (the right parenthetical clause). Alternatively, the expression of subjective value below is re-arranged so that an opportunity cost subtraction appears in the numerator and where the time of the pursuit appears in the denominator with a scaling factor, $\frac{1}{t_{out}}$, resembling the standard temporal discounting form.

$$sv_{in} = \frac{r_{in} - \frac{r_{out}}{t_{out}} t_{in}}{1 + \frac{1}{t_{out}} * t_{in}} \text{ (Equation, 10, RHS, in Main Text)}$$

## Appendix 9

### Derivation of choice policies that optimize global reward rate

Let $t_{in1} > t_{in2}$

Choose option $in1$ if $\rho_{1+\forall i \neq in1,in2} > \rho_{2j+\forall i \neq in1,in2}$

$$\rho_{1+\forall i \neq in1,in2} > \rho_{2+\forall i \neq in1,in2}$$

$$\frac{r_{in1} + \left(\sum_{i \neq in1,in2} f_i r_i + \rho_d\right) /f_{in1,in2}}{t_{in} + \left(\sum_{i \neq in1,in2} f_i t_i + 1\right) /f_{in1}} > \frac{r_{in2} + \left(\sum_{i \neq in1,in2} f_i r_i + \rho_d\right) /f_{in1,in2}}{t_{in2} + \left(\sum_{i \neq in1,in2} f_i t_i + 1\right) /f_{in1,in2}}$$

$f_{in1,in2}$: the frequency at which the choice between option $in1$ and $in2$ are presented.
$\rho_{out}$: the reward rate earned outside of the $in1$ v. $in2$ choice
$t_{out}$: the average time per choice spent outside of $in1$ or $in2$.

$$\frac{r_{in1} + \rho_{out} t_{out}}{t_{in1} + t_{out}} > \frac{r_{in2} + \rho_{out} t_{out}}{t_{in2} + t_{out}}$$

$$\left(r_{in1} + \rho_{out} t_{out}\right)\left(t_{in2} + t_{out}\right) > \left(r_{in2} + \rho_{out} t_{out}\right)\left(t_{in1} + t_{out}\right)$$

$$r_{in1} t_{in2} + r_{in1} t_{out} + \rho_{out} t_{out} t_{in2} + \rho_{out} t_{out}^2 > r_{in2} t_{in1} + r_{in2} t_{out} + \rho_{out} t_{out} t_{in1} + \rho_{out} t_{out}^2$$

$$r_{in1} t_{in2} + t_{out}\left(r_{in1} - r_{in2}\right) > r_{in2} t_{in1} + \rho_{out} t_{out}\left(t_{in1} - t_{in2}\right)$$

$$\left(r_{in1} - r_{in2}\right) t_{in2} + t_{out}\left(r_{in1} - r_{in2}\right) > r_{in2}\left(t_{in1} - t_{in2}\right) + \rho_{out} t_{out}\left(t_{in1} - t_{in2}\right)$$

$$\left(r_{in1} - r_{in2}\right)\left(t_{in2} + t_{out}\right) > \left(r_{in2} + \rho_{out} t_{out}\right)\left(t_{in1} - t_{in2}\right)$$

$$\frac{r_{in1} - r_{in2}}{t_{in1} - t_{in2}} > \frac{r_{in2} + \rho_{out} t_{out}}{t_{in2} + t_{out}}$$

$$\rho_{in1+\forall i \neq in1,in2}, \rho_{in2+\forall i \neq in1,in2}$$

$$\frac{r_{in1} + \rho_{out} t_{out}}{t_{in1} + t_{out}} = \frac{r_{in2} + \rho_{out} t_{out}}{t_{in2} + t_{out}} \frac{t_{in2} + t_{out}}{t_{in1} + t_{out}} + \frac{r_{in1} - r_{in2}}{t_{in1} - t_{in2}} \frac{t_{in1} - t_{in2}}{t_{in1} + t_{out}}$$

$$\frac{r_{in2} + \rho_{out} t_{out}}{t_{in2} + t_{out}} \frac{t_{in2} + t_{out}}{t_{in1} + t_{out}} = \frac{r_{in1} + \rho_{out} t_{out}}{t_{in1} + t_{out}} - \frac{r_{in1} - r_{in2}}{t_{in1} - t_{in2}} \frac{t_{in1} - t_{in2}}{t_{in1} + t_{out}}$$

$$\frac{r_{in2} + \rho_{out} t_{out}}{t_{in2} + t_{out}} = \frac{t_{in1} + t_{out}}{t_{in2} + t_{out}}\left(\frac{r_{in1} + \rho_{out} t_{out}}{t_{in1} + t_{out}} - \frac{r_{in1} - r_{in2}}{t_{in1} - t_{in2}} \frac{t_{in1} - t_{in2}}{t_{in1} + t_{out}}\right)$$

$$\frac{r_{in1} - r_{in2}}{t_{in1} - t_{in2}} > \frac{t_{in1} + t_{out}}{t_{in2j} + t_{out}}\left(\frac{r_{in1} + \rho_{out} t_{out}}{t_{in1} + t_{out}} - \frac{r_{in1} - r_{in2}}{t_{in1} - t_{in2}} \frac{t_{in1} - t_{in2}}{t_{in1} + t_{out}}\right)$$

$$\frac{r_{in1} - r_{in2}}{t_{in1} - t_{in2}}\left(1 + \frac{t_{in1} - t_{in2}}{t_{in2} + t_{out}}\right) > \frac{t_{in1} + t_{out}}{t_{in2} + t_{out}}\left(\frac{r_{in1} + \rho_{out} t_{out}}{t_{in1} + t_{out}}\right)$$

$$\frac{r_{in1} - r_{in2}}{t_{in1} - t_{in2}} \frac{t_{in1} + t_{out}}{t_{in2} + t_{out}} > \frac{t_{in1} + t_{out}}{t_{in2j} + t_{out}}\left(\frac{r_{in1} + \rho_{out} t_{out}}{t_{in1} + t_{out}}\right)$$

$$\frac{r_{in1} - r_{in2}}{t_{in1} - t_{in2}} > \frac{r_{in1} + \rho_{out}t_{out}}{t_{in1} + t_{out}}$$

$$\frac{r_{in1} - r_{in2}}{t_{in1} - t_{in2}} > \frac{r_{in1} + \rho_{out}t_{out}}{t_{in1} + t_{out}} > \frac{r_{in2} + \rho_{out}t_{out}}{t_{in2} + t_{out}}$$

$\rho_g^*$ : the maximum reward rate

If $\frac{r_{in1} + \rho_{out}t_{out}}{t_{in1} + t_{out}} > \frac{r_{in2} + \rho_{out}t_{out}}{t_{in2} + t_{out}}, \rho_g^* = \frac{r_{in1} + \rho_{out}t_{out}}{t_{in1} + t_{out}}$ ,

$$\frac{r_{in1} - r_{in2}}{t_{in1} - t_{in2}} > \rho_g^*$$

Choose option $in2$ if $\rho_{in1 + \forall i \neq in1, in2} < \rho_{in2 + \forall i \neq in1, in2}$

If $\frac{r_{in1} + \rho_{out}t_{out}}{t_{in1} + t_{out}} < \frac{r_{in2} + \rho_{out}t_{out}}{t_{in2} + t_{out}}, \rho_g^* = \frac{r_{in2} + \rho_{out}t_{out}}{t_{in1} + t_{out}}$

$$\frac{r_{in1} - r_{in2}}{t_{in1} - t_{in2}} < \frac{r_{in2} + \rho_{out}t_{out}}{t_{in2} + t_{out}}$$

$$\frac{r_{in1} - r_{in2}}{t_{in1} - t_{in2}} < \rho_g^*$$

Option $in2$ and option $in1$ are equivalent if $\rho_{1 + \forall i \neq in1, in2} = \rho_{2 + \forall i \neq in1, in2}$

If $\frac{r_{in1} + \rho_{out}t_{out}}{t_{in1} + t_{out}} = \frac{r_{in2} + \rho_{out}t_{out}}{t_{in2} + t_{out}}, \rho_g^* = \frac{r_{in2} + \rho_{out}t_{out}}{t_{in2} + t_{out}}$

$$\frac{r_{in1} - r_{in2}}{t_{in1} - t_{in2}} = \frac{r_{in2} + \rho_{out}t_{out}}{t_{in} + t_{out}} = \frac{r_{in1} + \rho_{out}t_{out}}{t_{in1} + t_{out}} = \rho_g^*$$

$$\frac{r_{in1} - r_{in2}}{t_{in1} - t_{in2}} = \rho_g^*$$

## Appendix 10

### Equivalent immediate subjective value policies that optimize global reward rate

Choose option $in1$ over pursuit $in2$ if $\rho_g(in1) > \rho_g(in2)$

$$\rho_g(in1) > \rho_g(in2)$$

$$\frac{r_{in1} + \rho_{out}t_{out}}{t_{in1} + t_{out}} > \frac{r_{in2} + \rho_{out}t_{out}}{t_{in2} + t_{out}}$$

$$\frac{r_{in1} + \rho_{out}t_{out}}{t_{ink} + t_{out}} - \rho_{out} > \frac{r_{in2} + \rho_{out}t_{out}}{t_{in2} + t_{out}} - \rho_{out}$$

$$\frac{r_{in1} + \rho_{out}t_{out}}{t_{in1} + t_{out}} - \rho_{out}\frac{t_{in1} + t_{out}}{t_{in1} + t_{out}} > \frac{r_{in2} + \rho_{out}t_{out}}{t_{in2} + t_{out}} - \rho_{out}\frac{t_{in2} + t_{out}}{t_{in2} + t_{out}}$$

$$\frac{r_{in1} + \rho_{out}t_{out}}{t_{in1} + t_{out}} - \frac{\rho_{out}t_{in1} + \rho_{out}t_{out}}{t_{in1} + t_{out}} > \frac{r_{in2} + \rho_{out}t_{out}}{t_{in2} + t_{out}} - \frac{\rho_{out}t_{in2} + \rho_{out}t_{out}}{t_{in2} + t_{out}}$$

$$\frac{r_{in1} - \rho_{out}t_{in1}}{t_{in1} + t_{out}} > \frac{r_{in2} - \rho_{out}t_{in2}}{t_{in2} + t_{out}}$$

$$\frac{r_{in1} - \rho_{out}t_{in1}}{t_{in1}/t_{out} + 1} > \frac{r_{in2} - \rho_{out}t_{in2}}{t_{in2}/t_{out} + 1}$$

$$sv_{in1} > sv_{in2}$$

$$\rho_g(in1) > \rho_g(in2) \leftrightarrow sv_{in1} > sv_{in2}$$

Choose option $in2$ over option $in1$ if $\rho_g(in2) > \rho_g(in1)$

$$\rho_g(in2) > \rho_g(in1)$$

$$\frac{r_{in2} + \rho_{out}t_{out}}{t_{in2} + t_{out}} > \frac{r_{in1} + \rho_{out}t_{out}}{t_{in1} + t_{out}}$$

$$\frac{r_{in2} - \rho_{out}t_{in2}}{t_{in2}/t_{out} + 1} > \frac{r_{in1} - \rho_{out}t_{in1}}{t_{in1}/t_{out} + 1}$$

$$\rho_g(in2) > \rho_g(in1) \leftrightarrow sv_{in2} > sv_{in1}$$

Option $in2$ and option $in1$ are equivalent if $\rho_g(in1) = \rho_g(in2)$

$$\rho_g(in1) = \rho_g(in2)$$

$$\frac{r_{in1} + \rho_{out}t_{out}}{t_{in1} + t_{out}} = \frac{r_{in2} + \rho_{out}t_{out}}{t_{in2} + t_{out}}$$

$$\frac{r_{in1} - \rho_{out}t_{in1}}{t_{in1}/t_{out} + 1} = \frac{r_{in2} - \rho_{out}t_{in2}}{t_{in2}/t_{out} + 1}$$

$$sv_{in1} = sv_{in2}$$

$$\rho_g(in1) = \rho_g(in2) \leftrightarrow sv_{in1} = sv_{in2}$$

## Appendix 11

### Conditions wherein overestimation of global reward rate leads to suboptimal choice behavior

If $t_{LL} > t_{SS}$ and $\frac{r_{SS}}{t_{SS}} > \frac{r_{LL}}{t_{LL}}$ and $r_{LL} > r_{SS}$

$$\frac{r_{SS}}{t_{SS}} > \frac{r_{LL}}{t_{LL}}$$

$$r_{SS}t_{LL} > r_{LL}t_{SS}$$

$$r_{SS}t_{LL} - r_{SS}t_{SS} > r_{LL}t_{SS} - r_{SS}t_{SS}$$

$$r_{SS}\left(t_{LL} - t_{SS}\right) > \left(r_{LL} - r_{SS}\right)t_{SS}$$

$$\frac{r_{SS}}{t_{SS}} > \frac{r_{LL} - r_{SS}}{t_{LL} - t_{SS}}$$

$$r_{SS}t_{LL} > r_{LL}t_{SS}$$

$$r_{SS}t_{LL} + r_{LL}t_{LL} > r_{LL}t_{SS} + r_{LL}t_{LL}$$

$$r_{LL}t_{LL} - r_{LL}t_{SS} > r_{LL}t_{LL} - r_{SS}t_{LL}$$

$$r_{LL}\left(t_{LL} - t_{SS}\right) > \left(r_{LL} - r_{SS}\right)t_{LL}$$

$$r_{LL}\left(t_{LL} - t_{SS}\right) > \left(r_{LL} - r_{SS}\right)t_{LL}$$

$$\frac{r_{LL}}{t_{LL}} > \frac{r_{LL} - r_{SS}}{t_{LL} - t_{SS}}$$

$$\frac{r_{SS}}{t_{SS}} > \frac{r_{LL}}{t_{LL}} > \frac{r_{LL} - r_{SS}}{t_{LL} - t_{SS}}$$

$$\rho_{SS} > \rho_{LL} > \frac{r_{LL} - r_{SS}}{t_{LL} - t_{SS}}$$

$$\frac{r_{LL}}{t_{LL}} = \frac{r_{LL} - r_{SS}}{t_{LL} - t_{SS}}\frac{t_{LL} - t_{SS}}{t_{LL}} + \frac{r_{SS}}{t_{SS}}\frac{t_{SS}}{t_{LL}}$$

$$\rho_{LL} = \frac{r_{LL} - r_{SS}}{t_{LL} - t_{SS}}\frac{t_{LL} - t_{SS}}{t_{LL}} + \rho_{SS}\frac{t_{SS}}{t_{LL}}$$

$$\frac{r_{LL} - r_{SS}}{t_{LL} - t_{SS}} = \rho_{LL}\frac{t_{LL}}{t_{LL} - t_{SS}} - \rho_{SS}\frac{t_{SS}}{t_{LL} - t_{SS}}$$

pursuit LL is optimal if $\frac{r_{LL} - r_{SS}}{t_{LL} - t_{SS}} > \rho_g^*$ and $sv_{LL} > sv_{SS}$
 Policy from global reward rate overestimation
 $s\hat{v}_{LL} > s\hat{v}_{SS}$, the animal will choose pursuit LL

$$s\hat{v}_{LL} > s\hat{v}_{SS}$$

$$r_{LL} - \hat{\rho}_g t_{LL} > r_{SS} - \hat{\rho}_g t_{SS}$$

$$r_{LL} - r_{SS} > \hat{\rho}_g t_{LL} - \hat{\rho}_g t_{SS}$$

$$r_{LL} - r_{SS} > \hat{\rho}_g\left(t_{LL} - t_{SS}\right)$$

$$\frac{r_{LL} - r_{SS}}{t_{LL} - t_{SS}} > \hat{\rho}_g$$

$s\hat{v}_{LL} < s\hat{v}_{SS}$, the animal will choose pursuit SS

$$s\hat{v}_{LL} < s\hat{v}_{SS}$$

$$r_{LL} - \hat{\rho}_g t_{LL} < r_{SS} - \hat{\rho}_g t_{SS}$$

$$r_{LL} - r_{SS} < \hat{\rho}_g t_{LL} - \hat{\rho}_g t_{SS}$$

$$r_{LL} - r_{SS} < \hat{\rho}_g \left(t_{LL} - t_{SS}\right)$$

$$\frac{r_{LL} - r_{SS}}{t_{LL} - t_{SS}} < \hat{\rho}_g$$

pursuit is optimal if $\frac{r_{LL} - r_{SS}}{t_{LL} - t_{SS}} > \rho_g^*$ and pursuit is chosen if $\frac{r_{LL} - r_{SS}}{t_{LL} - t_{SS}} > \hat{\rho}_g$
pursuit is optimal if $\frac{r_{LL} - r_{SS}}{t_{LL} - t_{SS}} > \rho_g^*$ but pursuit SS is chosen if $\frac{r_{LL} - r_{SS}}{t_{LL} - t_{SS}} < \hat{\rho}_g$
The policy from overestimation is suboptimal if $\rho_g^* < \frac{r_{LL} - r_{SS}}{t_{LL} - t_{SS}} < \hat{\rho}_g$
The policy from overestimation is suboptimal if $s\hat{v}_{LL} < s\hat{v}_{SS}$ but $sv_{LL} > sv_{SS}$

$$s\hat{v}_{LL} = sv_{LL} - t_{LL} \left(\hat{\rho}_g - \rho_g^*\right)$$

$$s\hat{v}_{LL} < s\hat{v}_{SS}$$

$$sv_{LL} - t_{LL} \left(\hat{\rho}_g - \rho_g^*\right) < sv_{SS} - t_{SS} \left(\hat{\rho}_g - \rho_g^*\right)$$

$$sv_{LL} - sv_{SS} < \left(t_{LL} - t_{SS}\right) \left(\hat{\rho}_g - \rho_g^*\right)$$

$$sv_{LL} > sv_{SS} \rightarrow 0 < sv_{LL} - sv_{SS}$$

$$0 < sv_{LL} - sv_{SS} < \left(t_{LL} - t_{SS}\right) \left(\hat{\rho}_g - \rho_g^*\right)$$

$$0 < \frac{sv_{LL} - sv_{SS}}{t_{LL} - t_{SS}} < \hat{\rho}_g - \rho_g^*$$

$$\hat{\rho}_g^* = max \left(\hat{w}_{LL} \left(\rho_{LL} - \rho_{out}\right), \hat{w}_{SS} \left(\rho_{SS} - \rho_{out}\right)\right) + \rho_{out}$$

$$\hat{\rho}_g^* = \rho_g^* + \left(\hat{w}_{LL} - w_{LL}\right) \left(\rho_{LL} - \rho_{out}\right) \, or \, \hat{\rho}_g^* = \hat{\rho}_g^* + \left(\hat{w}_l - w_l\right) \left(l_l - \rho_{out}\right)$$

$$\hat{\rho}_g^* - \rho_g^* = \left(\hat{w}_{LL} - w_{LL}\right) \left(\rho_{LL} - \rho_{out}\right) \, or \, \hat{\rho}_g^* - \rho_g^* = \left(\hat{w}_l - w_l\right) \left(l_l - \rho_{out}\right)$$

$$0 < \frac{sv_{LL} - sv_{SS}}{t_{LL} - t_{SS}} < max \left(\left(\hat{w}_{LL} - w_{LL}\right) \left(l_{LL} - \rho_{out}\right), \left(\hat{w}_l - w_l\right) \left(l_l - \rho_{out}\right)\right)$$

$$0 < \frac{sv_{LL} - sv_{SS}}{t_{LL} - t_{SS}} < \left(\hat{w}_l - w_l\right) \left(l_l - \rho_{out}\right)$$

$$0 < \frac{sv_{LL} - sv_{SS}}{\left(l_l - \rho_{out}\right) \left(t_{LL} - t_{SS}\right)} < \hat{w}_l - w_l$$

$$w_l < w_l + \frac{sv_{LL} - sv_{SS}}{\left(l_l - \rho_{out}\right) \left(t_{LL} - t_{SS}\right)} < \hat{w}_l$$

## Appendix 12

### Situations in which the rewarding option does not exclude the animal from receiving outside reward

$$\frac{sv + \rho_{out}t_{out}}{t_{out}} = \frac{r_{in} + \rho_{out}\left(t_{in} + t_{out}\right)}{t_{in} + t_{out}}$$

$$\frac{sv}{t_{out}} + \rho_{out} = \frac{r_{in}}{t_{in} + t_{out}} + \rho_{out}$$

$$\frac{sv}{t_{out}} = \frac{r_{in}}{t_{in} + t_{out}}$$

$$sv = \frac{r_{in}}{1 + t_{in}/t_{out}} \text{(Equation 12 in Main Text)}$$

