## [Editor Report · eLife Assessment]

The paper presents a **valuable** theoretical treatment of the role of passage of time in optimal decision strategies in pursuit based tasks. The computational evidence and methodologies employed are novel, and the authors offer **solid** evidence for the majority of the claims.

---

## [Referee Report · Reviewer #2 (Public review)]

Summary:

This paper from Sutlief et al. focuses on an apparent contradiction observed in experimental data from two related types of pursuit-based decision tasks. In "forgo" decisions, where the subject is asked to choose whether or not to accept a presented pursuit, after which they are placed into a common inter-trial interval, subjects have been shown to be nearly optimal in maximizing their overall rate of reward. However, in "choice" decisions, where the subject is asked which of two mutually-exclusive pursuits they will take, before again entering a common inter-trial interval, subjects exhibit behavior that is believed to be sub-optimal. To investigate this contradiction, the authors derive a consistent reward-maximizing strategy for both tasks using a novel and intuitive geometric approach that treats every phase of a decision (pursuit choice and inter-trial interval) as vectors. From this approach, the authors are able to show that previously-reported examples of sub-optimal behavior in choice decisions are in fact consistent with a reward-maximizing strategy. Additionally, the authors are able to use their framework to deconstruct the different ways the passage of time impacts decisions, demonstrating the time cost contains both an opportunity cost and an apportionment cost, as well as examine how a subject's misestimation of task parameters impacts behavior.

Strengths:

The main strength of the paper lies in the authors' geometric approach to studying the problem. The authors chose to simplify the decision process by removing the highly technical and often cumbersome details of evidence accumulation that is common in most of the decision-making literature. In doing so, the authors were able to utilize a highly accessible approach that is still able to provide interesting insights into decision behavior and the different components of optimal decision strategies.

Weaknesses:

The authors have made great improvements to the strength of their evidence through revision, especially concerning their treatment of apportionment cost. However, I am concerned that the story this paper tells is far from concise, and that this weakness may limit the paper's audience and overall impact. I would strongly suggest making an effort to tighten up the language and structure of the paper to improve its readability and accessibility.

---

## [Referee Report · Reviewer #3 (Public review)]

Summary:

The goal of the paper is to examine the objective function of total reward rate in an environment to understand behavior of humans and animals in two types of decision-making tasks: (1) stay/forgo decisions and (2) simultaneous choice decisions. The main aims are to reframe the equation of optimizing this normative objective into forms that are used by other models in the literature like subjective value and temporally discounted reward. One important contribution of the paper is the use of this theoretical analysis to explain apparent behavioral inconsistencies between forgo and choice decisions observed in the literature.

Strengths:

The paper provides a nice way to mathematically derive different theories of human and animal behavior from a normative objective of global reward rate optimization. As such, this work has value in trying to provide a unifying framework for seemingly contradictory empirical observations in literature, such as differentially optimal behaviors in stay-forgo v/s choice decision tasks. The section about temporal discounting is particularly well motivated as it serves as another plank in the bridge between ecological and economic theories of decision-making. The derivation of the temporal discounting function from subjective reward rate is much appreciated as it provides further evidence for potential equivalence between reward rate optimization and hyperbolic discounting, which is known to explain a slew of decision-making behaviors in the economics literature.

Weaknesses:

(1) Readability and organization:

While I appreciate the detailed analysis and authors' attempts to provide as many details as possible, the paper would have benefitted from a little selectivity on behalf of the authors so that the main contributions aren't buried by the extensive mathematical detail provided.

For instance, in Figure 5, the authors could have kept the most important figures (A, B and G) to highlight the most relevant terms in the subjective value instead of providing all possible forms of the equation.

Further, in subfigure 5E, is there a reason that the outside reward r_out is shown to be zero? The text referencing 5E is also very unclear: "In so downscaling, the subjective value of a considered pursuit (green) is to the time it would take to traverse the world were the pursuit not taken, 𝑡_out, as its opportunity cost subtracted reward (cyan) is to the time to traverse the world were it to be taken (𝑡_in+ 𝑡_out) (Figure 5E)."

In the abstract, the malapportionment of time is mentioned as a possible explanation for reconciling observed empirical results between simultaneous and sequential decision-making. However, perhaps due to the density of mathematical detail presented, the discussion of the malapportionment hypothesis is pushed all the way to the end of the discussion section.

(2) Apportionment Cost definition and interpretation

This additional cost arises in their analyses from redefining the opportunity cost in terms of just "outside" rewards so that the subjective value of the current pursuit and the opportunity cost are independent of each other. However, in doing so, an additional term arises in defining the subjective value of a pursuit, named here the "apportionment cost". The authors have worked hard to provide a definition to conceptualize the apportionment cost though it remains hard to intuit, especially in comparison to the opportunity cost. The additive form of apportionment cost (Equation 9) doesn't add much in way of intuition or their later analyses for the malapportionment hypothesis. It appears that the most important term is the apportionment scaling term so just focusing on this term will help the reader through the subsequent analyses.

(3) Malapportionment Hypothesis: From where does this malapportionment arise?

The authors identify the range of values for t_in and t_out in Figure 18, the terms comprising the apportionment scaling term, that lead to optimal forgo behaviors despite suboptimally rejecting the larger-later (LL) choice in choice decisions. They therefore conclude that a lower apportionment scale, which arises from overestimating the time required outside the pursuit (t_out) or underestimating the time required at the current pursuit (t_in). What is not discussed though is whether and how the underestimation of t_out and overestimation of t_in can be dissociated, though it is understood that empirical demonstration of this dissociation is outside the scope of this work.

---

## [Author Response]

The following is the authors’ response to the original reviews.

**Public Reviews:**

**Reviewer #1 (Public review):**
(1) Although there are many citations acknowledging relevant previous work, there often isn't a very granular attribution of individual previous findings to their sources. In the results section, it's sometimes ambiguous when the paper is recapping established background and when it is breaking new ground. For example, around equation 8 in the results (sv = r - rho*t), it would be good to refer to previous places where versions of this equation have been presented. Offhand, McNamara 1982 (Theoretical Population Biology) is one early instance and Fawcett et al. 2012 (Behavioural Processes) is a later one. Line 922 of the discussion seems to imply this formulation is novel here.

We would like to clarify that original manuscript equation 8, sv=rin−ρgtin, as we derive, is not new, as it is similarly expressed in prior foundational work by McNamara (1982), and we thank the reviewer for drawing our attention to the extension of this form by Fawcett, McNamara, Houston (2012).

We now so properly acknowledge this foundational work and extension in the results section…

“This global reward-rate equivalent immediate reward (see Figure 4) is the subjective value of a pursuit, svPursuit (or simply, sv, when the referenced pursuit can be inferred), as similarly expressed in prior foundational work (McNamara 1982), and subsequent extensions (see Fawcett, McNamara, Houston (2012)).”

…and in the Discussion section at the location referenced by the reviewer:

“From it, we re-expressed the pursuit’s worth in terms of its global reward rate-equivalent immediate reward, i.e., its ‘subjective value’, reprising McNamara’s foundational formulation (McNamara 1982).”

(2) The choice environments that are considered in detail in the paper are very simple. The simplicity facilitates concrete examples and visualizations, but it would be worth further consideration of whether and how the conclusions generalize to more complex environments. The paper considers "forgo" scenario in which the agent can choose between sequences of pursuits like A-B-A-B (engaging with option B at all opportunities, which are interleaved with a default pursuit A) and A-A-A-A (forgoing option B). It considers "choice" scenarios where the agent can choose between sequences like A-B-A-B and A-C-A-C (where B and C are larger-later and smaller-sooner rewards, either of which can be interleaved with the default pursuit). Several forms of additional complexity would be valuable to consider. [A] One would be a greater number of unique pursuits, not repeated identically in a predictable sequence, akin to a prey-selection paradigm. It seems to me this would cause t_out and r_out (the time and reward outside of the focal prospect) to be policy-dependent, making the 'apportionment cost' more challenging to ascertain. Another relevant form of complexity would be if there were [B] variance or uncertainty in reward magnitudes or temporal durations or if [C] the agent had the ability to discontinue a pursuit such as in patch-departure scenarios.

A) We would like to note that the section “Deriving Optimal Policy from Forgo Decision-making worlds”, addresses the reviewer’s scenario of n-number of pursuits”, each occurring at their own frequency, as in prey selection, not repeating identically in a predictable sequence. Within our subsection “Parceling the world…”, we introduce the concept of dividing a world (such as that) into the considered pursuit type, and everything outside of it. ‘Outside’ would include any number of other pursuits currently part of any policy, as the reviewer intuits, thus making t^out^ and r^out^ policy dependent. Nonetheless, a process of excluding (forgoing) pursuits by comparing the ‘in’ to the ‘out’ reward rate (section “Reward-rate optimizing forgo policy…”) or its equivalent sv (section “The forgo decision can also be made from subjective value), would iteratively lead to the global reward-rate maximizing policy. This manner of parceling into ‘in’ and ‘out’ thus simplifies visualization of what can be complex worlds. Simpler cases that resemble common experimental designs are given in the manuscript to enhance intuition.

We thank the reviewer for this keen suggestion. We now include example figures (Supplemental 1 & 2) for multi-pursuit worlds which have the same (Supplemental 1) and different pursuit frequencies (Supplemental 2), which illustrate how this evaluation leads to reward-rate optimization. This addition demonstrates how an iterative policy would lead to reward-rate maximization and emphasizes how parcellating a world into ‘in’ and ‘out’ of the pursuit type applies and is a useful device for understanding the worth of any given pursuit in more complex worlds. The policy achieving the greatest global reward rate can be realized through an iterative process where pursuits with lower reward rates than the reward rate obtained from everything other than the considered pursuit type are sequentially removed from the policy.

B) We would also emphasize that the formulation here contends with variance or uncertainty in the reward magnitudes or temporal durations. The ‘in’ pursuit is the average reward and the average time of the considered pursuit type, as is the ‘out’ the average reward and average time outside of the considered pursuit type.

C) In this work, we consider the worth of initiating one-or-another pursuit (from having completed a prior one), and not the issue of continuing within a pursuit (having already engaged it), as in patch/give-up. Handling worlds in which the agent may depart from within a pursuit, which is to say ‘give-up’ (as in patch foraging), is outside the scope of this work.

(3) I had a hard time arriving at a solid conceptual understanding of the 'apportionment cost' around Figure 5. I understand the arithmetic, but it would help if it were possible to formulate a more succinct verbal description of what makes the apportionment cost a useful and meaningful quality to focus on.

We thank the reviewer for pressing for a succinct and intuitive verbal description.

We added the following succinct verbal description of apportionment cost… “Apportionment cost is the difference in reward that can be expected, on average, between a policy of taking versus a policy of not taking the considered pursuit, over a time equal to its duration.” This definition appears in new paragraphs (as below) describing apportionment cost in the results section “Time’s cost: opportunity & apportionment costs determine a pursuit’s subjective value”, and is accompanied by equations for apportionment cost, and a figure giving its geometric depiction (Figure 5). We also expanded original figure 5 and its legend (so as to illustrate the apportionment scaling factor and the apportionment cost), and its accompanying main text, to further illustrate and clarify apportionment cost, and its relationship to opportunity cost, and time’s cost.

“What, then, is the amount of reward by which the opportunity cost-subtracted reward is scaled down to equal the *sv* of the pursuit? This amount is the apportionment cost of time. The apportionment cost of time (height of the brown vertical bar, Figure 5F) is the global reward rate after taking into account the opportunity cost (slope of the magenta-gold dashed line in Figure 5F) times the time of the considered pursuit. Equally, the difference between the inside and outside reward rates, times the time of the pursuit, is the apportionment cost when scaled by the pursuit’s weight, i.e., the fraction that the considered pursuit is to the total time to traverse the world (Equation 9, right hand side). From the perspective of decision-making policies, apportionment cost is the difference in reward that can be expected, on average, between a policy of taking versus a policy of not taking the considered pursuit, over a time equal to its duration (Equation 9 center, Figure 5F).=tin(ρg−ρout)=tin(rintin−routtout)(tintin+tout)

Equation 9. Apportionment Cost.

While this difference is the apportionment cost of time, the opportunity cost of time is the amount that would be expected from a policy of not taking the considered pursuit over a time equal to the considered pursuit’s duration. Together, they sum to Time’s Cost (Figure 5G). Expressing a pursuit’s worth in terms of the global reward rate obtained under a policy of accepting the pursuit type (Figure 5 left column), or from the perspective of the outside reward and time (Figure 5 right column), are equivalent. However, the latter expresses *sv* in terms that are independent of one another, conveys the constituents giving rise to global reward rate, and provides the added insight that time’s cost comprises an apportionment as well as an opportunity cost.”

The above definition of apportionment cost adds to other stated relationships of apportionment cost found throughout the paper (original lines 434,435,447,450).

I think Figure 6C relates to this, but I had difficulty relating the axis labels to the points, lines, and patterned regions in the plot.

We thank the reviewer for pointing out that this figure can be made to be more easily understood.

We have done so by breaking its key features over a greater number of plots so that no single panel is overloaded. We have also changed text in the legend to clarify how apportionment and opportunity costs add to constitute time’s cost, and also correspondingly in the main text.

I also was a bit confused by how the mathematical formulation was presented. As I understood it, the apportionment cost essentially involves scaling the rest of the SV expression by t^out^/(t^in^ + t^out^).

The reviewer’s understanding is correct: the amount of reward of the pursuit that remains after subtracting the opportunity cost, when so scaled, is equivalent to the subjective value of that pursuit. The amount by which that scaling decreases the rest of the SV expression is equal to the apportionment cost of time.

The way this scaling factor is written in Figure 5C, as 1/(1 + (1/t^out^) t^in^), seems less clear than it could be.

To be sure, we present the formula in original Figure 5C in this manner to emphasize the opportunity cost subtraction as separable from the apportionment rescaling, expressing the opportunity cost subtraction and the apportionment scaling component of the equation as their own terms in parentheses.

But we understand the reviewer to be referring to the manner by which we chose to express the scaling term. We presented it in this way in the original manuscript, (rather than its more elegant form recognized by the reviewer) to make direct connection to temporal discounting literature. In this literature, discounting commonly takes the same mathematical form as our apportionment cost scaling, but whereas the steepness of discounting in this literature is controlled by a free fit parameter, *k*, we show how for a reward-rate maximizing agent, the equivalent *k* term isn’t a free fit parameter, but rather is the reciprocal of the time spent outside the considered pursuit type.

We take the reviewer’s advice to heart, and now first express subjective value in the format that emphasizes opportunity cost subtraction followed by an apportionment downscaling, identifying the apportionment scaling term, t^out^/(t^out^ + t^in^), ie the outside weight. Figure 5 now shows the geometric representation of apportionment scaling and apportionment cost. Only subsequently in the discounting function section then do we now in the revised manuscript rearrange this subjective value expression to resemble the standard discounting function form.

Also, the apportionment cost is described in the text as being subtracted from *sv* rather than as a multiplicative scaling factor.

What we describe in the original text is how apportionment cost is a component of time’s cost, and how *sv* is the reward less time’s cost. It would be correct to say that apportionment cost and opportunity cost are subtracted from the pursuit’s reward to yield the subjective value of the pursuit. This is what we show in the original Figure 5D graphically. Original Figure 5 and accompanying formulas at its bottom show the equivalence of expressing *sv* in terms of subtracting time’s cost as calculated from the global reward rate under a policy of accepting the considered pursuit, or, of subtracting opportunity cost and then scaling the opportunity cost subtracted reward by the apportionment scaling term, thereby accounting for the apportionment cost of time.

The revision of original figure 5, its figure legend, and accompanying text now make clear the meaning of apportionment cost, how it can be considered a subtraction from the reward of a pursuit, or, equivalently, how it can be thought of as the result of scaling down of opportunity cost subtracted reward.

It could be written as a subtraction, by subtracting a second copy of the rest of the SV expression scaled by t_in/(t_in + t_out). But that shows the apportionment cost to depend on the opportunity cost, which is odd because the original motivation on line 404 was to resolve the lack of independence between terms in the SV expression.

On line 404 of the original manuscript, we point out that the simple equation―which is a reprisal of McNamara’s insight―is problematic in that its terms on the RHS are not independent: the global reward rate is dependent on the considered pursuit’s reward (see Fig5B). The alternative expression for subjective value that we derive expresses *sv* in terms that are all independent of one another. We may have unintentionally obscured that fact by having already defined rho^in^ as r^in^/ t^in^ and rho^out^ as r^out^/t^out^ on lines 306 and 307.

Therefore, in the revision, Ap 8 is expressed so to keep clear that it uses terms that are all independent of one another, and only subsequently express this formula with the simplifying substitution, rho^out^.

That all said, we understand the reviewer’s point to be that the parenthetical terms relating the opportunity cost and the apportionment rescaling both contain within them the parameter t^out^, and in this way these concepts we put forward to understand the alternative equation are non-independent. That is correct, but it isn’t at odds with our objective to express SV in terms that are independent with one another (which we do). Our motivation in introducing these concepts is to provide insight and intuition into the cost of time (especially now with a clear and simple definition of apportionment cost stated). We go to lengths to demonstrate their relationship to each other.

(4) In the analysis of discounting functions (line 664 and beyond), the paper doesn't say much about the fact that many discounting studies take specific measures to distinguish true time preferences from opportunity costs and reward-rate maximization.

We understand the reviewer’s comment to connote that temporal decision-making worlds in which delay time does not preclude reward from outside the current pursuit is a means to distinguish time preference from the impact of opportunity cost. One contribution of this work is to demonstrate that, from a reward-rate maximization framework, an accounting of opportunity cost is not sufficient to understand apparent time preferences as distinguishable from reward-rate maximization. The apportionment cost of time must also be considered to have a full appreciation of the cost of time. For instance, let us consider a temporal decision-making world in which there is no reward received outside the considered pursuit. In such a world, there is no opportunity cost of time, so apparent temporal discounting functions would appear as if purely hyperbolic as a consequence of the apportionment cost of time alone. Time preference, as revealed experimentally by the choices made between a SS and a LL reward, then, seem confounding, as preference can reverse from a SS to a LL option as the displacement of those options (maintaining their difference in time) increases (Green, Fristoe, and Myerson 1994; Kirby and Herrnstein 1995). While this shift, the so-called “Delay effect”, could potentially arise as a consequence of some inherent time preference bias of an agent, we demonstrate that a reward-rate maximal agent exhibits hyperbolic discounting, and therefore it would also exhibit the Delay effect, even though it has no time preference.

In the revision we now make reference to the Delay Effect (in abstract, results new section “The Delay Effect” with new figure 14, and in the discussion), which is taken as evidence of time preference in human and animal literature, and note explicitly how a reward-rate maximizing agent would also exhibit this behavior as a consequence of apparent hyperbolic discounting.

In many of the human studies, delay time doesn't preclude other activities.

Our framework is generalizable to worlds in which being in pursuit does not preclude an agent from receiving reward during that time at the outside reward rate. Original Ap 13 solves for such a condition, and shows that in this context, the opportunity cost of time drops out of the SV equation, leaving only the consequences of the apportionment cost of time. We made reference to this case on lines 1032-1034 of the original manuscript: “In this way, such hyperbolic discounting models [models that do not make an accounting of opportunity cost] are only appropriate in worlds with no “outside” reward, or, where being in a pursuit does not exclude the agent from receiving rewards at the rate that occurs outside of it (Ap. 13).”

The note and reference is fleeting in the original work. We take the reviewer’s suggestion and now add paragraphs in the discussion on the difference between humans and animals in apparent discounting, making specific note of human studies in which delay time doesn’t preclude receiving outside reward while engaged in a pursuit. Relatedly, hyperbolic discounting is oft considered to be less steep in humans than in animals. As the reviewer points out, these assessments are frequently made under conditions in which being in a pursuit does not preclude receiving reward from outside the pursuit. When humans are tested under conditions in which outside rewards are precluded, they exhibit far steeper discounting. We now include citation to that observation (Jimura et al. 2009). We handle such conditions in original AP 13, and show how, in such worlds, the opportunity cost of time drops out of the equation. The consequence of this is that the apparent discounting function would become less steep (the agent would appear as if more patient), consistent with reports.

“Relating to the treatment of opportunity cost, we also note that many investigations into temporal discounting do not make an explicit distinction between situations in which (1) subjects continue to receive the usual rewards from the environment during the delay to a chosen pursuit, and (2) situations in which during a chosen pursuit’s delay no other rewards or opportunities will occur (Kable & Glimcher, 2007; Kirby & Maraković, 1996; McClure, Laibson, Loewenstein, & Cohen, 2004). Commonly, human subjects are asked to answer questions about their preferences between options for amounts they will not actually earn after delays they will not actually have to wait, during which it is unclear whether they are really investing time away from other options or not (Rosati et al., 2007). In contrast, in most animal experiments, subjects actually receive reward after different delays during which they do not receive new options or rewards. By our formulation, when a pursuit does not exclude the agent from receiving rewards at the rate that occurs outside, the opportunity cost of time drops out of the subjective value equation (Ap 12).sv=rin 1+tin /tout 

Equation 10. The value of initiating a pursuit when pursuit does not exclude receiving rewards at the outside rate (Ap 12)

Therefore, the reward-rate maximizing discounting function in these worlds is functionally equivalent to the situation in which the outside reward rate is zero, and will―lacking an opportunity cost―be less steep. This rationalizes why human discounting functions are often reported to be longer (gentler) than animal discounting functions: they are typically tested in conditions that negate opportunity cost, whereas animals are typically tested in conditions that enforce opportunity costs. Indeed, when humans are made to wait for actually received reward, their observed discounting functions are much steeper (Jimura et al. 2009). “

In animal studies, rate maximization can serve as a baseline against which to measure additional effects of temporal discounting. This is an important caveat to claims about discounting anomalies being rational under rate maximization (e.g., line 1024).

We agree that the purpose of this reward-rate maximizing framework is to serve as a point of comparison in which effects of temporal intervals and rewards that define the environment can be analyzed to better understand the manner in which animals and humans deviate from this ideal behavior. Our interest in this work is in part motivated by a desire to have a deeper understanding of what “true” time preference means. Using the reward-rate maximizing framework here provides a means to speak about time preferences (ie biases) in terms of deviation from optimality. From this perspective, a reward-rate maximal agent doesn’t exhibit time preference: its actions are guided solely by reward-rate optimizing valuation. Therefore, one contribution of this work is to show that purported signs of time preference (hyperbolic discounting, magnitude, sign, and (now) delay effect) can be explained without invoking time preference. What errors from optimality that remain following an proper accounting of reward-rate maximizing behavior should then, and only then, be considered from the lens of time preference (bias).

(5) The paper doesn't feature any very concrete engagement with empirical data sets. This is ok for a theoretical paper, but some of the characterizations of empirical results that the model aims to match seem oversimplified. An example is the contention that real decision-makers are optimal in accept/reject decisions (line 816 and elsewhere). This isn't always true; sometimes there is evidence of overharvesting, for example.

We would like to note that the scope of this paper is limited to examining the value of initiating a pursuit, rather than the value of continuing within a pursuit. The issue of continuing within a pursuit constitutes a third fundamental topology, which could be called give-up or patch-foraging, and is complex and warrants its own paper. In Give-up topologies, which are distinct from Forgo, and Choice topologies, the reviewer is correct in pointing out that the preponderance of evidence demonstrates that animals and humans are as if overpatient, adopting a policy of investing too much time within a pursuit, than is warranted_._ In Forgo instances, however, the evidence supports near optimality.

(6) Related to the point above, it would be helpful to discuss more concretely how some of this paper's theoretical proposals could be empirically evaluated in the future. Regarding the magnitude and sign effects of discounting, there is not a very thorough overview of the several other explanations that have been proposed in the literature. It would be helpful to engage more deeply with previous proposals and consider how the present hypothesis might make unique predictions and could be evaluated against them.

We appreciate the reviewer’s point that there are many existing explanations for these various ‘anomalous’ effects. We hold that the point of this work is to demonstrate that these effects are consistent with a reward-rate maximizing framework so do not require additional assumptions, like separate processes for small and large rewards, or the inclusion of a utility function.

Nonetheless, there is a diversity of explanations for the sign and magnitude effect, and, (now with its explicit inclusion in the revision) the delay effect. Therefore, we now also include reference to additional work which proffers alternative explanations for the sign and magnitude effects, (as reviewed by (Kalenscher and Pennartz 2008; Frederick et al. 2002)), as well as a scalar timing account of non-stationary time preference (Gibbon, 1977).

With respect to making predictions, this framework makes the following in regards to the magnitude, sign, and (now in the revision) delay effect: in Discussion, Magnitude effect subsection: “The Magnitude Effect should be observed, experimentally, to diminish when (1) increasing the outside time while holding the outside reward constant, (thus decreasing the outside reward rate), or when (2) decreasing the outside reward while holding the outside time constant (thus decreasing the outside reward rate). However, (3) the Magnitude Effect would exaggerate as the outside time increased while holding the outside reward rate constant.”, in Sign effect subsection: “…we then also predict that the size of the Sign effect would diminish as the outside reward rate decreases (and as the outside time increases), and in fact would invert should the outside reward rate turn negative (become net punishing), such that punishments would appear to discount more steeply than rewards.” Delay effect subsection: “...a sign of irrationality is that a preference reversal occurs at delays greater than what a reward-rate-maximizing agent would exhibit.”

A similar point applies to the 'malapportionment hypothesis' although in this case there is a very helpful section on comparisons to prior models (line 1163). The idea being proposed here seems to have a lot in common conceptually with Blanchard et al. 2013, so it would be worth saying more about how data could be used to test or reconcile these proposals.

We thank the reviewer for holding that the section of model comparisons to be very helpful. We believe the text previously dedicated to this issue to be sufficient in this regard. We have, however, adding substantively to the Malapportionment Hypothesis section (Discussion) and its accompanying figure, to make explicit a number of predictions from the Malapportionment hypothesis as it relates to Hyperbolic discounting, the Delay Effect, and the Sign and Magnitude Effects.

**Reviewer #1 Recommendations**
(1) As a general note about the figures, it would be helpful to specify, either graphically or in the caption, what fixed values of reward sizes and time intervals are being assumed for each illustration.

Thank you for the suggestion. We attempted to keep graphs as uncluttered as possible, but agree that for original figures 4,5,16, and 17, which didn’t have numbered axes, that we should provide the amounts in the captions in the revised figures (4,5, and now 17,18). These figures did not have numerics as their shapes and display are to illustrate the form of the relationship between vectors, being general to the values they may take.

We now include in the captions for these figures the parameter amounts used.

(2) Should Equation 2 have t in the denominator instead of r?

Indeed. We thank the reviewer for catching this typographical error.

We have corrected it in the revision.

(3) General recommendation:My view is that in order for the paper's eLife assessment to improve, it would be necessary to resolve points 1 through 4 listed under "weaknesses" in my public review, which pertain to clarity and acknowledgement of prior work. I think a lot hinges on whether the authors can respond to point #3 by making a more compelling case for the usefulness and generality of the 'apportionment cost' concept, since that idea is central to the paper's contribution.

We believe these critical points (1-4) to improve the paper will now have been addressed to the reviewer’s satisfaction.

**Reviewer #2 (Public review):**
While the details of the paper are compelling, the authors' presentation of their results is often unclear or incomplete:(1) The mathematical details of the paper are correct but contain numerous notation errors and are presented as a solid block of subtle equation manipulations. This makes the details of the authors' approach (the main contribution of the paper to the field) highly difficult to understand.

We thank the reviewers for having detected typographical errors regarding three equations. They have been corrected. The first typographical error in the original main text (Line 277) regards equation 2 and will be corrected so that equation 2 appears correctly asρg=rin+∑i≠innfiri+ρdfintin+∑i≠innfiti+1fin

The second typo regards the definition of the considered pursuit’s reward rate which appear in the original main text (line 306), and has been corrected to appear asρin=rintin

The third typographical error occurred in conversion from Google Sheets to Microsoft Word appearing in the original main text (line 703) and regards the subjective value expression when no reward is received in an intertrial interval (ITI). It has been corrected to appear assv=rin1+tin tout 

(2) One of the main contributions of the paper is the notion that time’s cost in decision-making contains an apportionment cost that reflects the allocation of decision time relative to the world. The authors use this cost to pose a hypothesis as to why subjects exhibit sub-optimal behavior in choice decisions. However, the equation for the apportionment cost is never clearly defined in the paper, which is a significant oversight that hampers the effectiveness of the authors' claims.

We thank the reviewer for pressing on this critical point. Reviewers commonly identified a need to provide a concise and intuitive definition of apportionment cost, and to explicitly solve and provide for its mathematical expression.

We added the following succinct verbal description of apportionment cost… “Apportionment cost is the difference in reward that can be expected, on average, between a policy of taking versus a policy of not taking the considered pursuit, over a time equal to its duration.” This definition appears in new paragraphs (as below) describing apportionment cost in the results section “Time’s cost: opportunity & apportionment costs determine a pursuit’s subjective value”, and is accompanied by equations for apportionment cost, and a figure giving its geometric depiction (Figure 5). We also expanded original figure 5 and its legend (so as to illustrate the apportionment scaling factor and the apportionment cost), and its accompanying main text, to further illustrate and clarify apportionment cost, and its relationship to opportunity cost, and time’s cost.

“What, then, is the amount of reward by which the opportunity cost-subtracted reward is scaled down to equal the *sv* of the pursuit? This amount is the apportionment cost of time. The apportionment cost of time (height of the brown vertical bar, Figure 5F) is the global reward rate after taking into account the opportunity cost (slope of the magenta-gold dashed line in Figure 5F) times the time of the considered pursuit. Equally, the difference between the inside and outside reward rates, times the time of the pursuit, is the apportionment cost when scaled by the pursuit’s weight, i.e., the fraction that the considered pursuit is to the total time to traverse the world (Equation 9, right hand side). From the perspective of decision-making policies, apportionment cost is the difference in reward that can be expected, on average, between a policy of taking versus a policy of not taking the considered pursuit, over a time equal to its duration (Equation 9 center, Figure 5F).=tin(ρg−ρout)=tin(rintin−routtout)(tintin+tout)

Equation 9. Apportionment Cost.

While this difference is the apportionment cost of time, the opportunity cost of time is the amount that would be expected from a policy of not taking the considered pursuit over a time equal to the considered pursuit’s duration. Together, they sum to Time’s Cost (Figure 5G). Expressing a pursuit’s worth in terms of the global reward rate obtained under a policy of accepting the pursuit type (Figure 5 left column), or from the perspective of the outside reward and time (Figure 5 right column), are equivalent. However, the latter expresses *sv* in terms that are independent of one another, conveys the constituents giving rise to global reward rate, and provides the added insight that time’s cost comprises an apportionment as well as an opportunity cost.”

(3) Many of the paper's figures are visually busy and not clearly detailed in the captions (for example, Figures 6-8). Because of the geometric nature of the authors' approach, the figures should be as clean and intuitive as possible, as in their current state, they undercut the utility of a geometric argument.

We endeavored to make our figures as simple as possible. We have made in the revision changes to figures that we believe improve their clarity. These include: (1) breaking some figures into more panels when more than one concept was being introduced (such as in revised Figure 5 , 6, 7, and 8), (2) using the left hand y axis for the outside reward, and the right hand axis for the inside reward when plotting the “in” and “outside” reward, and indicating their respective numerics (which run in opposite directions), (3) adding a legend to the figures themselves where needed (revised figures 10, 11, 12, 14) (4) adding the values used to the figure captions, where needed, and (5) ensuring all symbols are indicated in legends.

(4) The authors motivate their work by focusing on previously-observed behavior in decision experiments and tell the reader that their model is able to qualitatively replicate this data. This claim would be significantly strengthened by the inclusion of experimental data to directly compare to their model's behavior. Given the computational focus of the paper, I do not believe the authors need to conduct their own experiments to obtain this data; reproducing previously accepted data from the papers the authors' reference would be sufficient.

Our objective was not to fit experimentally observed data, as is commonly the goal of implementation/computational models. Rather, as a theory, our objective is to rationalize the broad, curious, and well-established pattern of temporal decision-making behaviors under a deeper understanding of reward-rate maximization, and from that understanding, identify the nature of the error being committed by whatever learning algorithm and representational architecture is actually being used by humans and animals. In doing so, we make a number of important contributions. By identifying and analyzing reward-rate-maximizing equations, we (1) provide insight into what composes time’s cost and how the temporal structure of the world in which it is embedded (its ‘context’) impacts the value of a pursuit, (2) rationalize a diverse assortment of temporal decision-making behaviors (e.g., Hyperbolic discounting, the Magnitude Effect, the Sign Effect, and the Delay effect), explaining them with no assumed free-fit parameter, and then, by analyzing error in parameters enabling reward-rate maximization, (3) identify the likely source of error and propose the Malapportionment Hypothesis. The Malapportionment Hypothesis identifies the underweighting of a considered pursuit’s “outside”, and not error in pursuit’s reward rates, as the source of error committed by humans and animals. It explains why animals and humans can present as suboptimally ‘impatient’ in Choice, but as optimal in Forgo. At the same time, it concords with numerous and diverse observations in decision making regarding whether to initiate a pursuit. The nature of this error also, then, makes numerous predictions. These insights inform future computational and experimental work by providing strong constraints on the nature of the algorithm and representational architecture used to learn and represent the values of pursuits. Rigorous test of the Malapportionment Hypothesis will require wholly new experiments.

In the revision, we also now emphasize and add predictions of the Malapportionment Hypothesis, updated its figure (Figure 21), its legend, and its paragraphs in the discussion.

“We term this reckoning of the source of error committed by animals and humans the Malapportionment Hypothesis, which identifies the underweighting of the time spent outside versus inside a considered pursuit but not the misestimation of pursuit rates, as the source of error committed by animals and humans (Figure 21). This hypothesis therefore captures previously published behavioral observations (Figure 21A) showing that animals can make decisions to take or forgo reward options that optimize reward accumulation (Krebs et al., 1977; Stephens and Krebs, 1986; Blanchard and Hayden, 2014), but make suboptimal decisions when presented with simultaneous and mutually exclusive choices between rewards of different delays (Logue et al., 1985; Blanchard and Hayden, 2015; Carter and Redish, 2016; Kane et al., 2019). The Malapportionment Hypothesis further predicts that apparent discounting functions will present with greater curvature than what a reward-rate-maximizing agent would exhibit (Figure 21B). While experimentally observed temporal discounting would have greater curvature, the Malapportionment Hypothesis also predicts that the Magnitude (Figure 21C) and Sign effect (Figure 21D) would be less pronounced than what a reward-rate-maximizing agent would exhibit, with these effects becoming less pronounced the greater the underweighting. Finally, with regards to the Delay Effect (Figure 21E), the Malapportionment Hypothesis predicts that preference reversal would occur at delays greater than that exhibited by a reward-rate-maximizing agent, with the delay becoming more pronounced the greater the underweighting outside versus inside the considered pursuit by the agent.”

(5) While the authors reference a good portion of the decision-making literature in their paper, they largely ignore the evidence-accumulation portion of the literature, which has been discussing time-based discounting functions for some years. Several papers that are both experimentally-(Cisek et al. 2009, Thurs et al. 2012, Holmes et al. 2016) and theoretically-(Drugowitsch et al. 2012, Tajima et al. 2019, Barendregt et al. 22) driven exist, and I would encourage the authors to discuss how their results relate to those in different areas of the field.

In this manuscript, we consider the worth of initiating one or another pursuit having completed a prior one, and not the issue of continuing within a pursuit having already engaged in it. The worth of continuing a pursuit, as in patch-foraging/give-up tasks, constitutes a third fundamental time decision-making topology which is outside the scope of the current work. It engages a large and important literature, encompassing evidence accumulation, and requires a paper on the value of continuing a pursuit in temporal decision making, in its own right, that can use the concepts and framework developed here. The excellent works suggested by the reviewer will be most relevant to that future work concerning patch-foraging/give-up topologies.

**Reviewer #2 Recommendations:**
(1) In Equation 1, the term rho_d is referred to as the reward rate of the default pursuit, when it should be the reward of the default pursuit.

Regarding Equation 1, it is formulated to calculate the average reward received and average time spent per unit time spent in the default pursuit. So, *fi* is the encounter rate of pursuit *i* for one unit of time spent in the default pursuit (lines 259-262). Added to the summation in the numerator, we have the average reward obtained in the default pursuit per unit time (ρd∗1) and in the denominator we have the time spent in the default pursuit per unit time (1).ρg=∑i=1nfiri+ρd∑i=1nfiti+1

We have added clarifying text to assist in meaning of the equation in Ap 1, and thank the reviewer for pointing out this need.

(2) The notation for "in" and "out" of a considered pursuit type begins as being used to describe the contribution from a single pursuit (without inter-trial interval) towards global reward rate and the contribution of all other factors (other possible pursuits and inter-trial interval) towards global reward rate, respectively, but is then used to describe the pursuit's contribution and the inter-trial interval's contribution, respectively, to the global reward rate. This should be cleaned up to be consistent throughout, or at the very least, it should be addressed when this special case is considered the default.

As understood by the reviewer, “in” and “out” of the considered pursuit type describes the general form by which a world can be cleaved into these two parts: the average time and reward received outside of the considered pursuit type for the average time and reward received within that pursuit type. A specific, simple, and common experimental instance would be a world composed of one or another pursuit and an intertrial interval.

We now make clear how such a world composed of a considered pursuit and an inter trial interval would be but one special case. In example cases where t^out^ represents the special case of an inter-trial interval, this is now stated clearly. For instance, we do so when discussing how a purely hyperbolic discounting function would apply in worlds in which no reward is received in t^out^, stating that this is often the case common to experimental designs where t^out^ represents an intertrial interval with no reward. Importantly, by the new inclusion of illustrated worlds in the revision that have n-number pursuits that could occur from a default pursuit and (1) equal frequency (Supplemental 1), and (2) at differing frequencies (Supplemental 2), we make more clear the generalizability and utility of this t^out^/tin concept.

(3) Figure 5 should make clear the decomposition of time's cost both graphically and functionally. As it stands, the figure does not define the apportionment cost.

In the revision of original fig 5, we now further decompose the figure to effectively convey (1) what opportunity cost, and (especially) (2) the apportionment cost is, both graphically and mathematically, (3) how time’s cost is comprised by them, (4) how the apportionment scaling term scales the opportunity-cost-subtracted reward by time’s allocation to equal the subjective value, and (4) the equivalence between the expression of time’s cost using terms that are not independent of one another with the expression of time’s cost using terms that are independent of one another.

(4) Figures 6-8 do not clearly define the dots and annuli used in panels B and C.

We have further decomposed figures 6-8 so that the functional form of opportunity, apportionment, and time’s cost can be more clearly appreciated, and what their interrelationship is with respect to changing outside reward and outside time, and clearly identify symbols used in the corresponding legends.

(5) The meaning of a negative subjective value should be specifically stated. Is it the amount a subject would pay to avoid taking the considered pursuit?

As the reviewer intuits, negative subjective value can be considered the amount an agent ought be willing to pay to avoid taking the considered pursuit.

We now include the following lines in “The forgo decision can also be made from subjective value” section in reference to negative subjective value…

“A negative subjective value thus indicates that a policy of taking the considered pursuit would result in a global reward rate that is less than a policy of forgoing the considered pursuit. Equivalently, a negative subjective value can be considered the amount an agent ought be willing to pay to avoid having to take the considered pursuit.”

(6) Why do you define the discounting function as the normalized subjective value? This choice should be justified, via literature citations or a well-described logical argument.

The reward magnitude normalized subjective value-time function is commonly referred to as the temporal discounting function as it permits comparison of the discount rate isolated from a difference in reward magnitude and/or sign and is deeply rooted in historical precedent. As the reviewer points out, the term is overloaded, however, as investigations in which comparisons between the form of subjective value-time functions is not needed tend to refer to these functions as temporal discounting functions as well.

We make clear in the revised text in the introduction our meaning and use of the term, the justification in doing so, and its historical roots.

“Historically, temporal decision-making has been examined using a temporal discounting function to describe how delays in rewards influence their valuation. Temporal discounting functions describe the subjective value of an offered reward as a function of when the offered reward is realized. To isolate the form of discount rate from any difference in reward magnitude and sign, subjective value is commonly normalized by the reward magnitude when comparing subjective value-time functions (Strotz, 1956, Jimura, 2009). Therefore, we use the convention that temporal discounting functions are the magnitude-normalized subjective value-time function (Strotz, 1956).”

Special addition. In investigating the historical roots of the discounting function prompted by the reviewer, we learned (Grüne-Yanoff 2015) that it was Mazur that simply added the “1+k” in the denominator of the hyperbolic discounting function. Our derivation for the reward-rate optimal agent makes clear why apparent temporal discounting functions ought have this general form.

Therefore, we add the following to the “Hyperbolic Temporal Discounting Function section in the discussion…

“It was Ainslie (Ainslie, 1975) who first understood that the empirically observed “preference reversals” between SS and LL pursuits could be explained if temporal discounting took on a hyperbolic form, which he initially conjectured to arise simply from the ratio of reward to delay (Grüne-Yanoff 2015). This was problematic, however, on two fronts: (1) as the time nears zero, the value curve goes to infinity, and (2) there is no accommodation of differences observed within and between subjects regarding the steepness of discounting. Mazur (Mazur, 1987) addressed these issues by introducing 1 + k into the denominator, providing for the now standard hyperbolic discounting function, =svr=11+kt. Introduction of “1” solved the first issue, though “it never became fully clear how to interpret this 1” (Grüne-Yanoff 2015; interviewing Ainslie). Introduction of the free-fit parameter, k, accommodated the variability observed across and within subjects by controlling the curvature of temporal discounting, and has become widely interpreted as a psychological trait, such as patience, or willingness to delay gratification (Frederick et al., 2002).”

…continuing later in that section to explain why the reward-rate optimal agent would exhibit this general form…

“Regarding form, our analysis reveals that the apparent discounting function of a reward-rate-maximizing agent is a hyperbolic function…=1−ρout ∗tin rin 1+1tout tin 

…which resembles the standard hyperbolic discounting function, 11+kt, in the denominator, where k=1tout . Whereas Mazur introduced 1 + k to *t* in the denominator to (1) force the function to behave as *t* approaches zero, and (2) provide a means to accommodate differences observed within and between subjects, our derivation gives cause to the terms 1 and k, their relationship to one another, and to *t* in the denominator. First, from our derivation, “1” actually signifies taking *tout* amount of time expressed in units of *tout* (*tout*/*tout*=1) and adding it to *tin* amount of time expressed in units of *tout* (ie, the total time to make a full pass through the world expressed in terms of how the agent apportions its time under a policy of accepting the considered pursuit).”

Additional Correction. In revising the section, “Hyperbolic Temporal Discounting Functions” in the discussion, we also detected an error in our description of the meaning of suboptimal bias for SS. In the revision, the sentence now reads…

**“**More precisely, what is meant by this suboptimal bias for SS is that the switch in preference from LL to SS occurs at an outside reward rate that is lower—and/or an outside time that is greater —than what an optimal agent would exhibit.”

(7) Figure 15B should have negative axes defined for the pursuit's now negative reward.

Yes- excellent point.

To remove ambiguity regarding the valence of inside and outside reward magnitudes, we have changed all such figures so that the left hand y-axis is used to signify the outside reward magnitude and sign, and so that the right hand y-axis is used to signify the inside reward magnitude and sign.

With respect to the revision of original 15B, this change now makes clear that the inside reward label and numerics on the right hand side of the graph run from positive (top) to negative (bottom) values so that it can now be understood that the magnitude of the inside reward is negative in this figure (ie, a punishment). The left hand y-axis labeling the outside reward magnitude has numerics that run in the opposite direction, from negative (top) to positive (bottom). In this figure, the outside reward rate is positive whereas the inside reward rate is negative.

(8) When comparing your discounting function to the TIMERR and Heuristic models, it would be useful to include a schematic plot illustrating the different obtainable behaviors from all models rather than just telling the reader the differences.

We hold that the descriptions and references are sufficient to address these comparisons.

(9) I would strongly suggest cleaning up all appendices for notation…

The typographical errors that have been noted in these reviews have all been corrected. We believe the reviewer to be referring here to the manner that we had cross-referenced Equations in the appendices and main text which can lead to confusion between whether an equation number being referenced is in regard to its occurrence in the main text or its occurrence in the appendices.

In the revision, we eliminate numbering of equations in the appendices except where an equation occurs in an appendix that is referenced within the main text. In the main text, important equations are numbered sequentially and note the appendix from which they derive. If an equation in an appendix is referenced in the main text, it is noted within the appendix it derives.

…and replacing some of the small equation manipulations with written text describing the goal of each derivation.

To increase clarity, we have taken the reviewer’s helpful suggestion, adding helper text in the appendices were needed, and have bolded the equations of importance within the Appendices (rather than removing equation manipulations making clear steps of derivation).

(10) I would suggest moving the table in Appendix 11 to the main text where misestimation is referenced.

So moved. This appendix now appears in the main text as table 1 “Definitions of misestimating global reward rate-enabling parameters”.

**Reviewer #3 (Public review):**
One broad issue with the paper is readability. Admittedly, this is a complicated analysis involving many equations that are important to grasp to follow the analyses that subsequently build on top of previous analyses.But, what's missing is intuitive interpretations behind some of the terms introduced, especially the apportionment cost without referencing the equations in the definition so the reader gets a sense of how the decision-maker thinks of this time cost in contrast with the opportunity cost of time.

We thank the reviewer for encouraging us to formulate a succinct and intuitive statement as to the nature of apportionment cost. We thank the reviewer for pressing for a succinct and intuitive verbal description.

We added the following succinct verbal description of apportionment cost… “Apportionment cost is the difference in reward that can be expected, on average, between a policy of taking versus a policy of not taking the considered pursuit, over a time equal to its duration.” This definition appears in a new paragraph (as below) describing apportionment cost in the results section “Time’s cost: opportunity & apportionment costs determine a pursuit’s subjective value”, and is accompanied by equations for apportionment cost, and a figure giving its geometric depiction (Figure 5). We also expanded original figure 5 and its legend (so as to illustrate the apportionment scaling factor and the apportionment cost), and its accompanying main text, to further illustrate and clarify apportionment cost, and its relationship to opportunity cost, and time’s cost.

“What, then, is the amount of reward by which the opportunity cost-subtracted reward is scaled down to equal the *sv* of the pursuit? This amount is the apportionment cost of time. The apportionment cost of time (height of the brown vertical bar, Figure 5F) is the global reward rate after taking into account the opportunity cost (slope of the magenta-gold dashed line in Figure 5F) times the time of the considered pursuit. Equally, the difference between the inside and outside reward rates, times the time of the pursuit, is the apportionment cost when scaled by the pursuit’s weight, i.e., the fraction that the considered pursuit is to the total time to traverse the world (Equation 9, right hand side). From the perspective of decision-making policies, apportionment cost is the difference in reward that can be expected, on average, between a policy of taking versus a policy of not taking the considered pursuit, over a time equal to its duration (Equation 9 center, Figure 5F).=tin(ρg−ρout)=tin(rintin−routtout)(tintin+tout)

Equation 9. Apportionment Cost.

While this difference is the apportionment cost of time, the opportunity cost of time is the amount that would be expected from a policy of not taking the considered pursuit over a time equal to the considered pursuit’s duration. Together, they sum to Time’s Cost (Figure 5G). Expressing a pursuit’s worth in terms of the global reward rate obtained under a policy of accepting the pursuit type (Figure 5 left column), or from the perspective of the outside reward and time (Figure 5 right column), are equivalent. However, the latter expresses *sv* in terms that are independent of one another, conveys the constituents giving rise to global reward rate, and provides the added insight that time’s cost comprises an apportionment as well as an opportunity cost.”

The above definition of apportionment cost adds to other stated relationships of apportionment cost found throughout the paper (original lines 434,435,447,450).

Re-analysis of some existing empirical data through the lens of their presented objective functions, especially later when they describe sources of error in behavior.

Our objective was not to fit experimentally observed data, as is commonly the goal of implementation/computational models. Rather, as a theory, our objective is to rationalize the broad, curious, and well-established pattern of temporal decision-making behaviors under a deeper understanding of reward-rate maximization, and from that understanding, identify the nature of the error being committed by whatever learning algorithm and representational architecture is actually being used by humans and animals. In doing so, we make a number of important contributions. By identifying and analyzing reward-rate-maximizing equations, we (1) provide insight into what composes time’s cost and how the temporal structure of the world in which it is embedded (its ‘context’) impacts the value of a pursuit, (2) rationalize a diverse assortment of temporal decision-making behaviors (e.g., Hyperbolic discounting, the Magnitude Effect, the Sign Effect, and the Delay effect), explaining them with no assumed free-fit parameter, and then, by analyzing error in parameters enabling reward-rate maximization, (3) identify the likely source of error and propose the Malapportionment Hypothesis. The Malapportionment Hypothesis identifies the underweighting of a considered pursuit’s “outside”, and not error in pursuit’s reward rates, as the source of error committed by humans and animals. It explains why animals and humans can present as suboptimally ‘impatient’ in Choice, but as optimal in Forgo. At the same time, it concords with numerous and diverse observations in decision making regarding whether to initiate a pursuit. The nature of this error also, then, makes numerous predictions. These insights inform future computational and experimental work by providing strong constraints on the nature of the algorithm and representational architecture used to learn and represent the values of pursuits. Rigorous test of the Malapportionment Hypothesis will require wholly new experiments.

In the revision, we also now emphasize and add predictions of the Malapportionment Hypothesis, augmenting its figure (Figure 21), its legend, and its paragraphs in the discussion.

“We term this reckoning of the source of error committed by animals and humans the Malapportionment Hypothesis, which identifies the underweighting of the time spent outside versus inside a considered pursuit but not the misestimation of pursuit rates, as the source of error committed by animals and humans (Figure 21). This hypothesis therefore captures previously published behavioral observations (Figure 21A) showing that animals can make decisions to take or forgo reward options that optimize reward accumulation (Krebs et al., 1977; Stephens and Krebs, 1986; Blanchard and Hayden, 2014), but make suboptimal decisions when presented with simultaneous and mutually exclusive choices between rewards of different delays (Logue et al., 1985; Blanchard and Hayden, 2015; Carter and Redish, 2016; Kane et al., 2019). The Malapportionment Hypothesis further predicts that apparent discounting functions will present with greater curvature than what a reward-rate-maximizing agent would exhibit (Figure 21B). While experimentally observed temporal discounting would have greater curvature, the Malapportionment Hypothesis also predicts that the Magnitude (Figure 21C) and Sign effect (Figure 21D) would be less pronounced than what a reward-rate-maximizing agent would exhibit, with these effects becoming less pronounced the greater the underweighting. Finally, with regards to the Delay Effect (Figure 21E), the Malapportionment Hypothesis predicts that preference reversal would occur at delays greater than that exhibited by a reward-rate-maximizing agent, with the delay becoming more pronounced the greater the underweighting outside versus inside the considered pursuit by the agent.”

**Reviewer #3 Recommendations:**
As mentioned above, the readability of this paper should be improved so that the readers can follow the derivations and your analyses better. To this end, careful numbering of equations, following consistent equation numbering formats, and differentiating between appendix referencing and equation numbering would have gone a long way in improving the readability of this paper. Some specific questions are noted below.

To increase clarity, in the revision we eliminated numbering of equations in the appendices except where an equation occurs in an appendix that is referenced within the main text. In the main text, important equations are thus numbered sequentially as they appear and note the appendix from which they derive. If an equation in an appendix is referenced in the main text, it is noted within the appendix it derives.

(1) In general, it is unclear what the default pursuit is. From the schematic on the left (forgo decision), it appears to be the time spent in between reward-giving pursuits. However, this schematic also allows for smaller rewards to be attained during the default pursuit as do subsequent equations that reference a default reward rate. Here is where an example would have really benefited the authors in getting their point across as to what the default pursuit is in practice in the forgo decisions and how the default reward rate could be modulated.

**(**1) The description of the default pursuit has been modified in section “Forgo and Choice decision topologies” to now read… “After either the conclusion of the pursuit, if accepted, or immediately after rejection, the agent returns to a pursuit by default (the “default” pursuit). This default pursuit effectively can be a waiting period over which reward could be received, and reoccurs until the next pursuit opportunity becomes available.” (2) Additionally, helper text has been added to Ap1 regarding the meaning of time and reward spent in the default pursuit. Finally, (3) new figures concerning n-pursuits occurring at the same (Supplement 1) or different (Supplement 2) frequencies from a default pursuit is now added, providing examples as suggested by the reviewer.

(2) I want to clarify my understanding of the topologies in Figure 1. In the forgo, do they roam in the "gold" pursuit indefinitely before they are faced with the purple pursuit? In general, comparing the 2 topologies, it seems like in the forgo decision, they can roam indefinitely in the gold topology or choose the purple but must return to the gold.

The reviewer’s understanding of the topology is correct. The agent loops across one unit time in the default gold pursuit indefinitely, though the purple pursuit (or any pursuit that might exist in that world) occurs on exit from gold at its frequency per unit time. The default gold pursuit will then itself have an average duration in units of time spent in gold. As the reviewer states, the agent can re-enter into gold from having exited gold, and can enter gold from having exited purple, but cannot re-enter purple from having exited purple; rather, it must enter into the default pursuit.

…Another point here is that this topology is highly simplified (only one considered pursuit). So it may be helpful to either add a schematic for the full topology with multiple pursuits or alternatively, provide the corresponding equations (at least in appendix 1 and 2) for the simplified topology so you can drive home the intuition behind derived expressions in these equations.

We understand the reviewer to be noting that, while, the illustrated example is of the simple topology, the mathematical formulation handles the case of n-number pursuits, and that illustrating a world in which there are a greater number of pursuits, corresponding to original appendices 1&2, would assist readers in understanding the generality of these equations.

An excellent suggestion. We have now n-pursuit world illustrations where each pursuit occurs at the same (Supplemental Figure 1) and at different frequencies (Supplemental Figure 2) to the manuscript, and have added text to assist in understanding the form of the equation and its relationship to unit time in the default pursuit in the main and in the appendices.

(3) In Equation and Appendix 1, there are a few things that are unclear. Particularly, why is the expected time of the default option E(t_default) = 1/(∑_(i=1)^n f_i)? Similarly, why is the E(r_default) = ρ_d/(∑_(i=1)^n f_i)? Looking at the expression for E(r_default), it implies that across all pursuits 1 through n, the default option is encountered only once. Ultimately, in Equation 1.4, (and Equation 1), the units of the two terms in the numerator don't seem to match. One is a reward rate (ρ_d) and the other is a reward value. This is the most important equation of the paper since the next several equations build upon this. Therefore, the lack of clarity here makes the reader less likely to follow along with the analysis in rigorous detail. Better explanations of the terms and better formatting will help alleviate some of these issues.

The equation is formulated to calculate the average reward received and average time spent per unit time spent in the default pursuit. So, *fi* is the encounter rate of pursuit *i* for one unit of time spent in the default pursuit. Added to the summation in the numerator we have the average reward obtained in the default pursuit per unit time (ρd∗1) and in the denominator we have the time spent in the default pursuit per unit time (1).

Text explaining the above equation has been added to Ap 1.

(4) In equation and appendix 2, I'm trying to relate the expressions for t_out and r_out to the definitions "average time spent outside the considered pursuit". If I understand the expression in Equation 2.4 on the right-hand side, the numerator is the total time spent in all of the pursuits in the environment and the denominator refers to the number of times the considered pursuit is encountered. It is unclear as to why this is the average time spent outside the considered pursuit. In my mind, the expression for average time spent outside the considered pursuit would look something like t_out=1+ ∑_(i≠in)〖p_i t_i〗=1+ ∑_(i≠in)〖f_i/(∑_(j=1)^n f_j) * t_i〗. It is unclear how these expressions are then equivalent.

Regarding the following equation,tout =∑i≠infiti+1fin 

*fi* is the probability that pursuit *i* will be encountered during a single unit of time spent in the default pursuit. The numerator of the expression is the average amount of time spent across all pursuits, excepting the considered pursuit, per unit time spent in the default pursuit. Note that the + 1 in the numerator is accounting for the unit of time spent in the default pursuit and is added outside of the sum. Since *fin* is the probability that the considered pursuit will be encountered per unit of time spent in the default pursuit, 1fin is the average amount of time spent in the default pursuit between encounters of the considered pursuit. By multiplying the average time spent across all outside pursuits per unit of time in the default pursuit by the average amount of time spent in the default pursuit between encounters of the considered pursuit, we get the average amount of time spent outside the considered pursuit per encounter of the considered pursuit. This is calculated as if the pursuit encounters are mutually exclusive within a single unit of time spent within the default pursuit, as this is the case as the length of our unit time (delta t) approaches zero.

The above text explaining the equation has been added to Ap 2.

(5) In Figure 3, one huge advantage of this separation into in-pursuit and out-of-pursuit patches is that the optimal reward-rate maximizing rule becomes one that compares ρ_in and ρ_out. This contrasts with an optimal foraging rule which requires comparing to the global reward rate and therefore a circularity in solution. In practice, however, it is unclear how ρ_out will be estimated by the agent.

How, in practice, a human or animal estimates the reward rates―be they the outside and/or global reward rate under a policy of accepting a pursuit―is the crux of the matter. This work identifies equations that would enable a reward-rate maximizing agent to calculate and execute optimal policies and emphasizes that the effective reward rates and weights of pursuits must be accurately appreciated for global reward rate optimization. In so doing, it makes a reckoning of behaviors commonly but erroneously treated as suboptimal. Then, by examining the consequences of misestimation of these enabling parameters, it identifies mis-weighting pursuits as the nature of the error committed by whatever algorithm and representational architecture is being used by humans and animals (the Malapportionment Hypothesis). This curious pattern identified and analyzed in this work thus provides a clue into the nature of the learning algorithm and means of representing the temporal structure of the environment that is used by humans and animals―the subject of future work.

We note, however, that we do discuss existing models that grapple with how, in practice, how a human or animal may estimate the outside reward rate. Of particular importance is the TIMERR model, which estimates the outside reward rate from its past experience, and can make an accounting of many qualitative features widely observed. However, while appealing, it would mix prior ‘in’ and ‘outside’ experiences within that estimate, and so would fail to perform forgo tasks optimally. Something is still amiss, as this work demonstrates.

(6) The apportionment time cost needs to be explained a little bit more intuitively. For instance, it is clear that the opportunity cost of time is the cost of not spending time in the rest of the environment relative to the current pursuit. But given the definition of apportionment cost here in lines 447- 448 "The apportionment cost relates to time's allocation in the world: the time spent within a pursuit type relative to the time spent outside that pursuit type, appearing in the denominator." The reference to the equation (setting aside the confusion regarding which equation) within the definition makes it a bit harder to form an intuitive interpretation of this cost. Please reference the equation being referred to in lines 447-448, and again, an example may help the authors communicate their point much better

We thank the reviewer for pressing on this critical point.

Action: We added the following succinct verbal description of apportionment cost… “Apportionment cost is the difference in reward that can be expected, on average, between a policy of taking versus a policy of not taking the considered pursuit, over a time equal to its duration.” This definition appears in a new paragraph (as below) describing apportionment cost in the results section “Time’s cost: opportunity & apportionment costs determine a pursuit’s subjective value”, and is accompanied by equations for apportionment cost, and a figure giving its geometric depiction (Figure 5).

“What, then, is the amount of reward by which the opportunity cost-subtracted reward is scaled down to equal the *sv* of the pursuit? This amount is the apportionment cost of time. The apportionment cost of time (height of the brown vertical bar, Figure 5F) is the global reward rate after taking into account the opportunity cost (slope of the magenta-gold dashed line in Figure 5F) times the time of the considered pursuit. Equally, the difference between the inside and outside reward rates, times the time of the pursuit, is the apportionment cost when scaled by the pursuit’s weight, i.e., the fraction that the considered pursuit is to the total time to traverse the world (Equation 9, right hand side). From the perspective of decision-making policies, apportionment cost is the difference in reward that can be expected, on average, between a policy of taking versus a policy of not taking the considered pursuit, over a time equal to its duration (Equation 9 center, Figure 5F).=tin(ρg−ρout)=tin(rintin−routtout)(tintin+tout)

Equation 9. Apportionment Cost.

While this difference is the apportionment cost of time, the opportunity cost of time is the amount that would be expected from a policy of not taking the considered pursuit over a time equal to the considered pursuit’s duration. Together, they sum to Time’s Cost (Figure 5G). Expressing a pursuit’s worth in terms of the global reward rate obtained under a policy of accepting the pursuit type (Figure 5 left column), or from the perspective of the outside reward and time (Figure 5 right column), are equivalent. However, the latter expresses *sv* in terms that are independent of one another, conveys the constituents giving rise to global reward rate, and provides the added insight that time’s cost comprises an apportionment as well as an opportunity cost.”

(7) The analyses in Figures 6 and 7 give a nice visual representation of how the time costs are distributed as a function of outside reward and time spent. However, without an expression for apportionment cost it is hard to intuitively understand these visualizations. This also relates to the previous point of requiring a more intuitive explanation of apportionment costs in relation to the opportunity cost of time. Based on my quick math, it seems that an expression for apportionment cost would be as follows: (r_in- ρ_out*t_in)*(t_in⁄t_out)/(t_in⁄t_out +1). The condition described in Figure 7 seems like the perfect place to compute the value of just apportionment cost when the opportunity cost is zero. It would be helpful to introduce the equation here.

We designed original figure 7, as the reviewer appreciates, to emphasize that time has a cost even when there is no opportunity cost, being due entirely to the apportionment cost of time.

We now provide the mathematical expression of apportionment cost and apportionment scaling in Figure 5, the point in the main text of its first occurrence.=tin(ρg−ρout)=tin(rintin−routtout)(tintin+tout)

…and have expanded original figure 5, its legend (so as to illustrate the apportionment scaling factor and the apportionment cost), and its accompanying main text, to further illustrate and clarify apportionment cost, and its relationship to opportunity cost, and time’s cost.

(8) The analysis regarding choice decisions is relatively straightforward, pending the concerns for the main equations listed above for the forgo decisions. Legends certainly would have helped me grasp Figures 10-12 better.

We believe the reviewer is referring to missing labels for the smaller-sooner pursuit, and the larger-later pursuit in these figures? We used the same conventions as in Figure 9, but we see now that adding these labels to these figures would be helpful, and add them in the revision.

We have now added to the figures themselves figure legends indicating the smaller-sooner pursuit and the larger-later pursuit. We have also added to the main text to emphasize the points made in these figures regarding the impact of opportunity cost and apportionment cost.

(9) The derivation of the temporal discounting function from subjective reward rate is much appreciated as it provides further evidence for potential equivalence between reward rate optimization and hyperbolic discounting, which is known to explain a slew of decision-making behaviors in the economics literature.

We thank and greatly appreciate the reviewer for this recognition.

In response to the reviewer’s comment, we have added text that further relates reward rate optimization to hyperbolic discounting…

(1) We add discussion of how our normative derivation gives explanation to Mazur’s *ad hoc* addition of 1 + *k* to Ainslie’s reward/time hyperbolic discounting conception. See new first paragraph under “Hyperbolic Temporal Discounting Functions” for the historical origins of the standard hyperbolic equation (which are decidedly *not* normatively derived). And then see our discussion (new second paragraph in sections “The apparent discounting function of global….”) of how our normative derivation gives explanation to “1”, “*k”,* and their relationship to each other.

(2) We add explicit treatment of the Delay Effect in a new “The Delay Effect” section of the results along with a figure, and in its corresponding Discussion section.

Minor comments:(1) Typo in equation 2, should be t_i in the denominator within the summation, not r_i .

We thank the reviewer for catching this typo, and have corrected it in the revision.

(2) Before equation 6, typo when defining ρ_in = r_in/(t_in.). Should be t_in in the denominator, not r_out.

We thank the reviewer for catching this typo, and have corrected it in the revision.

(3) Please be consistent with equation numbers, placement of equation references, and the reason for placing appendix numbers. This will improve readability immensely.

To increase clarity, in the revision we eliminated numbering of equations in the appendices except where an equation occurs in an appendix that is referenced within the main text. In the main text, important equations are thus numbered sequentially and note the appendix from which they derive. If an equation in an appendix is referenced in the main text, it is noted within the appendix it derives.

(4) Line 505 - "dominants" should be dominates.

Typo fixed as indicated

(5) Figures 10-12: add legends to the figures.

Now so included.

(6) Lines 701-703: please rewrite the equation separately. It is highly unclear what rt is here.

We thank the reviewer for bringing attention to this error. The error arose in converting from Google Sheets to Microsoft Word.

The equation has now been corrected.

Additional citations noted in reply and appearing in Main text

Ainslie, George. 1975. “Specious Reward: A Behavioral Theory of Impulsiveness and Impulse Control.” Psychological Bulletin 59: 257–72.

Frederick, Shane, George Loewenstein, Ted O. Donoghue, and T. E. D. O. Donoghue. 2002. “Time Discounting and Time Preference : A Critical Review.” Journal of Economic Literature 40: 351–401.

Gibbon, John. 1977. “Scalar Expectancy Theory and Weber’s Law in Animal Timing.” Psychological Review 84: 279–325.

Green, Leonard, Nathanael Fristoe, and Joel Myerson. 1994. “Temporal Discounting and Preference Reversals in Choice between Delayed Outcomes.” Psychonomic Bulletin & Review 1: 383–89.

Grüne-Yanoff, Till. 2015. “Models of Temporal Discounting 1937-2000: An Interdisciplinary Exchange between Economics and Psychology.” Science in Context 28 (4): 675–713.

Jimura, Koji, Joel Myerson, Joseph Hilgard, Todd S. Braver, and Leonard Green. 2009. “Are People Really More Patient than Other Animals? Evidence from Human Discounting of Real Liquid Rewards.” Psychonomic Bulletin & Review 16: 1071–75.

Kalenscher, Tobias, and Cyriel M. A. Pennartz. 2008. “Is a Bird in the Hand Worth Two in the Future? The Neuroeconomics of Intertemporal Decision-Making.” Progress in Neurobiology 84 (3): 284–315.

Kirby, Kris N., and R. J. Herrnstein. 1995. “Preference Reversals Due to Myopic Discounting of Delayed Reward.” Psychological Science 6 (2): 83–89.

Mazur, James E. 1987. “An Adjusting Procedure for Studying Delayed Reinforcement.” In The Effect of Delay and of Intervening Events on Reinforcement Value., 55–73. Quantitative Analyses of Behavior, Vol. 5. Hillsdale, NJ, US: Lawrence Erlbaum Associates, Inc.

McNamara, John. 1982. “Optimal Patch Use in a Stochastic Environment.” Theoretical Population Biology 21 (2): 269–88.

Rosati, Alexandra G., Jeffrey R. Stevens, Brian Hare, and Marc D. Hauser. 2007. “The Evolutionary Origins of Human Patience: Temporal Preferences in Chimpanzees, Bonobos, and Human Adults.” Current Biology: CB 17: 1663–68.

Strotz, R. H. 1956. “Myopia and Inconsistency in Dynamic Utility Maximization.” The Review of Economic Studies 23: 165–80.